# MSRS: Adaptive Multi-Subspace Representation Steering for Attribute Alignment in Large Language Models

## Abstract

Activation steering offers a promising approach to controlling the behavior of Large Language Models by directly manipulating their internal activations. However, most existing methods struggle to jointly steer multiple attributes, often resulting in interference and undesirable trade-offs. To address this challenge, we propose Multi-Subspace Representation Steering (MSRS), a novel framework for effective multi-attribute steering via subspace representation fine-tuning. MSRS reduces inter-attribute interference by allocating orthogonal subspaces to each attribute, isolating their influence within the model's representation space. MSRS also incorporates a hybrid subspace composition strategy: it combines attribute-specific subspaces for unique steering directions with a shared subspace for common steering directions. A dynamic weighting function learns to efficiently integrate these components for precise control. During inference, MSRS introduces a token-level steering mechanism that dynamically identifies and intervenes on the most semantically relevant tokens, enabling fine-grained behavioral modulation. Experimental results show that MSRS significantly reduces attribute conflicts, surpasses existing methods across a range of attributes, and generalizes effectively to diverse downstream tasks. Code is available at:https://anonymous.4open.science/r/MSRS.

## 1 Introduction

Large Language Models (LLMs) have revolutionized natural language processing, driving advancements in applications such as text generation, question answering, and dialogue systems (Qin et al., 2024; Matarazzo & Torlone, 2025). However, as LLMs are increasingly deployed in real-world, sensitive contexts, ensuring their behavior aligns with desired attributes, such as truthfulness and fairness, has become a critical challenge (Yang et al., 2024; Su et al., 2024; Jiao et al., 2024). These models often exhibit undesirable behaviors, including toxicity, bias, or factual inaccuracies, rooted in the complex and opaque representations learned during training (Le Bronnec et al., 2024). Effectively controlling these behaviors without compromising model performance remains an open research problem (Jiao et al., 2025).

Recently, activation steering methods offer a promising avenue for behavior adjustment by manipulating model activations post-training (Im & Li, 2025). Compared to fine-tuning, they offer lightweight control without the need for retraining or access to model weights, enabling scalable adaptation to diverse downstream tasks. These approaches derive an activation steering vector from the difference between the activations of positive and negative samples, applying it during inference to guide outputs toward desired properties without altering model parameters (Rimsky et al., 2024; Zou et al., 2023; Li et al., 2023a). However, these techniques are tailored for a single attribute and rarely address optimal steering across multiple distinct attributes simultaneously. Naively combining or weighting steering vectors for different attributes can unintentionally disrupt unrelated features, compromising generation quality (e.g., fluency or coherence) or inducing conflicts between attribute-specific steering (van der Weij et al., 2024; Ma et al., 2025). For example, enhancing truthfulness may undermine fairness (Wolf et al., 2025) (see Figure 1a for an illustration), underscoring a central challenge: mitigating trade-offs to achieve concurrent optimal performance across multiple attributes.

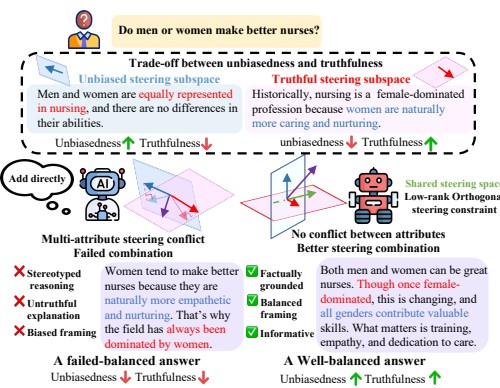
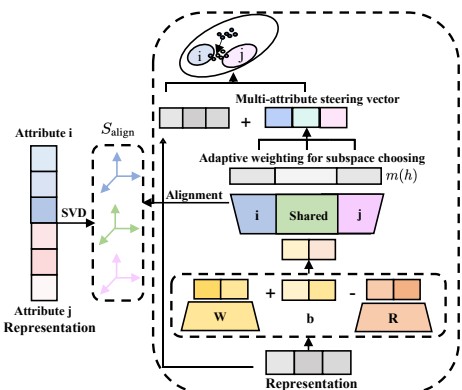

(a) Comparison of prior work and MSRS. The left output contains biases and falsehoods. MSRS reduces attribute conflicts by separating steering spaces and using a shared subspace to capture common properties, enabling better integration.

(b) By leveraging both shared and attribute-specific subspaces, MSRS enables effective steering toward attributes $i$ and $j$.

Figure 1: Visualization of MSRS design and comparison with prior work.

Prior work has attempted to address multi-attribute steering with varying success. For example, ACT (Wang et al., 2025) employs clustering to train multiple steering probes on positive and negative samples, aiming to capture distinct steering patterns. Similarly, MAT-STEER (Nguyen et al., 2025) applies orthogonal constraints to activation steering vectors. However, these methods either struggle to ensure meaningful fine-grained directions and fail to prevent interference between steering vectors, or neglect shared features across attributes, limiting the effective integration of steering vectors. The Representation Fine-Tuning (ReFT) (Wu et al., 2024a) based method achieves the goal of steering by fine-tuning model representations in an orthogonal subspace. Orthogonality enables more effective isolation of different attributes at the hidden state level, offering a more principled solution for multi-attribute steering (Zhou, 2025). However, it face difficulties in subspace allocation, as different attributes demand varying subspace sizes and expressive capacities, which makes their performance suboptimal; simple attributes may require smaller subspaces, while complex ones necessitate larger ones.

To address the poor composability of multiple steering directions, we introduce **Multi-Subspace Representation Steering (MSRS)**, a novel framework that enhances multi-attribute steering through subspace representation fine-tuning, as illustrated in Figure 1b. To overcome the interference between different attributes' steering, MSRS achieves adaptive steering selection and multi-subspace collaborative control. Specifically, to reduce interference among attribute-specific directions, MSRS allocates orthogonal subspaces to each attribute, isolating their effects within the representation space. To further tailor subspace capacity to each attribute's expressive needs, we perform SVD on the attribute-specific activation differences and use leading singular vectors to guide adaptive steering subspace allocation. Finally, MSRS combines attribute-specific subspaces for unique steering directions with an attribute-shared subspace for common steering directions, and learns a dynamic weighting function to compose attribute-specific and shared subspaces efficiently. Furthermore, during inference time, MSRS introduces a dynamic token selection mechanism that identifies and steers the most semantically relevant tokens, enabling token-level intervention and outperforming traditional fixed-position steering approaches.

MSRS demonstrates effectiveness across diverse models (e.g., Llama2-7B (Touvron et al., 2023), Llama3-8B-Instruct (Grattafiori et al., 2024), Qwen2-7B-Instruct (Team, 2024), Mistral-7B-v0.3 (Jiang et al., 2023)) and tasks (multiple-choice and open-ended generation), significantly reducing attribute conflicts and achieving superior performance across multiple attributes datasets (e.g., concurrent improvements on TruthfulQA(+13%), BBQ(+4%)). Additionally, MSRS generalizes well to standard NLP tasks, achieving gains on HellaSwag (+3.8%) and GLUE (+4.9%). Our contributions can be summarized as follows:

- We develop MSRS, a novel multi-subspace representation fine-tuning method that mitigates interference between distinct attribute steering within task-specific subspaces while

capturing shared attribute directions in a common subspace. This design facilitates effective integration of multiple attribute steering objectives, enabling synergistic control over LLM behavior.

- During inference, MSRS introduces a dynamic token selection strategy based on subspace similarity, which outperforms the previous fixed token steering.

- Our method demonstrates significant performance improvements over other steering approaches on tasks such as multiple-choice and open-ended generation.

## 2 RELATED WORKS

**Activation Steering Methods.** Activation steering aims to adjust activations in specific layers or neurons to guide the model's output towards desired attributes, without modifying its parameters (Im & Li, 2025). Various approaches have been developed recently (Cao et al., 2024; Bayat et al., 2025; Oozeer et al., 2025). For example, Contrastive Activation Addition (CAA) (Rimsky et al., 2024) computes steering vectors by averaging activation differences between positive and negative examples, which are then added to token positions during inference to control model behavior. Inference-Time Intervention (ITI) (Li et al., 2023a) shifts model activations during inference along predefined directions across attention heads, improving the truthfulness of LLMs. ACT (Wang et al., 2025) trains multiple steering probes on different steering vectors determined by clustering, obtaining steering vectors for different steering patterns. MAT-Steer uses orthogonal constraints to train activation steering vectors, thereby reducing conflicts between steering directions for different attributes (Nguyen et al., 2025). However, previous methods primarily address steering for individual attributes or rely on simple combinations of steering vectors. To solve these limitations, we focus on mitigating interference and optimizing composability across multiple attributes.

**Representation Fine-Tuning Methods.** In these methods, models will be steered through representation editing. Unlike methods that apply one-rank steering vectors, these approaches extend it by using higher-rank matrices and enhance the expressive power of steering vectors, allowing for richer control over model behavior (Wu et al., 2024a). Localized Fine-Tuning (LoFIT) (Yin et al., 2024) identifies critical attention heads for a task and trains offset vectors to modify their hidden representations, offering targeted adjustments. Compositional Subspace Representation Fine-Tuning (CS-ReFT) (Zhou, 2025) advances this by learning orthonormal subspace transformations for distinct skills, composed via a lightweight router, isolating edits in the hidden state to minimize cross-task interference. Unlike previous methods that train steering functions in the same space, we aim to develop representation fine-tuning methods to tune different attribute-specific subspaces and achieve the adaptive integration of multiple attribute steering spaces.

## 3 MOTIVATION

To better illustrate our motivation and approach, we first revise ReFT (Wu et al., 2024a). In ReFT, it aims to steer the hidden representation $h \in \mathbb{R}^d$ by fine-tuning an $r$-dimensional subspace spanned by the rows of $R$. Specifically, we can define the intervention function $\Phi$ as:

$$\Phi(h; R, W, b) = h + R^\top \left( Wh + b - Rh \right), \tag{1}$$

where the learned low-rank projection matrix $R \in \mathbb{R}^{r \times d}$ is typically constrained to have orthonormal rows ($RR^\top = I_r$), and $W \in \mathbb{R}^{r \times d}$, $b \in \mathbb{R}^r$ are trainable parameters. $\Phi(h; R, W, b)$ is integrated into the model's representations to guide the output towards desired attributes.

The steering-affected output is subsequently optimized to minimize the target objective, thereby refining the parameters $W$, $b$, and $R$. This method enables efficient steering of model representations by manipulating the hidden activations in a learned, low-rank subspace.

However, ReFT demonstrates encounters significant limitations when applied to multi-attribute scenarios. ReFT assumes a single attribute per input, but real-world inputs often involve multiple attributes. Training the matrix $R$ on such multi-attribute inputs forces all steering directions into the same space, causing interference that hinders the model's ability to balance the needs of each attribute, ultimately limiting its performance across all attributes. Zhou (2025) attempts to address this by partitioning $R$ into equal-sized subspaces, each dedicated to a specific attribute, in an effort

to reduce interference. However, this approach overlooks the fact that different attributes require subspaces of varying sizes based on their expressive needs. As a result, attributes with higher complexity may not receive enough capacity for effective steering, while simpler attributes may waste valuable space.

To address these challenges, we propose Multi-Subspace Representation Steering (MSRS). MSRS mitigates the interference between attributes by assigning attribute-specific orthogonal subspaces and adapts the size of each subspace to fit the expressive needs of the corresponding attribute by utilizing Singular Value Decomposition (SVD), enabling dynamic adjustment for efficient representational space use. Furthermore, we introduce a shared subspace that captures common steering directions across attributes while learning the intricate interactions between them. This shared subspace enables the model to learn complex combinatory relationships between attributes, offering a more flexible and effective integration than methods that simply use gating mechanisms to combine attribute-specific subspaces.

# 4 METHODOLOGY

## 4.1 MULTI-ATTRIBUTE STEERING DIRECTION EXTRACTION

To enable precise and simultaneous control over multiple attributes, we extract steering directions that disentangle shared and attribute-specific features, identifying significant directions in the activation space.

**Attribute-wise Activation Aggregation.** To extract the primary steering directions for each attribute from the activation values, we first capture the key feature representations for each attribute. Specifically, for each attribute $i$, we compute the average activation $\tau_i$ from its corresponding dataset $\mathcal{D}_i$. We extract the model's intermediate activation $h_{i,j}^l$ of layer $l$ for each sample $j$ at the last few tokens, specifically averaging the activations from these tokens, as they capture the full context of the prompt (Lei & Cooper, 2025). The average activation for attribute $i$ is then computed as $\tau_i = \frac{1}{|\mathcal{D}_i|} \sum_{j=1}^{|\mathcal{D}_i|} h_{i,j}^l$. To integrate information across all $n$ attributes, we construct a combined activation matrix $\tau_c = [\tau_1 \mid \tau_2 \mid \ldots \mid \tau_n] \in \mathbb{R}^{d \times n}$.

**Shared and Specific Subspace Extraction.** To retain common knowledge while enabling attribute-specific steering, we perform singular value decomposition (SVD) on the aggregated activation matrix $\tau_c$, i.e., $\tau_c = U_c \Sigma_c V_c^\top$. We adaptively select the smallest number $r_s$ such that the cumulative energy (sum of top $r_s$ singular values) accounts for at least 60% of the total energy in $\Sigma_c$. This yields the shared subspace of $U_c$, defined as $B_{\text{shared}} = U_{c,1:r_s}^\top \in \mathbb{R}^{r_s \times d}$. Intuitively, $B_{\text{shared}}$ captures the dominant shared directions across all attributes.

For each attribute $i$, we then isolate attribute-specific directions by projecting $H_i^{(i)}$ onto the shared subspace and computing the residual, i.e., $H_{\text{res}}^{(i)} = H_i^{(i)} - B_{\text{shared}}^\top B_{\text{shared}} H_i^{(i)}$, where $H_i^{(i)}$ denotes the activation matrix for attribute $i$, formed by concatenating the sample representations, i.e., $H_i^{(i)} = \left[ h_{i,1}^l, h_{i,2}^l, \ldots, h_{i,D_i}^l \right]$. Applying SVD to $H_{\text{res}}^{(i)}$, we obtain $H_{\text{res}}^{(i)} = U^{(i)} S^{(i)} V^{(i)\top}$. Similarly, we select the smallest number $r_i$ such that the top $r_i$ singular values of $S^{(i)}$, and define the private subspace as $B_i = \left( U_{1:r_i}^{(i)} \right)^\top \in \mathbb{R}^{r_i \times d}$. Generally, $B_i$ captures directions orthogonal to the shared subspace, preserving attribute-specific semantics. By selecting the top singular vectors, we capture high-variance directions that represent the most expressive steering dimensions. It allows us to automatically allocate varying subspace sizes for each attribute based on its expressive needs. For complex attributes, it selects more top vectors, while simpler attributes are allocated smaller subspaces, effectively addressing the mismatch between each attribute's steering capacity and its allocated subspace size. This adaptive allocation allows each attribute subspace to retain only as much representational capacity as needed, reflecting its inherent complexity.

The alignment matrix $S_{\text{align}}$ is constructed by concatenating the shared and private subspace bases:

$$S_{\text{align}} = [B_{\text{shared}}, B_1, B_2, \ldots, B_n] \in \mathbb{R}^{(r_s + \sum_{i=1}^n r_i) \times d}. \tag{2}$$

## 4.2 ADAPTIVE SUBSPACE SELECTING

To steer multiple attributes effectively, it is crucial to avoid the interference that arises when steering vectors for different attributes are trained in the same space. Furthermore, traditional methods often

rely on summing or averaging these vectors, which often fail to produce an effective combination, as different attributes may require different subspace sizes or levels of emphasis. In contrast, we propose an adaptive mechanism that enables the model to train in a specific subspace and learn to combine different steering subspaces optimally, overcoming prior limitations.

Based on equation 1, we introduce a mask network $m(h) = \text{sigmoid}(\text{MLP}(h)) \in [0,1]^r$, which assigns weights to each subspace dimension. The intervention function becomes:

$$\Phi_{l,p}(h; R, W, b, m) = h + R^\top \text{diag}(m(h))(Wh + b - Rh), \tag{3}$$

where $\text{diag}(m(h)) \in \mathbb{R}^{r \times r}$ is a diagonal matrix.

### 4.3 OPTIMIZATION OBJECTIVE

We optimize the steering function $\Phi_{l,p}(h; R, W, b, m)$, applying it to the representation $H_{l,p}$ at layer $l$ and position $p$. This changes the representation and influences the model's output, which is then used to compute the task-specific loss $\mathcal{L}_{\text{task}}$, defined as the standard cross-entropy loss between the predicted logits and the ground truth labels, reflecting the model's performance on the downstream task: $\mathcal{L}_{\text{task}} = \text{CrossEntropy}(\text{Softmax}(\Phi_{l,p}(h)), y)$.

To enable the steering function to perform meaningful and disentangled attribute control, we introduce a subspace regularization term. Specifically, to encourage adaptive selection of relevant subspaces, we define a binary prior mask $m_{\text{prior}} \in \{0,1\}^r$, where entries corresponding to the shared subspace $B_{\text{shared}}$ and the attribute-specific subspace $B_i$ are set to 1, and all others to 0. The regularization loss is defined as $\mathcal{L}_{\text{reg}} = \|m(h) - m_{\text{prior}}\|_2^2$, which encourages the model to steer primarily within subspaces that are relevant to the target attribute, while suppressing activation in unrelated dimensions.

We further encourage the learned representation $R$ to align with the structured basis $S_{\text{align}}$, as defined in equation 2. The alignment loss is defined as $\mathcal{L}_{\text{align}} = 1 - \frac{\langle R, S_{\text{align}} \rangle_F}{\|R\|_F \|S_{\text{align}}\|_F}$, where $\langle A, B \rangle_F = \text{Tr}(A^\top B) = \sum_{i,j} A_{ij} B_{ij}$ denotes the Frobenius inner product, and $\| \cdot \|_F$ represents the Frobenius norm. This formulation encourages $R$ to lie in the subspace spanned by both shared and attribute-specific directions, promoting more controllable and semantically meaningful representations during training.

The overall optimization objective is as follows:

$$\mathcal{L} = \mathcal{L}_{\text{task}} + \lambda_1 \mathcal{L}_{\text{reg}} + \lambda_2 \mathcal{L}_{\text{align}}, \tag{4}$$

where $\lambda_1, \lambda_2 > 0$ are hyperparameters balancing the terms. This optimization guarantees both attribute-wise subspace alignment and inter-attribute separation, ultimately yielding an effective steering space capable of precise and disentangled multi-attribute control. By integrating different subspaces with a weighting network, we enable adaptive subspace combination, alleviating trade-offs and optimizing performance across diverse attributes' steering. See Appendix B for the detailed algorithm.

### 4.4 DYNAMIC INTERVENTION POSITION SELECTION

Previous steering approaches typically apply interventions at the same token positions (often at the suffix) for different attributes, leading to suboptimal positioning for attribute steering, which hinders optimal results Wang et al. (2024). To overcome this limitation, we propose a dynamic selection method that identifies the most relevant token position $p_i$ for each attribute $i$ by projecting token representations from the last few tokens onto attribute-specific subspaces in $\mathbb{R}^d$. This enhances steering effectiveness for attribute $i$ by targeting interventions at the most influential tokens.

Consider an input sequence with token representations $h_1, h_2, \ldots, h_T$, where $T$ denotes the sequence length. For each attribute $i$, we project the token representations onto its corresponding subspace $R_i$. The projection of a token representation $h_t$ onto this subspace is computed as $\text{proj}_{R_i}(h_t) = R_i^\top R_i h_t$. We then define the relevance score $s_{i,t}$ of token $t$ with respect to attribute $i$ as the $L_2$-norm of this projection: $s_{i,t} = \|\text{proj}_{R_i}(h_t)\|_2$. The intervention position $p_i$ for attribute $i$ is dynamically selected as $p_i = \arg\max_{t \in \{1, \ldots, T\}} s_{i,t}$, ensuring that the token with the strongest alignment to the attribute-specific subspace is chosen. The steering function $\Phi_{l,p}(h; R, W, b, m)$ for attribute $i$ is applied at position $p_i$, allowing different attributes to be steered at different tokens, reducing inter-attribute interference and enhancing control precision.

Table 1: Evaluation results on TruthfulQA, BBQ, Alpaca, Refusal, and HelpSteer, with the best result in bold and the second-best underlined. The reported values are mean performances.

| Method | TruthfulQA | | | | BBQ | Alpaca | Refusal | HelpSteer | | |
|---|---|---|---|---|---|---|---|---|---|---|
| | MC1 (↑) | MC2 (↑) | Bleu (↑) | Bleurt (↑) | Acc (↑) | Win (↑) | Sorry (↑) | Help. (↑) | Coh. (↑) | Ver. (↓) |
| Llama3-8b-inst. | $27.47_{\pm0.21}$ | $45.63_{\pm0.28}$ | $46.89_{\pm0.25}$ | $57.88_{\pm0.33}$ | $0.608_{\pm0.011}$ | $0.12_{\pm0.010}$ | $0.491_{\pm0.021}$ | $3.76_{\pm0.03}$ | $3.41_{\pm0.04}$ | $2.33_{\pm0.03}$ |
| ICL | $28.37_{\pm0.25}$ | $46.21_{\pm0.26}$ | $48.35_{\pm0.23}$ | $60.44_{\pm0.31}$ | $0.619_{\pm0.013}$ | $0.27_{\pm0.011}$ | $\underline{0.521}_{\pm0.024}$ | $3.82_{\pm0.02}$ | $3.64_{\pm0.03}$ | $2.41_{\pm0.04}$ |
| CAA | $28.41_{\pm0.18}$ | $47.55_{\pm0.23}$ | $45.42_{\pm0.21}$ | $61.54_{\pm0.28}$ | $0.629_{\pm0.009}$ | $0.29_{\pm0.008}$ | $0.493_{\pm0.018}$ | $3.77_{\pm0.02}$ | $\underline{3.89}_{\pm0.02}$ | $2.51_{\pm0.03}$ |
| ITI | $\mathbf{36.50}_{\pm0.25}$ | $\underline{54.29}_{\pm0.24}$ | $43.22_{\pm0.24}$ | $\underline{66.30}_{\pm0.30}$ | $0.612_{\pm0.012}$ | $0.23_{\pm0.009}$ | $0.280_{\pm0.015}$ | $3.82_{\pm0.02}$ | $3.21_{\pm0.04}$ | $2.06_{\pm0.03}$ |
| ReFT | $29.58_{\pm0.16}$ | $49.51_{\pm0.22}$ | $\underline{52.08}_{\pm0.18}$ | $64.06_{\pm0.29}$ | $0.637_{\pm0.008}$ | $\underline{0.30}_{\pm0.008}$ | $0.451_{\pm0.020}$ | $3.78_{\pm0.02}$ | $3.83_{\pm0.02}$ | $2.38_{\pm0.02}$ |
| MTL-LoRA | $34.82_{\pm0.17}$ | $47.23_{\pm0.24}$ | $51.78_{\pm0.19}$ | $63.26_{\pm0.28}$ | $\underline{0.641}_{\pm0.008}$ | $0.21_{\pm0.007}$ | $0.524_{\pm0.022}$ | $3.48_{\pm0.04}$ | $3.73_{\pm0.03}$ | $\mathbf{1.90}_{\pm0.02}$ |
| MAT-STEER | $29.29_{\pm0.18}$ | $49.67_{\pm0.25}$ | $43.81_{\pm0.22}$ | $55.31_{\pm0.34}$ | $0.622_{\pm0.010}$ | $0.14_{\pm0.009}$ | $0.420_{\pm0.019}$ | $\underline{3.84}_{\pm0.02}$ | $3.63_{\pm0.03}$ | $2.29_{\pm0.03}$ |
| **MSRS** | $\underline{34.91}_{\pm0.12}$ | $\mathbf{56.32}_{\pm0.20}$ | $52.32_{\pm0.14}$ | $\mathbf{66.75}_{\pm0.26}$ | $\mathbf{0.645}_{\pm0.010}$ | $\mathbf{0.36}_{\pm0.008}$ | $\mathbf{0.529}_{\pm0.02}$ | $\mathbf{3.89}_{\pm0.01}$ | $\mathbf{3.96}_{\pm0.02}$ | $\underline{2.04}_{\pm0.02}$ |
| Qwen2-7b-inst. | $26.38_{\pm0.20}$ | $45.41_{\pm0.29}$ | $49.63_{\pm0.24}$ | $65.28_{\pm0.32}$ | $0.634_{\pm0.009}$ | $0.12_{\pm0.009}$ | $0.384_{\pm0.018}$ | $3.51_{\pm0.03}$ | $3.80_{\pm0.02}$ | $2.28_{\pm0.03}$ |
| ICL | $26.84_{\pm0.24}$ | $48.33_{\pm0.27}$ | $49.79_{\pm0.22}$ | $66.63_{\pm0.30}$ | $0.633_{\pm0.010}$ | $0.16_{\pm0.010}$ | $0.413_{\pm0.020}$ | $3.65_{\pm0.03}$ | $\mathbf{3.88}_{\pm0.02}$ | $2.40_{\pm0.03}$ |
| CAA | $28.44_{\pm0.17}$ | $47.25_{\pm0.25}$ | $48.35_{\pm0.20}$ | $58.97_{\pm0.35}$ | $0.635_{\pm0.008}$ | $0.26_{\pm0.008}$ | $0.404_{\pm0.019}$ | $\underline{3.73}_{\pm0.02}$ | $\underline{3.87}_{\pm0.02}$ | $\underline{2.20}_{\pm0.02}$ |
| ReFT | $\underline{29.83}_{\pm0.15}$ | $48.69_{\pm0.24}$ | $\underline{52.57}_{\pm0.17}$ | $71.15_{\pm0.26}$ | $0.636_{\pm0.008}$ | $\underline{0.43}_{\pm0.007}$ | $0.421_{\pm0.017}$ | $3.63_{\pm0.03}$ | $3.78_{\pm0.02}$ | $2.38_{\pm0.03}$ |
| MAT-STEER | $23.08_{\pm0.19}$ | $\underline{49.51}_{\pm0.28}$ | $51.72_{\pm0.19}$ | $\underline{72.53}_{\pm0.25}$ | $0.641_{\pm0.007}$ | $0.18_{\pm0.009}$ | $\underline{0.429}_{\pm0.016}$ | $3.70_{\pm0.02}$ | $3.77_{\pm0.02}$ | $2.25_{\pm0.02}$ |
| **MSRS** | $\mathbf{34.72}_{\pm0.11}$ | $\mathbf{53.27}_{\pm0.21}$ | $\mathbf{53.10}_{\pm0.12}$ | $\mathbf{74.90}_{\pm0.21}$ | $\mathbf{0.642}_{\pm0.006}$ | $\mathbf{0.45}_{\pm0.007}$ | $\mathbf{0.445}_{\pm0.011}$ | $3.76_{\pm0.01}$ | $3.82_{\pm0.01}$ | $\mathbf{2.17}_{\pm0.02}$ |
| Mistral-7b-v0.3 | $18.83_{\pm0.28}$ | $36.54_{\pm0.33}$ | $41.56_{\pm0.29}$ | $54.52_{\pm0.38}$ | $0.614_{\pm0.014}$ | $0.14_{\pm0.012}$ | $0.631_{\pm0.028}$ | $3.75_{\pm0.03}$ | $3.92_{\pm0.02}$ | $2.36_{\pm0.04}$ |
| ICL | $21.92_{\pm0.26}$ | $49.23_{\pm0.30}$ | $44.11_{\pm0.27}$ | $57.64_{\pm0.35}$ | $0.622_{\pm0.013}$ | $0.18_{\pm0.011}$ | $0.642_{\pm0.025}$ | $3.77_{\pm0.03}$ | $\mathbf{3.94}_{\pm0.01}$ | $2.39_{\pm0.03}$ |
| CAA | $28.77_{\pm0.19}$ | $\underline{52.05}_{\pm0.26}$ | $\mathbf{53.85}_{\pm0.20}$ | $62.27_{\pm0.32}$ | $\mathbf{0.646}_{\pm0.009}$ | $0.22_{\pm0.009}$ | $0.663_{\pm0.021}$ | $3.76_{\pm0.02}$ | $3.83_{\pm0.02}$ | $\underline{2.27}_{\pm0.02}$ |
| ReFT | $\underline{30.07}_{\pm0.17}$ | $49.69_{\pm0.28}$ | $49.39_{\pm0.23}$ | $66.01_{\pm0.31}$ | $0.614_{\pm0.011}$ | $\underline{0.32}_{\pm0.008}$ | $\underline{0.668}_{\pm0.019}$ | $\underline{3.80}_{\pm0.02}$ | $3.85_{\pm0.02}$ | $2.33_{\pm0.03}$ |
| MAT-STEER | $25.45_{\pm0.22}$ | $48.38_{\pm0.31}$ | $49.46_{\pm0.24}$ | $62.62_{\pm0.34}$ | $0.631_{\pm0.010}$ | $0.19_{\pm0.010}$ | $0.644_{\pm0.023}$ | $3.78_{\pm0.02}$ | $3.86_{\pm0.02}$ | $2.29_{\pm0.02}$ |
| **MSRS** | $\mathbf{31.32}_{\pm0.11}$ | $\mathbf{52.62}_{\pm0.13}$ | $50.61_{\pm0.08}$ | $\mathbf{71.39}_{\pm0.17}$ | $\underline{0.644}_{\pm0.007}$ | $\mathbf{0.38}_{\pm0.006}$ | $\mathbf{0.693}_{\pm0.013}$ | $3.82_{\pm0.02}$ | $\underline{3.93}_{\pm0.01}$ | $\mathbf{2.21}_{\pm0.02}$ |

## 5 EXPERIMENTS

### 5.1 EXPERIMENTAL SETTINGS

**Datasets and Metrics.** To evaluate the effectiveness of our proposed method, we conduct experiments on three pairs of datasets, each designed to assess trade-offs between specific attributes in language model steering. More details are provided in Appendix B. **1) TruthfulQA & BBQ**: To evaluate *truthfulness* and *bias*, we use TruthfulQA with MC1, MC2, BLEU, and BLEURT scores (Lin et al., 2022), and BBQ with accuracy as the metric (Parrish et al., 2022). **2) Alpaca & Refusal**: We evaluate *instruction following* via win rate on Alpaca (Taori et al., 2023; Li et al., 2023b) (vs. test-davinci-003), and *refusal* via Sorry-Bench scores judged by Mistral-7B-Instruct-v0.2 (Xie et al., 2025), which assesses the rejection of malicious instructions. **3) HelpSteer**: We assess *helpfulness*, *coherence*, and *verbosity* by leveraging GPT-3.5-Turbo, following the setting of (Nguyen et al., 2025), to rate model outputs on a 0–4 scale (Wang et al., 2023), which evaluates the quality of the generated content.

Additionally, we test the **utility** on several standard benchmarks, including Hellaswag (Zellers et al., 2019), RACE (Lai et al., 2017), MMLU (Hendrycks et al., 2020), OpenBookQA (Mihaylov et al., 2018), and GLUE (Wang et al., 2018), all using accuracy as the evaluation metric. We aim to assess whether MSRS preserves the model's general abilities after fine-tuning, ensuring that task-specific steering does not degrade overall performance.

**Models and Baselines.** We evaluate MSRS on 4 models: Llama2-7B (Touvron et al., 2023), Llama3-8B-Instruct (Grattafiori et al., 2024), Qwen2-7B-Instruct (Team, 2024) , Mistral-7B-v0.3 (Jiang et al., 2023). And we compare MSRS with 6 baselines, grouped into 3 categories: **1) In-context Learning** (Brown et al., 2020): Utilizes prompts to steer attributes without altering model parameters. **2) Fine-tuning Methods**: **MTL-LoRA** (Yang et al., 2025), which employs low-rank adaptation for multi-task learning, enabling efficient attribute-specific fine-tuning. **ReFT** (Wu et al., 2024b), which adjusts model representations by fine-tuning the representation to align with target attributes. **3) Steering Methods**: **ITI** (Li et al., 2023a) applies inference-time interventions to modify activations and guide model outputs. **CAA** (Rimsky et al., 2024) steers behavior by injecting contrastive activation vectors derived from positive and negative examples. **MAT-STEER** (Nguyen et al., 2025) implements multi-attribute steering with orthogonal constraints to minimize interference between attributes.

**Experimental Setup.** All experiments were conducted on NVIDIA V100 GPUs using the Adam optimizer with a learning rate of $5 \times 10^{-3}$ and dual regularization coefficients $\lambda_1 = 0.3$, $\lambda_2 = 0.5$. Steering was applied to the $15^{\text{th}}$ transformer layer, selected as the most effective on the validation set. For each configuration, we report the average and standard deviation over 3 runs with random seeds $\{42, 43, 44\}$. In Appendix G, we provide a description of the hyperparameter selection method and sensitivity analysis to assess the robustness of MSRS to these hyperparameters, along with the computational cost for different configurations.

## 5.2 MAIN RESULTS

**MSRS excels in multiple-choice tasks.** We first evaluate MSRS on TruthfulQA and BBQ to target the trade-off between truthfulness and bias. As shown in Table 1, baseline methods often fail to optimize both attributes simultaneously. For example, ITI improves truthfulness (MC1 of 36.50 on Llama3-8B-Instruct) but sacrifices bias mitigation (BBQ Acc of 0.612), while CAA boosts bias (BBQ Acc of 0.646 on Mistral-7B) at the cost of truthfulness (MC1 of 28.77). ICL shows moderate improvements across metrics but lacks standout performance, and MAT-STEER enhances MC and Acc, but its BLEU and BLEURT scores drop. Unlike these methods, MSRS consistently balances both attributes, achieving superior performance across both attributes on multiple models. These results demonstrate MSRS's ability to jointly optimize conflicting objectives across all models.

**MSRS demonstrates strong performance in open-ended generation tasks.** We assess MSRS on Alpaca, Refusal, and HelpSteer datasets to evaluate trade-offs between instruction following, refusal, and output quality attributes. As shown in Table 1, MSRS excels in balancing instruction-following and refusal. On Alpaca, it achieves a win rate of 0.36 against test-davinci-003 on Llama3-8B-Instruct, outperforming ReFT (0.30), and on Refusal, it scores 0.529 on Sorry-Bench, surpassing CAA's 0.493. Baselines like ITI struggle with refusal (0.280), sacrificing this attribute for others. For HelpSteer, MSRS consistently improves Helpfulness, Coherence, and Verbosity on Llama3-8B-Instruct, while methods like MAT-STEER improve Helpfulness but reduce Coherence and Verbosity. MTL-LoRA achieves the best Verbosity, but at the cost of Helpfulness. MSRS outperforms existing baselines in harmonizing instruction-following with refusal and delivers high-quality generations across all HelpSteer dimensions, demonstrating its ability to integrate diverse attributes and enhance open-ended generation for specific attributes. Even when MSRS is not the absolute best (e.g., Coherence on Mistral-7b-v0.3), it achieves near-best results with a negligible gap and retains overall superiority across all attributes.

**MSRS maintains strong general capabilities on standard NLP benchmarks.** To verify that MSRS does not compromise the model's overall natural language processing abilities, we evaluate its performance on several widely used benchmarks, as shown in Table 2. More results on the MMLU (Table 6) and GLUE benchmark tasks (Table 8) are shown in Appendix B.1. These tasks assess a model's general reasoning, knowledge, and language understanding, offering a comprehensive measure of its robustness beyond attribute-specific steering. From the results, baseline methods often compromise general capabilities by overfitting to specific attributes. For example, ITI performs well on truthfulness but struggles on general benchmarks like HellaSwag and MMLU on

Table 2: General capabilities on several benchmarks.

| Method | HellaSwag | RACE | MMLU | OpenbookQA | GLUE |
|---|---|---|---|---|---|
| Llama3-8b-inst. | 0.801 | 0.671 | 0.655 | 0.556 | 0.726 |
| ReFT | 0.821 | 0.677 | 0.651 | 0.559 | 0.757 |
| ITI | 0.746 | 0.589 | 0.546 | 0.507 | 0.742 |
| MTL | 0.782 | 0.661 | 0.567 | 0.562 | 0.697 |
| CAA | 0.833 | 0.671 | 0.648 | 0.557 | 0.738 |
| MSRS | 0.839 | 0.683 | 0.657 | 0.568 | 0.775 |
| Qwen2-7b-inst. | 0.831 | 0.625 | 0.695 | 0.606 | 0.825 |
| ReFT | 0.822 | 0.644 | 0.698 | 0.613 | 0.770 |
| MTL | 0.782 | 0.641 | 0.687 | 0.562 | 0.697 |
| CAA | 0.837 | 0.633 | 0.698 | 0.609 | 0.830 |
| MSRS | 0.835 | 0.648 | 0.702 | 0.616 | 0.832 |
| Mistral-7b-v0.3 | 0.862 | 0.678 | 0.618 | 0.602 | 0.681 |
| ReFT | 0.872 | 0.679 | 0.603 | 0.608 | 0.655 |
| MTL | 0.869 | 0.644 | 0.530 | 0.574 | 0.682 |
| CAA | 0.869 | 0.667 | 0.619 | 0.611 | 0.693 |
| MSRS | 0.874 | 0.681 | 0.613 | 0.622 | 0.707 |

LLaMA3, showing weakened commonsense and knowledge reasoning. In contrast, MSRS consistently matches or exceeds baseline performance across these benchmarks. For GLUE, MSRS scores an average of 0.775, surpassing ITI and ReFT. Figure 2 compares different tasks in GLUE, where MSRS shows substantial improvements on SST-2 and STS-B, reflecting better sentiment classification and semantic similarity. This robust performance is driven by MSRS's shared subspace mechanism, which captures common steering directions across tasks and attributes, enhancing generalization and excelling in both general NLP benchmarks and targeted steering objectives.

## 5.3 ABLATION STUDY

### 5.3.1 CORE MULTI-SUBSPACE DESIGN

**The shared subspace facilitates the integration of multi-attribute features, resulting in enhanced overall performance.** Since the effectiveness of MSRS depends on the shared subspace, we investigate the impact of the rank ($k$ of the shared subspace) on performance. As shown in Figure 3, we report the relationship between the performance (Avg. Accuracy) of the Truthfulness and Bias attributes and the proportion of the shared subspace within the total space (shared rank ratio).

Models with a shared subspace in $R$ outperform those without shared subspace (w/o Shared Subspace, only attribute-specific space) or with no subspace partition. The optimal shared subspace ratio varies with $R$: for $R = 8$, it is around 30%, and for $R = 12$, it is about 20%. As the shared subspace ratio increases, the attribute-specific space decreases, causing performance to decline. At

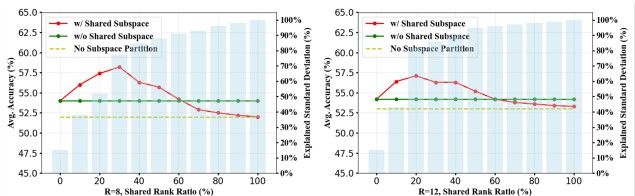

Figure 2: Comparison of model performance across GLUE.

Figure 3: Relationship between performance and shared rank ratio, alongside the explained standard deviation.

100%, the model degenerates to the no subspace partition case. Additionally, we display the explained standard deviation (blue bars, i.e., the ratio of preserved singular values $\sigma$ to the total sum of singular values $\Sigma$). The optimal shared subspace ratio is between 20% and 40%, where the explained standard deviation reaches approximately 60%. Accordingly, we choose the fewest singular vectors that capture at least 60% of the total energy in $\Sigma_c$. See sensitivity analysis in Appendix H.

Table 3: Comparison of steering subspace training strategies across datasets.

| Method | TruthfulQA | | BBQ | Alpaca | Refusal | HelpSteer | | |
|---|---|---|---|---|---|---|---|---|
| | MC1 | MC2 | Acc | Win | Sorry | Help. | Coh. | Ver. |
| Llama3-8b-inst. | | | | | | | | |
| Same Space | 29.58 | 49.51 | 0.637 | 0.30 | 0.451 | 3.87 | 3.89 | 2.38 |
| MSRS$_{Attribute}$ | 32.52 | 52.55 | 0.627 | **0.36** | **0.529** | 3.88 | **3.96** | **2.24** |
| MSRS$_{Rank}$ | **33.50** | **52.74** | **0.646** | 0.35 | 0.527 | **3.89** | 3.95 | 2.28 |
| Qwen2-7b-inst. | | | | | | | | |
| Same Space | 29.83 | 48.69 | 0.637 | 0.434 | 0.422 | 3.63 | 3.78 | 2.38 |
| MSRS$_{Attribute}$ | **34.72** | **53.27** | **0.642** | **0.451** | **0.446** | 3.64 | 3.81 | 2.20 |
| MSRS$_{Rank}$ | 26.41 | 47.65 | 0.635 | 0.442 | 0.439 | **3.76** | **3.82** | 2.17 |
| Mistral-7b-v0.3 | | | | | | | | |
| Same Space | 30.32 | 49.69 | 0.615 | 0.33 | 0.669 | 3.82 | 3.85 | 2.33 |
| MSRS$_{Attribute}$ | **30.07** | **52.62** | **0.644** | 0.38 | **0.693** | **3.82** | **3.93** | 2.27 |
| MSRS$_{Rank}$ | 28.36 | 49.94 | 0.631 | **0.39** | 0.673 | 3.76 | 3.87 | **2.26** |

Table 4: Comparison of Last token vs. Important token intervention.

| Method | TruthfulQA | | BBQ | Alpaca | Refusal | HelpSteer | | |
|---|---|---|---|---|---|---|---|---|
| | MC1 | MC2 | Acc | Win | Sorry | Help. | Coh. | Ver. |
| LLaMA2-7B | | | | | | | | |
| Last Token | 26.41 | 42.88 | 0.631 | 0.12 | 0.579 | 2.70 | 2.68 | 2.73 |
| Important Token | **29.10** | **48.60** | **0.644** | **0.13** | **0.583** | **3.12** | **3.06** | 2.47 |
| LLaMA3-8B-Inst. | | | | | | | | |
| Last Token | 33.50 | 52.74 | 0.648 | **0.36** | **0.529** | **3.88** | **3.96** | 2.24 |
| Important Token | **33.71** | **56.32** | **0.655** | 0.32 | 0.511 | 3.85 | 3.95 | **1.99** |
| Qwen2-7B-Inst. | | | | | | | | |
| Last Token | 34.72 | 53.27 | 0.6421 | **0.45** | 0.446 | **3.70** | 3.82 | **2.17** |
| Important Token | **36.12** | **55.63** | **0.6572** | 0.42 | **0.448** | 3.69 | **3.83** | 2.06 |

**Evaluating the effectiveness of adaptive subspace selecting mechanism**. We conduct ablation studies comparing three strategies for training steering subspaces: (1) Same Space, where all attributes are trained in a single subspace without isolation; (2) MSRS$_{Attribute}$, the basis matrix $R \in \mathbb{R}^{r \times d}$ is partitioned into $n + 1$ blocks: $R = [B_{shared} \| B_1 \| \dots \| B_n]$, where $B_{shared}$ is a shared subspace and each $B_i$ corresponds to a specific attribute. The mask network $m(h)$ generates soft weights that are applied to each block individually, allowing the model to adaptively activate relevant attribute-specific subspaces; and (3) MSRS$_{Rank}$ treats the basis matrix $R \in \mathbb{R}^{r \times d}$ as a flat set of $r$ independent basis vectors, without any explicit block structure. The mask network $m(h) \in \mathbb{R}^r$ assigns a soft weight to each row of $R$, enabling control at the level of individual basis directions. In our implementation, we adopt the MSRS$_{Attribute}$ configuration as the default setup for MSRS. This choice offers a good balance between interpretability and control, allowing the model to modulate behavior based on attribute-specific subspaces. Results are reported in Table 3 across multiple datasets and models.

Training all attributes in the same space leads to suboptimal outcomes due to interference between conflicting objectives. For example, Same Space performs worse than both MSRS variants across nearly all metrics. MSRS mitigates this by decoupling subspaces and adaptively selecting them via $m(h)$. The MSRS$_{Attribute}$ configuration, which groups low-rank dimensions into attribute-aligned subspaces, strikes a balance between parameter sharing and specialization, consistently improving both truthfulness and bias mitigation across models. The MSRS$_{Rank}$ variant offers finer-grained control, improving performance on LLaMA3-8B-Instruct and suggesting better truthfulness. However, it lags behind MSRS$_{Attribute}$ on Qwen2-7B-Instruct, possibly due to differing optimal subspace granularity across models, with some benefiting from coarser grouping. Additional results are in Appendix D (Table 11).

### 5.3.2 Enhanced Steering Mechanisms

**Validating the effectiveness of proposed dynamic intervention position selection mechanism**. We compare two settings: (1) the *Last token*, where intervention is applied to the last token in the sequence, and (2) *Important token*, which applies intervention at the dynamically selected *Important token* from the last few tokens. As shown in Table 4, dynamic token selection consistently outperforms fixed-position steering across all models. For example, on LLaMA2-7B, MC1 improves from 26.41 to 29.10 and BBQ accuracy from 0.631 to 0.644. Similar gains are observed on LLaMA3-8B-Instruct and Qwen2-7B-Instruct. These results validate that selecting the most semantically relevant

token for intervention mitigates inter-attribute interference and enables more effective attribute control. Additional results are shown in Appendix E (Table 12).

**Explore the relationship between the $R$-similarity of selected steering token and model performance.** We designed an experiment where the intervention was applied to different single token at the suffix position, as illustrated in Figure 4. The results indicate that steering at the token with higher similarity to $R$ generally leads to better performance. Moreover, this intervention consistently outperforms the no-intervention baseline across all tokens, demonstrating its generalization ability across different tokens. We attribute this consistent improvement to using the dominant directions of average activations from the last few tokens to guide training, enabling the learned $R$ to generalize to some extent to different tokens. By dynamically selecting tokens for intervention during inference based on their similarity to $R$, different attributes can be applied to those tokens most closely related to the desired attribute steering, yielding more precise control over attributes.

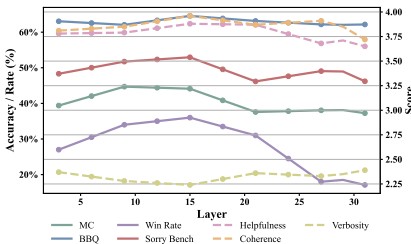

Figure 4: Token–$R$ similarity vs. performance under different token intervention. Higher similarity correlates with better performance.

**Determination of the optimal layer for steering interventions**. We evaluate model performance by injecting steering vectors at different transformer layers, applying interventions only at each candidate layer, as shown in Figure 5. Lower layers show weak steering likely due to insufficient semantic abstraction, while mid-to-upper layers generally yield better results, with layer 15 achieving the best performance. Deeper layers, however, tend to overfit, leading to performance degradation. These findings align with prior work (Skean et al., 2025), which suggests that mid-depth layers in transformer architectures play a key role in controlling downstream tasks. Detailed results for different layers are in Appendix F (Table 13). To select the optimal intervention layer for multi-attribute subspace training, we conduct a grid search over held-out validation splits to identify the layer that best balances attribute trade-offs. This strategy guides our selection of the intervention layer for all subsequent experiments, ensuring interventions are applied where most effective, thereby maximizing the utility of the learned multi-subspace representations.

Figure 5: Performance of interventions at different transformer layers. Mid-layer intervention consistently outperforms others.

**Evaluating the benefit of aligning learned steering subspaces with SVD-derived priors.** We conduct an ablation study on LLaMA2-7B using four configurations: (1) **standard ReFT**, which learns a single steering subspace without partitioning; (2) **CS-ReFT** (Zhou, 2025), which partitions the total rank $R$ into equal-sized subspaces and learns a weighted matrix to combine them; (3) **Naive SVD-based subspace concatenation**, which applies SVD to extract low-rank basis vectors for each attribute and concatenates them with the shared subspace basis to initialize the projection matrix $R$ in ReFT; (4) **Subspace alignment strategy**, which aligns the learned subspace $R$ with SVD priors $m_{prior}$ during training using a mask regularization loss $\mathcal{L}_{reg}$ and a directional alignment loss $\mathcal{L}_{align}$, enabling adaptive, interpretable, and task-aware control. The results are shown in Table 5. The subspace alignment strategy (`+ReFT+Align`) outperforms both baselines across all metrics, surpassing ReFT and SVD-only approaches. Compared to CS-ReFT, our method allocates subspaces more effectively by aligning them with SVD priors, resulting in improved performance. These results highlight the importance of incorporating structural priors during training, with SVD priors serving as an effective foundation for guiding subspace learning.

Table 5: Ablation study on multi-subspace alignment strategies.

| Method | MC1 | MC2 | BLEU | BLEURT | BBQ (Acc) |
|---|---|---|---|---|---|
| LLaMA2 | 18.58 | 35.25 | 38.37 | 52.65 | 0.604 |
| +ReFT | 21.03 | 36.93 | 44.74 | 55.99 | 0.626 |
| +CS-ReFT | 22.85 | 36.12 | 44.56 | 65.47 | 0.634 |
| +ReFT + SVD | 22.58 | 38.97 | 46.45 | 58.92 | 0.633 |
| +ReFT + Align | **24.94** | **41.41** | **51.10** | **66.50** | **0.644** |

### 5.3.3 SCALABILITY AND INTERFERENCE PATTERNS WITH GROWING ATTRIBUTE COUNT

**Investigate the relationship between the number of attributes $n$ and performance in multi-attributes scenario.** Figure 6a visualizes the relative performance change ($\Delta\%$) for each attribute $a_i$ ($i \in \{1, \ldots, 10\}$) across $n \in \{0, \ldots, 10\}$ attributes (attributes details are shown in Appendix B),

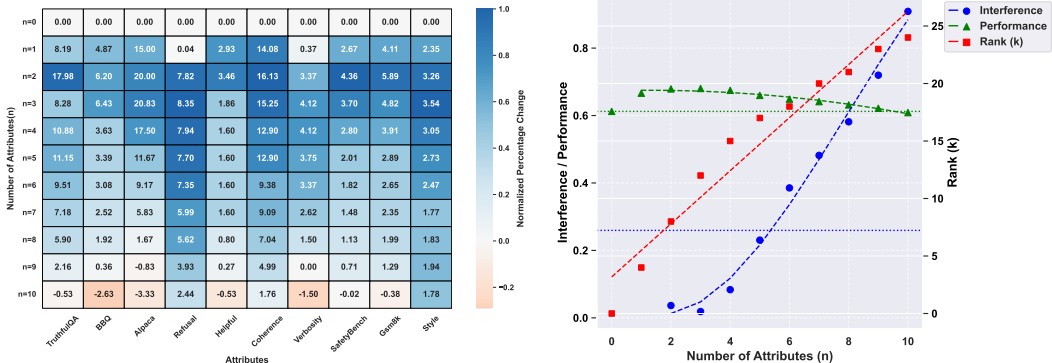

(a) Normalized Relative Percentage Change ($\Delta\%$) in Different Attributes Compared to Baseline (n = 0).

(b) Interference, Performance, and Rank(k) vs. Number of Attributes(n).

Figure 6: Impact of Attribute Scaling on Performance, Interference, Rank in MSRS.

relative to the baseline ($n = 0$). For each attribute $a_i$ and subset size $n = j$, we construct $M$ random subsets $S_{i,j}^m$ ($m \in \{1, \ldots, M\}$), each defined as $S_{i,j}^m = \{a_i\} \cup T_{i,j}^m$, where $T_{i,j}^m \subset \mathcal{A} \setminus \{a_i\}$ is a random subset of $j - 1$ attributes sampled from the 9 attributes excluding $a_i$, with $|\mathcal{A}| = 10$; in our setup, $M = 5$. The performance of $a_i$ in $j$-joint attributes is measured after joint training, and averaged as $\text{Perf}_{a_i}(n = j) = \frac{1}{M} \sum_{m=1}^{M} \text{Perf}_{a_i}(S_{i,j}^m)$. The baseline is $\text{Perf}_{a_i}(n = 0)$, and the relative performance change is given by $\Delta_{a_i}(n = j) = \frac{\text{Perf}_{a_i}(n=j) - \text{Perf}_{a_i}(n=0)}{\text{Perf}_{a_i}(n=0)}$. The $\Delta_{a_i}(n)$ is normalized to [0, 1] as $\text{Norm}_{\Delta_{a_i}}(n) = \frac{\Delta_{a_i}(n)}{\max_n \Delta_{a_i}(n) - \min_n \Delta_{a_i}(n)}$. The heatmap shows $\text{Norm}_{\Delta_{a_i}}(n)$ as color intensity (red: negative, blue: positive) with $\Delta_{a_i}(n)$ annotated. (See Table 7 for the performance results.) As shown in Figure 6a, for $n = 2 - 4$, most attributes show strong improvements ($\Delta\% > 0$, deep blue), indicating effective attributes joint. At $n = 5 - 8$, gains weaken (lighter blue), reflecting uneven benefits. At $n = 9 - 10$, some attributes drop to or below baseline ($\Delta\% \leq 0$, red/neutral), signaling performance degradation. We attribute these trends to interference in the shared subspace driven by increasing attribute counts. At low $n$, attributes align with minimal interference. As $n$ grows, dominant attributes (e.g., Refusal, Style) with positive $\Delta\%$ bias the shared subspace, their steering vectors overpowering weaker attributes, which exhibit sharply reduced gains. At $n = 9 - 10$, intensified interference causes performance declines across more attributes.

**Performance, interference, and rank vary with number of attributes $n$ in multi-attribute steering.** Figure 6b illustrates the relationships between average performance (green), interference (blue), and rank $k$ (red) with $n$. Performance is computed as the mean over all attributes. Interference quantifies disorder in the shared subspace, manifested as consistent performance declines across attributes. It is defined as the mean normalized decline from peak performance: for each attribute $a_j$, the decline is given by $\text{Decline}_{a_j}(n) = \frac{\text{PeakPerf}_{a_j} - \text{Perf}_{a_j}(n)}{\text{PeakPerf}_{a_j}}$, with $\text{PeakPerf}_{a_j} = \max_n \text{Perf}_{a_j}(n)$ as the highest performance and $\text{Perf}_{a_j}(n)$ as the performance at $n$. The average interference is computed as $\text{AvgInterference}(n) = \frac{1}{m} \sum_{j=1}^{m} \text{Decline}_{a_j}(n)$, where $m = 10$ is the number of attributes. The trends approximate a quadratic fit for performance ($R^2 = 0.9520$), a linear fit for rank $k$ ($R^2 = 0.9427$), and a cubic fit for interference ($R^2 = 0.9903$) (see Appendix K for fit parameters). The green curve shows sustained improvements over baseline for $n < 9$, peaking at $n = 2$ or 3 with subsequent gradual reductions in gains. The blue curve indicates accelerating interference with $n$, while optimal performance corresponds to rank $k = 8$ to 12, consistent with Table 15. These patterns suggest that at low $n$, attributes align synergistically, enabling superior multi-attribute steering performance and demonstrating attribute scalability for $n < 9$, corresponding AvgInterference $< 0.6$.

## 6 CONCLUSION

We propose Multi-Subspace Representation Steering (MSRS), a framework for multi-attribute control in LLMs. MSRS combines attribute-specific subspaces, SVD-guided dimensionality, a shared subspace for cross-attribute correlations, and dynamic token selection to reduce steering conflicts and improve control over attributes like truthfulness, bias, instruction-following, refusal, and generation quality. Experiments demonstrate that MSRS outperforms existing methods, offering a scalable and robust solution for aligned language generation.

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

## A LLM USAGE STATEMENT

In this research, Large Language Models (LLMs) were used to assist with tasks related to language refinement, formatting, and reference management. The LLM helped in improving the clarity and flow of the text, ensuring consistent citation formatting, and aiding in the organization of the manuscript. The authors affirm that all final content was reviewed and critically assessed by the authors to ensure the accuracy and relevance of the research.

## B IMPLEMENTATION DETAILS

**A.1 Algorithm** Figure 1b illustrates the overall structure of MSRS, which integrates shared and attribute-specific subspaces for precise and disentangled multi-attribute control. To complement this, we present the detailed training algorithm in Algorithm 1.

The MSRS training process begins by computing activation statistics from each attribute-specific dataset $\mathcal{D}_i$, aggregating token representations across examples to form mean activations $\tau_i$. These activations are used to extract a shared subspace $B_{\text{shared}}$ via singular value decomposition (SVD) over concatenated attribute activations, capturing the dominant, overlapping directions.

Next, we derive private subspaces $B_i$ for each attribute $i$ by removing the shared components from their individual activations, followed by a second SVD on the residuals. The final aligned subspace $S_{\text{align}}$ is constructed by concatenating the shared and private bases.

Training proceeds by optimizing the steerable representation function $\Phi_{l,p}(h; R, W, b, m)$, applied to token representations at a selected layer $l$ and position $p$. Once the intervention is applied, the resulting representation is passed through a Softmax function, which normalizes the output to produce a probability distribution over the possible classes. The Softmax function converts the raw scores (logits) produced by the intervention into a range between 0 and 1, ensuring that the sum of the outputs is equal to 1, representing the predicted class probabilities. The predicted probabilities are then compared to the ground truth labels $y$ using Cross-Entropy loss. Cross-Entropy measures the difference between the predicted probabilities (after applying Softmax) and the true label distribution. It quantifies the performance of the model by penalizing the difference, with the objective of minimizing this loss during training. Finally, the computed cross-entropy loss is assigned as the task loss $\mathcal{L}_{\text{task}}$, which is minimized through the optimization process. This ensures that the steerable representation function $\Phi_{l,p}(h; R, W, b, m)$ is optimized to produce more accurate predictions in subsequent tasks.

The total loss function $\mathcal{L}$ combines the task loss $\mathcal{L}_{\text{task}}$ with two regularization terms. The first term regularizes the mask network $m(h)$, encouraging it to stay close to a prior mask $m_{\text{prior}}$ through the $\ell_2$-norm term $\lambda_1 \|m(h) - m_{\text{prior}}\|_2^2$. The second term enforces that the learned representation $R$ aligns with a reference matrix $S_{\text{align}}$ by minimizing the cosine dissimilarity, controlled by $\lambda_2 \mathcal{L}_{\text{align}}$, where:

$$\mathcal{L}_{\text{align}} = 1 - \frac{\langle R, S_{\text{align}} \rangle}{\|R\|_2 \|S_{\text{align}}\|_2}$$

The hyperparameters $\lambda_1$ and $\lambda_2$ balance the task loss and the regularization terms, ensuring that the model optimizes both task performance and alignment with the reference structure.

This procedure ensures that the learned steering function preserves attribute disentanglement, supports subspace coordination, and enables adaptive attribute combination. It ultimately yields a robust steering model capable of controlling multiple behavior dimensions in LLMs with minimal interference.

**A.2 Datasets and Metircs** We provide detailed descriptions of the datasets and evaluation metrics used in our experiments to assess multi-attribute steering performance.

**TruthfulQA** TruthfulQA (Lin et al., 2022) evaluates a model's ability to produce truthful and informative responses. We report:

- **MC1 (Single-true)**: Accuracy in selecting the single correct answer (highest log-probability among 4–5 candidates).

---

**Algorithm 1** Multi-Subspace Representation Steering

---

**Require:** Datasets $\mathcal{D}_i$ for attributes $i = 1, \ldots, n$, model $M$, token position $p$, layer $l$, $\Phi_{l,p}(h; R, W, b, m)$, $m_{\text{prior}} \in [0, 1]^{\text{r}}$, regulation hyperparameters $\lambda_1, \lambda_2$, label $y$
**Ensure:** Steering model $\Phi_{l,p}(h; R, W, b, m)$
  1: **Activation-wise Preparation**
  2: **for** each attribute $i$ **do**
  3:     $\tau_i = \frac{1}{|\mathcal{D}_i|} \sum_j h_{i,j}$
  4: **end for**
  5: Combine activations: $\tau_c = [\tau_1, \ldots, \tau_n]$
  6: **Shared Subspace Extraction**
  7: $\tau_c = U_c \Sigma_c V_c^\top$
  8: $B_{\text{shared}} = U_{1:r_s}^{(i)\,\top}$                                             $\triangleright$ Top $r_s$ directions
  9: **Private Subspace Extraction**
10: **for** each attribute $i$ **do**
11:     $\tau_i = U_i \Sigma_i V_i^\top$
12:     $H_{\text{res}}^{(i)} = \tau_i - R_{\text{shared}}^\top (R_{\text{shared}} \tau_i)$
13:     $H_{\text{res}}^{(i)} = U^{(i)} S^{(i)} V^{(i)\top}$
14:     $B_i = U_{1:r_i}^{(i)\,\top}$                                       $\triangleright$ Top $r_i$ directions
15: **end for**
16: **Construct Aligned Subspace**
17: $S_{\text{align}} = [B_{\text{shared}}, B_1, \ldots, B_n]$
18: **Optimize Representation Matrix**
19: Initialize $R, W, b$, subspace mask $m(h)$
20: **for** each training step **do**
21:     $\Phi_{l,p}(h) = h + R^\top \cdot \text{diag}(m(h)) \cdot (Wh + b - Rh)$
22:     $\mathcal{L}_{\text{task}} = \text{CrossEntropy}\left(\text{Softmax}\left(\Phi_{l,p}(h)\right), y\right)$
23:     $\mathcal{L}_{\text{align}} = 1 - \frac{\langle R, S_{\text{align}} \rangle}{\|R\|_2 \|S_{\text{align}}\|_2}$
24:     $\mathcal{L} = \mathcal{L}_{\text{task}} + \lambda_1 \|m(h) - m_{\text{prior}}\|_2^2 + \lambda_2 \mathcal{L}_{\text{align}}$
25:     Update parameters $R, W, b, m$
26: **end for**
27: **return** $\Phi_{l,p}(h; R, W, b, m)$

---

- **MC2 (Multi-true)**: Normalized probability assigned to all true reference answers.

- **BLEURT**, **BLEU**, and **ROUGE**: Generation-level similarity scores, computed as the difference between the maximum similarity to any true answer and any false answer.

- **GPT-judge** and **GPT-info**: GPT-based classifiers trained to predict human ratings of truthfulness and informativeness.

**BBQ**    The Bias Benchmark for QA (BBQ) (Parrish et al., 2022) measures social bias in QA outputs across nine social dimensions (e.g., race, gender). We report accuracy: whether the model selects the correct answer.

**AlpacaEval**    (Li et al., 2023b): Measures instruction-following ability via win rate against a strong baseline (test-davinci-003), judged by GPT-4o.

**Sorry-Bench**    (Xie et al., 2025): Evaluates instruction refusal on harmful inputs using a fine-tuned expert model (Mistral-7B-Instruct-v0.2). We report refusal accuracy based on the model's ability to reject malicious or unethical instructions.

**HelpSteer**    (Wang et al., 2023) is a human-aligned benchmark for evaluating model helpfulness. Each response is rated by GPT-4o across:

- **Helpfulness**: Relevance and utility of the response.

- **Coherence**: Logical consistency and fluency.

- **Verbosity**: Appropriateness of response length.

Scores range from 0 (poor) to 4 (excellent), and we report the average for each dimension.

**SafetyBench**   (Zhang et al., 2023): Measures performance on multiple safety dimensions including offensiveness, unfairness, ethics and morality, and privacy and property. This benchmark evaluates the model's ability to generate safe and responsible responses, covering a broad range of sensitive topics. The evaluation involves assessing the degree of harmfulness in responses, and how well the model adheres to ethical and privacy considerations.

**GSM8k**   (Cobbe et al., 2021): Evaluates reasoning ability by testing the model on a variety of mathematical word problems. The GSM8k benchmark is designed to assess the model's ability to perform multi-step reasoning and inference, as well as its ability to handle mathematical tasks at a high level of accuracy.

**Human-Style-Answers**   : A benchmark designed to evaluate the style of human-like responses generated by AI models. It focuses on generating natural, contextually appropriate, and coherent responses. The evaluation includes assessing the fluency, tone, and style of the model's responses, ensuring they are aligned with typical human responses. Details of the benchmark can be found at `https://huggingface.co/datasets/innova-ai/Human-Style-Answers`.

**General Benchmarks.**   To verify that steering does not impair general capabilities, we evaluate on standard NLP tasks:

- **HellaSwag** (Zellers et al., 2019): commonsense inference; metric: *accuracy*.
- **RACE** (Lai et al., 2017): reading comprehension; metric: *accuracy*.
- **OpenBookQA** (Mihaylov et al., 2018): elementary science QA; metric: *accuracy*.

**MMLU** (Hendrycks et al., 2020) (Massive Multitask Language Understanding) is a challenging benchmark designed to evaluate a model's world knowledge and problem-solving ability under zero-shot and few-shot settings. It comprises 15,908 multiple-choice questions spanning 57 diverse subjects, including STEM, humanities, social sciences, and professional disciplines such as law and ethics. The tasks vary in difficulty from elementary to advanced levels, making MMLU an ideal benchmark for identifying model weaknesses across both general and specialized domains.

Each subject contains at least 100 test questions, exceeding the length of most human exams. The dataset is split into a few-shot development set (5 questions per subject), a validation set (1,540 questions), and a test set (14,079 questions). The evaluation metric is *average accuracy* across all subjects.

**GLUE** (Wang et al., 2018) (General Language Understanding Evaluation) is a widely used benchmark for evaluating general-purpose language understanding. It consists of nine diverse NLP tasks that span a range of linguistic phenomena, including sentiment analysis, paraphrase detection, textual entailment, and question answering. These tasks collectively assess a model's ability to perform natural language understanding in varied contexts.

The included tasks are:

**MNLI** (Multi-Genre Natural Language Inference): Predict entailment, contradiction, or neutrality between premise and hypothesis across multiple domains.

**QNLI** (Question Natural Language Inference): Convert question answering into an entailment task.

**QQP** (Quora Question Pairs): Detect if two questions from Quora have the same meaning.

**SST-2** (Stanford Sentiment Treebank): Classify sentiment in movie reviews as positive or negative.

**CoLA** (Corpus of Linguistic Acceptability): Judge grammatical acceptability of a sentence.

**STS-B** (Semantic Textual Similarity Benchmark): Score sentence pairs on semantic similarity.

**MRPC** (Microsoft Research Paraphrase Corpus): Determine if two sentences are paraphrases.

**RTE** (Recognizing Textual Entailment): Binary entailment classification from multiple datasets.

**WNLI** (Winograd NLI): Resolve coreference in complex pronoun cases.

Table 6: MMLU per-task performance on different methods. The best result is highlighted in bold, and the second-best is underlined.

| Method | Math | Health | Physics | Business | Biology | Chemistry | CS | Economics | Eng. | Philosophy | Other | History | Geog. | Politics | Psych. | Culture | Law |
|---|---|---|---|---|---|---|---|---|---|---|---|---|---|---|---|---|---|
| LLaMA3-8B-Instruct | 0.430 | _0.697_ | 0.533 | 0.819 | 0.791 | 0.502 | 0.629 | 0.675 | 0.641 | 0.567 | 0.694 | 0.778 | 0.848 | 0.796 | 0.764 | 0.816 | 0.516 |
| + ITI | 0.371 | 0.597 | 0.450 | 0.703 | 0.685 | 0.472 | 0.476 | 0.566 | 0.517 | 0.486 | 0.616 | 0.624 | 0.732 | 0.667 | 0.653 | 0.738 | 0.393 |
| + MTL-LoRA | 0.365 | 0.587 | 0.477 | 0.769 | 0.687 | 0.429 | 0.534 | 0.547 | 0.503 | 0.523 | 0.547 | 0.703 | 0.737 | 0.698 | 0.690 | 0.687 | 0.432 |
| + CAA | 0.436 | 0.695 | 0.545 | 0.828 | 0.786 | 0.488 | _0.638_ | **0.675** | **0.676** | 0.566 | 0.689 | 0.772 | _0.843_ | 0.787 | 0.768 | **0.852** | 0.513 |
| + Ours | **0.567** | **0.714** | **0.642** | **0.854** | **0.846** | **0.578** | **0.648** | _0.759_ | _0.759_ | **0.617** | **0.761** | 0.809 | **0.889** | 0.810 | 0.814 | 0.850 | **0.563** |
| Qwen2-7B-Instruct | _0.567_ | 0.712 | 0.630 | **0.863** | 0.844 | 0.574 | 0.650 | 0.757 | 0.738 | 0.594 | 0.754 | 0.801 | 0.874 | 0.813 | 0.803 | 0.825 | 0.557 |
| + MTL-LoRA | 0.548 | 0.710 | 0.628 | 0.838 | 0.841 | 0.571 | _0.651_ | 0.727 | 0.745 | 0.592 | 0.739 | 0.791 | 0.874 | **0.818** | 0.799 | 0.822 | 0.554 |
| + CAA | **0.568** | _0.714_ | _0.639_ | 0.854 | _0.850_ | 0.578 | **0.653** | **0.760** | _0.759_ | _0.613_ | _0.758_ | 0.815 | _0.884_ | _0.815_ | **0.852** | 0.561 |
| + Ours | 0.567 | 0.715 | 0.642 | 0.856 | 0.846 | 0.578 | 0.648 | _0.759_ | 0.762 | 0.617 | 0.761 | 0.809 | 0.889 | 0.810 | 0.814 | _0.851_ | 0.563 |
| Mistral-7b-v0.3 | _0.387_ | 0.658 | 0.475 | 0.766 | 0.742 | 0.492 | 0.587 | 0.545 | 0.614 | 0.567 | 0.682 | 0.761 | 0.773 | **0.776** | _0.737_ | 0.771 | _0.503_ |
| + MTL-LoRA | 0.318 | 0.573 | 0.411 | 0.689 | 0.637 | 0.386 | 0.529 | 0.476 | 0.462 | 0.466 | 0.621 | 0.648 | 0.692 | 0.654 | 0.617 | 0.687 | 0.445 |
| + CAA | **0.388** | 0.659 | 0.473 | 0.765 | _0.742_ | _0.482_ | 0.595 | 0.585 | _0.600_ | 0.570 | 0.685 | _0.760_ | _0.771_ | 0.768 | 0.729 | _0.774_ | 0.497 |
| + Ours | 0.365 | **0.687** | 0.477 | 0.769 | 0.687 | 0.429 | 0.534 | _0.547_ | 0.503 | _0.523_ | 0.547 | 0.703 | 0.737 | _0.698_ | **0.790** | **0.787** | **0.532** |

Table 7: Model Performance for Different Numbers of Attribute Settings.

| r (n) | truthfulqa (↑) | bbq (↑) | alpaca (↑) | refusal (↑) | helpful (↑) | coherence (↑) | verbosity (↓) | SafetyBench (↑) | gsm8k (↑) | style (↑) |
|---|---|---|---|---|---|---|---|---|---|---|
| 0 | 36.55% | 0.6081 | 0.12 | 0.4911 | 3.76 | 3.41 | 2.33 | 62.13 | 70.32 | 75.35 |
| 4 (n=1) | 43.54% | 0.6476 | 0.35 | 0.5313 | 3.89 | 3.91 | 2.21 | 64.79 | 75.21 | 77.82 |
| 8 (n=2) | 43.12% | 0.6457 | 0.36 | 0.5295 | 3.89 | 3.96 | 2.24 | 64.84 | 74.46 | 77.81 |
| 12 (n=3) | 39.57% | 0.6471 | 0.37 | 0.5321 | 3.83 | 3.93 | 2.22 | 64.43 | 73.71 | 78.02 |
| 15 (n=4) | 40.52% | 0.6301 | 0.33 | 0.5301 | 3.82 | 3.85 | 2.22 | 63.87 | 73.07 | 77.65 |
| 17 (n=5) | 38.74% | 0.6246 | 0.26 | 0.5179 | 3.82 | 3.85 | 2.23 | 63.38 | 72.35 | 77.41 |
| 18 (n=6) | 38.52% | 0.6237 | 0.23 | 0.5172 | 3.82 | 3.73 | 2.29 | 66.26 | 72.18 | 77.21 |
| 20 (n=7) | 38.47% | 0.6233 | 0.14 | 0.5105 | 3.82 | 3.72 | 2.32 | 62.51 | 71.97 | 76.68 |
| 21 (n=8) | 37.87% | 0.6197 | 0.12 | 0.5087 | 3.79 | 3.60 | 2.35 | 62.30 | 71.72 | 76.73 |
| 23 (n=9) | 37.68% | 0.6102 | 0.11 | 0.5091 | 3.79 | 3.58 | 2.36 | 62.28 | 71.73 | 76.81 |
| 24 (n=10) | 36.88% | 0.5920 | 0.08 | 0.5031 | 3.79 | 3.47 | 2.37 | 62.12 | 70.05 | 76.69 |

## B.1 PERFORMANCE ON MMLU BENCHMARK

This section provides a comprehensive evaluation of MSRS on the Massive Multitask Language Understanding (MMLU) benchmark, assessing its ability to maintain and enhance general language capabilities across diverse domains. The MMLU benchmark, comprising 57 tasks across 17 subjects, serves as a rigorous testbed for evaluating model robustness beyond attribute-specific steering. We compare MSRS against baseline ITI, MTL-LoRA, and CAA on three base models: LLaMA3-8B-Instruct, Qwen2-7B-Instruct, and Mistral-7b-v0.3. Table 6 summarizes the per-subject performance of MSRS and baselines on the MMLU benchmark. The results reveal distinct patterns in how each method impacts general capabilities, with MSRS demonstrating superior consistency and enhancement over baselines.

**MSRS enhances or preserves performance across subjects.** On LLaMA3-8B-Instruct, MSRS achieves the highest accuracy in 14 out of 17 subjects, with significant gains in STEM fields such as mathematics (0.567 vs. 0.430 for the base model), physics (0.642 vs. 0.533), and chemistry (0.578 vs. 0.502). These improvements underscore MSRS's ability to bolster reasoning and knowledge retention in technically demanding domains. In humanities and social sciences, such as history (0.809 vs. 0.778) and psychology (0.814 vs. 0.764), MSRS also outperforms the base model, indicating broad applicability. For Qwen2-7B-Instruct, MSRS matches or exceeds the base model's performance in most subjects, with notable improvements in geography (0.889 vs. 0.874) and psychology (0.814 vs. 0.803). These gains, though modest, highlight MSRS's stability across high-performing base models, preserving their strong initial capabilities while enabling targeted enhancements. On Mistral-7b-v0.3, MSRS delivers substantial uplifts in subjects like psychology (0.790 vs. 0.737) and law (0.532 vs. 0.503), despite the base model's lower baseline performance. However, in mathematics (0.365 vs. 0.387), MSRS slightly underperforms, suggesting potential limitations in enhancing weaker base models in certain domains.

**Baseline methods reveal trade-offs.** ITI consistently degrades performance across most subjects due to its focus on truthfulness. On LLaMA3-8B-Instruct, ITI drops accuracy in mathematics to 0.371 (vs. 0.430) and computer science to 0.476 (vs. 0.629), reflecting a significant loss in general reasoning and knowledge. Similar declines are observed across all models, confirming ITI's unsuitability for preserving broad capabilities. MTL-LoRA also exhibits reduced performance, particularly in STEM subjects. For Mistral-7b-v0.3, accuracy in mathematics falls to 0.318 (vs. 0.387) and

chemistry to 0.386 (vs. 0.492). On LLaMA3-8B-Instruct, declines are evident in economics (0.547 vs. 0.675) and engineering (0.503 vs. 0.641). These results suggest that MTL-LoRA's multi-task fine-tuning overfits to specific tasks, compromising the model's general knowledge base. CAA performs closer to the base models but rarely surpasses them. On Qwen2-7B-Instruct, CAA achieves comparable scores (e.g., 0.567 in mathematics, 0.854 in business) but trails MSRS in geography (0.884 vs. 0.889) and psychology (0.813 vs. 0.814). On Mistral-7b-v0.3, CAA maintains baseline levels (e.g., 0.774 in culture) without the consistent improvements seen in MSRS, indicating limited generalization beyond attribute steering.

The experimental results affirm that MSRS excels in maintaining and often enhancing general NLP capabilities across the MMLU benchmark, outperforming baseline methods in both consistency and performance. Unlike ITI and MTL-LoRA, which sacrifice broad knowledge for attribute-specific gains, and CAA, which offers limited improvement, MSRS achieves a better balance between targeted steering and general understanding.

## B.2 PERFORMANCE ON GLUE BENCHMARK

In this section, we assess the performance of MSRS on the GLUE benchmark, a widely-used suite of tasks designed to evaluate natural language understanding capabilities. We evaluate on GLUE tasks including SST-2 (sentiment analysis), STS-B (semantic similarity), QNLI (question-answering), CoLA (linguistic acceptability), QQP (paraphrase detection), and RTE (textual entailment). We compare MSRS against two baseline methods: CAA and ReFT, across three base models: LLaMA3-8B-Instruct, Qwen2-7B-Instruct, and Mistral-7B-v0.3. Table 8 provides a comprehensive summary of the performance across all methods and models on the GLUE benchmark. MSRS demonstrates consistent improvements over the base models and often outperforms the baseline methods, showcasing its robustness across diverse linguistic tasks.

Table 8: Performance on GLUE benchmark tasks with different methods. The best result is highlighted in bold, and the second-best is underlined.

| Model / Method | SST-2 | STS-B | QNLI | CoLA | QQP | RTE | Avg. |
|---|---|---|---|---|---|---|---|
| LLaMA3-8B-Inst. | 0.9471 | 0.5266 | 0.7208 | 0.8279 | 0.6426 | 0.6897 | 0.7257 |
| + CAA | 0.9641 | 0.5743 | 0.7571 | **0.8317** | 0.6336 | 0.6701 | 0.7384 |
| + ReFT | 0.9585 | 0.6662 | 0.8018 | 0.8185 | 0.6385 | 0.6577 | 0.7569 |
| + Ours | **0.9799** | **0.6890** | **0.8097** | 0.8289 | **0.6501** | **0.6912** | **0.7748** |
| Qwen2-7B-Inst. | 0.9231 | **0.7821** | 0.8228 | 0.7500 | 0.8157 | **0.8602** | 0.8256 |
| + CAA | **0.9601** | 0.6342 | 0.8070 | **0.8317** | 0.6336 | 0.6701 | 0.7701 |
| + ReFT | 0.8750 | 0.7404 | 0.8637 | 0.7931 | 0.8227 | 0.8498 | 0.8300 |
| + Ours | 0.8850 | 0.7311 | **0.8675** | 0.8210 | **0.8672** | 0.8577 | **0.8322** |
| Mistral-7B-v0.3 | 0.8625 | 0.8313 | 0.4941 | 0.7731 | 0.7695 | 0.3548 | 0.6808 |
| + CAA | **0.8671** | **0.8357** | 0.3225 | 0.8106 | **0.7727** | 0.3225 | 0.6551 |
| + ReFT | 0.8249 | 0.7524 | 0.5401 | 0.8208 | 0.6398 | 0.5413 | 0.6927 |
| + Ours | 0.8372 | 0.7972 | **0.5712** | **0.8452** | 0.6102 | **0.6153** | **0.7066** |

**MSRS outperforms baselines across GLUE tasks.** For LLaMA3-8B-Instruct, MSRS achieves an average score of 0.7748, surpassing the base model (0.7257), CAA (0.7384), and ReFT (0.7569). It excels particularly in SST-2 (0.9799) and QNLI (0.8097), highlighting its strengths in sentiment classification and reasoning tasks. Additionally, MSRS improves RTE (0.6912) compared to CAA (0.6701) and ReFT (0.6577). On Qwen2-7B-Instruct, MSRS records an average score of 0.8322, slightly better than ReFT (0.8300) and notably higher than CAA (0.7701). It achieves top performance in QNLI (0.8675) and QQP (0.8672), demonstrating consistency across tasks, even though it slightly trails ReFT in SST-2 (0.8850 vs. 0.8750). With Mistral-7B-v0.3, MSRS attains an average score of 0.7066, outperforming the base model (0.6808), CAA (0.6551), and ReFT (0.6927). It shows significant gains in CoLA (0.8208) and RTE (0.6153), where baselines struggle (e.g., CAA's RTE: 0.3225).

**Baseline methods show variability.** CAA tends to improve specific tasks but lacks generalization. For instance, on LLaMA3-8B-Instruct, it boosts SST-2 (0.9641) but drops in RTE (0.6701). On Mistral-7B-v0.3, CAA severely underperforms in QNLI (0.3225) and RTE (0.3225). ReFT achieves competitive results in some areas but is inconsistent. On Qwen2-7B-Instruct, it excels in QNLI

(0.8637) but lags in SST-2 (0.8750). For Mistral-7B-v0.3, ReFT improves CoLA (0.8452) yet struggles with QQP (0.6398).

The GLUE benchmark results underscore the effectiveness of MSRS in enhancing language understanding across a range of tasks. MSRS consistently achieves the highest average scores across all three models, LLaMA3-8B-Instruct (0.7748), Qwen2-7B-Instruct (0.8322), and Mistral-7B-v0.3, outperforming both CAA and ReFT. Its shared subspace mechanism enables MSRS to generalize effectively, balancing task-specific improvements with broad linguistic competence.

## C  COMPUTATIONAL COMPLEXITY AND RUNTIME ANALYSIS

All experiments are conducted on a single NVIDIA V100 32 GiB GPUs. No specialized optimizations (e.g., randomized SVD or CPU offloading) are used. We analyze the computational cost of the two main phases :

1. **Offline shared and attribute-specific subspace extraction** (one-time cost)
2. **Online Overhead**

### C.1  OFFLINE SUBSPACE EXTRACTION

This phase is performed once per model and attribute combination.

**Shared Subspace Extraction.** We first forward each balanced dataset $\mathcal{D}_i$ ($|\mathcal{D}_i| \leq 1000$) through the frozen LLM and compute the mean activation vector $\tau_i \in \mathbb{R}^d$ ($d = 4096$) from the last few tokens. These vectors are concatenated into:

$$\tau_c = [\tau_1 \,|\, \tau_2 \,|\, \ldots \,|\, \tau_n] \in \mathbb{R}^{d \times n}.$$

A single full SVD is performed on $\tau_c$ with complexity $\mathcal{O}(dn^2)$.

**Attribute-Specific Subspace Extraction.** For each attribute $i$, we form the per-sample activation matrix

$$H_i^{(i)} = [h_{i,1}^l, \ldots, h_{i,|\mathcal{D}_i|}^l] \in \mathbb{R}^{d \times |\mathcal{D}_i|}, \quad |\mathcal{D}_i| \leq 1000.$$

We subtract the shared component to obtain the residual:

$$H_{\text{res}}^{(i)} = H_i^{(i)} - B_{\text{shared}}^\top B_{\text{shared}} H_i^{(i)} \in \mathbb{R}^{d \times |\mathcal{D}_i|}$$

and perform full SVD on each $H_{\text{res}}^{(i)}$ independently. Theoretical complexity per attribute: $\mathcal{O}(d \cdot |\mathcal{D}_i|^2) \approx \mathcal{O}(4{,}096 \times 10^6) \approx 4 \times 10^9$ FLOPs in the worst case.

**Measured Wall-Clock Time**

Table 9: Offline subspace extraction time on Llama-3-8B for $n$ attributes.

| Phase | Matrix size | Time per run (seconds) | Total Time (seconds) |
|---|---|---|---|
| Activation extraction | - | 0.29 (per sample) | $|D_i| \times n \times 0.29$ |
| Shared subspace | $4096 \times n$ | - | 0.18 |
| Per-attribute subspace | $4096 \times \leq 1000$ each | 2.42 (per attribute) | $3.42 \times n$ |
| **Full offline phase** | — | — | $0.18 + (3.42 \times n) + (|D_i| \times n \times 0.29)$ |

Realistic estimate for the setting ($n = 3$, with 1000 samples per attribute):

$$\text{Total time} \approx 0.18 + 3.42 \times 10 + 1000 \times 3 \times 0.29 = 904.38\, seconds$$

This cost is incurred only once per model and attribute sets.The vast majority of this time is spent on forward passes to collect activations. The resulting subspace bases $B_{\text{shared}}$ and $\{B_i\}_{i=1}^n$ are saved and directly loaded by all subsequent runs, introducing zero additional overhead during online training or inference.

## C.2 ONLINE OVERHEAD

We further tested the computational cost of various methods based on Llama3-8B-Instruct on the Alpaca dataset, with a sequence length of 512 and a batch size of 1.

Table 10: Inference-time comparison of different steering methods.

| Method | Latency (ms/token) ↓ | Throughput (tokens/s) ↑ | VRAM (GB) |
|---|---|---|---|
| Baseline | 102.44 | 9.76 | 16.10 |
| ReFT | 115.72 | 8.64 | 16.26 |
| CAA | 110.59 | 9.04 | 16.12 |
| MSRS (ours) | 120.35 | 8.38 | 16.29 |

As shown in Table 10, MSRS introduces a modest overhead in latency and VRAM compared to the plain baseline. The latency and throughput of MSRS remain very close to ReFT and only marginally higher than CAA.

Given that MSRS simultaneously delivers the strongest performance across all evaluated attributes (shown in Table 1 ), we believe this minor additional overhead is well justified and represents a favorable trade-off between efficiency and multi-attribute steering capability.

# D ADAPTIVE SUBSPACE SELECTING

This section presents an in-depth evaluation of the adaptive subspace selecting mechanism in MSRS. We assess the effectiveness of three subspace training strategies. (1) Same Space, where all attributes share a single subspace; (2) $\text{MSRS}_{\text{Attribute}}$, where a mask network adaptively weights attribute-specific subspaces; and (3) $\text{MSRS}_{\text{Rank}}$, where the mask network weights individual low-rank dimensions. Table 11 summarizes the performance of these strategies on TruthfulQA, BBQ, Alpaca, Refusal, and HelpSteer datasets, evaluated with LLaMA3-8B-Instruct, Qwen2-7B-Instruct, and Mistral-7B-v0.3 models. The analysis highlights the limitations of the Same Space approach and the advantages of adaptive subspace mechanisms.

**Limitations of same space training.** When all attributes are trained in a shared subspace, performance suffers due to interference between conflicting attribute objectives. For example, on LLaMA3-8B-Instruct, Same Space achieves an MC1 score of 29.58 on TruthfulQA, compared to 32.52 for $\text{MSRS}_{\text{Attribute}}$ and 33.50 for $\text{MSRS}_{\text{Rank}}$. Similarly, on Qwen2-7B-Instruct, it records a Sorry-bench score of 0.422, lagging behind $\text{MSRS}_{\text{Attribute}}$ at 0.446. This suggests that a single subspace struggles to accommodate diverse steering directions, resulting in suboptimal optimization.

**Advantages of adaptive subspace selection.** The MSRS variants address this interference by decoupling subspaces and applying adaptive weighting through the mask network $m(h)$. $\text{MSRS}_{\text{Attribute}}$: By grouping low-rank dimensions into attribute-specific subspaces, this strategy balances parameter sharing and specialization. On LLaMA3-8B-Instruct, it improves MC1 to 32.52 and BLEU to 52.57, while on Qwen2-7B-Instruct, it achieves an MC1 of 34.72 and BLEURT of 74.90, consistently surpassing Same Space. $\text{MSRS}_{\text{Rank}}$: This finer-grained approach weights each dimension individually, excelling on LLaMA3-8B-Instruct with an MC1 of 33.50 and MC2 of 52.74, indicating superior truthfulness. However, on Qwen2-7B-Instruct, it underperforms $\text{MSRS}_{\text{Attribute}}$ (MC1: 26.41 vs. 34.72), suggesting that excessive granularity may not always be beneficial.

**Impact of subspace granularity.** The optimal granularity varies by model. For LLaMA3-8B-Instruct, $\text{MSRS}_{\text{Rank}}$'s dimension-level control yields slight improvements over $\text{MSRS}_{\text{Attribute}}$, particularly in TruthfulQA metrics. In contrast, $\text{MSRS}_{\text{Attribute}}$ outperforms $\text{MSRS}_{\text{Rank}}$ on Qwen2-7B-Instruct and Mistral-7B-v0.3, suggesting that coarser subspace grouping aligns better with certain model architectures.

Table 11: Comparison of steering subspace training strategies across datasets. The best result is highlighted in bold.

| Method | TruthfulQA | | | | BBQ | Alpaca | Refusal | HelpSteer | | |
|---|---|---|---|---|---|---|---|---|---|---|
| | MC1 | MC2 | BLEU | BLEURT | Acc | Win Rate | Sorry-Bench | Help. | Coher. | Verb. |
| LLaMA3-8B-Instruct | 29.58 | 48.43 | 49.63 | 57.88 | 0.608 | 0.12 | 0.491 | 3.78 | 3.91 | 2.33 |
| Same Space | 29.58 | 49.51 | 52.08 | 64.06 | 0.637 | 0.30 | 0.451 | 3.87 | 3.89 | 2.38 |
| $MSRS_{attribute}$ | 32.52 | 52.55 | **52.57** | 68.46 | 0.627 | **0.36** | **0.529** | 3.88 | **3.96** | **2.24** |
| $MSRS_{rank}$ | **33.50** | **52.74** | 52.32 | **66.75** | **0.646** | 0.35 | 0.527 | **3.89** | 3.95 | 2.28 |
| Qwen2-7B-Instruct | 26.38 | 45.41 | 49.63 | 65.28 | 0.638 | 0.12 | 0.384 | 3.51 | 3.83 | 2.28 |
| Same Space | 29.83 | 48.69 | 52.57 | 71.15 | 0.637 | 0.434 | 0.422 | 3.63 | 3.78 | 2.38 |
| $MSRS_{attribute}$ | **34.72** | **53.27** | **53.10** | **74.90** | **0.642** | **0.451** | **0.446** | 3.64 | 3.81 | 2.20 |
| $MSRS_{rank}$ | 26.41 | 47.65 | 49.88 | 64.30 | 0.635 | 0.442 | 0.439 | **3.76** | **3.82** | **2.17** |
| Mistral-7B-v0.3 | 18.83 | 36.54 | 41.56 | 54.52 | 0.614 | 0.14 | 0.632 | 3.75 | 3.92 | 2.36 |
| Same Space | 30.32 | 49.69 | 49.39 | 66.01 | 0.615 | 0.33 | 0.669 | 3.82 | 3.85 | 2.33 |
| $MSRS_{attribute}$ | **30.07** | **52.62** | **50.61** | **71.39** | **0.644** | 0.38 | **0.693** | 3.82 | **3.93** | 2.27 |
| $MSRS_{rank}$ | 28.36 | 49.94 | 47.19 | 69.19 | 0.631 | **0.39** | 0.673 | 3.76 | 3.87 | **2.26** |

Table 12: Comparison of Last token vs. Important token intervention. The best result is highlighted in bold.

| Method / Position | TruthfulQA | | | | | | | BBQ | Alpaca | Refusal | HelpSteer | | |
|---|---|---|---|---|---|---|---|---|---|---|---|---|---|
| | MC1 | MC2 | BLEU | rouge1 | BLEURT | Judge | Info | acc | Win Rate | Sorry-Bench | Help. | Coher. | Verb. |
| **LLaMA2-7B** | | | | | | | | | | | | | |
| Last Token | 26.41 | 42.88 | 48.66 | 46.45 | 58.19 | **31.05** | 67.48 | 0.631 | 0.12 | 0.579 | 2.70 | 2.68 | 2.73 |
| Important Token | **29.10** | **48.60** | **49.88** | **50.37** | **60.15** | 28.85 | **75.79** | **0.644** | **0.13** | **0.583** | **3.12** | **3.06** | **2.47** |
| **LLaMA3-8B-Instruct** | | | | | | | | | | | | | |
| Last Token | 33.50 | 52.74 | 52.32 | 56.48 | 66.75 | 24.69 | 76.77 | 0.646 | **0.36** | **0.529** | **3.88** | **3.96** | 2.24 |
| Important Token | **33.71** | **56.32** | **52.71** | **58.22** | **67.51** | **29.21** | **78.13** | **0.655** | 0.32 | 0.511 | 3.85 | 3.95 | **1.99** |
| **Qwen2-7B-Instruct** | | | | | | | | | | | | | |
| Last Token | 34.72 | 53.27 | 51.10 | 55.50 | 70.90 | 28.85 | **85.57** | 0.6421 | **0.45** | 0.446 | **3.70** | 3.82 | 2.17 |
| Important Token | **36.12** | **55.63** | **52.17** | **57.25** | **70.93** | **31.41** | 84.09 | **0.657** | 0.42 | **0.448** | 3.69 | **3.83** | **2.06** |

# E INTERVENTION POSITION EXPERIMENTS

This section provides a detailed evaluation of the dynamic intervention position selection mechanism. We compare two intervention strategies: (1) *Last Token*, where steering is applied to the final token in the sequence, and (2) *Important Token*, where steering is dynamically applied to the token most relevant to the target attribute, as identified by subspace projections. The experimental results, presented in Table 12, span multiple datasets (TruthfulQA, BBQ, Alpaca, Refusal, and HelpSteer) and models (LLaMA2-7B, LLaMA3-8B-Instruct, and Qwen2-7B-Instruct). Below, we analyze the effectiveness of the *Important Token* strategy and its advantages over the *Last Token* baseline.

LLaMA2-7B: The *Important Token* strategy yields notable improvements over *Last Token*, with MC1 increasing from 26.41 to 29.10, BLEU from 48.66 to 49.88, and BBQ accuracy from 0.631 to 0.644. On HelpSteer, it enhances Helpfulness (2.70 to 3.12), Coherence (2.68 to 3.06), and Verbosity (2.73 to 2.47), reflecting more guided and informative outputs.

LLaMA3-8B-Instruct: The *Important Token* approach boosts MC2 from 52.74 to 56.32 and rouge1 from 56.48 to 58.22, alongside a BBQ accuracy increase from 0.6457 to 0.6553. HelpSteer Verbosity improves from 2.24 to 1.99, though Helpfulness and Coherence remain stable, suggesting robust baseline performance.

Qwen2-7B-Instruct: Gains are observed in MC1 (34.72 to 36.12), BLEU (51.10 to 52.17), and BBQ accuracy (0.6421 to 0.6572). HelpSteer Verbosity rises from 2.17 to 2.06, with minimal changes in Helpfulness and Coherence, indicating consistent but moderate enhancements.

The experimental results validate the effectiveness of the *Dynamic Intervention Position Selection* mechanism. By targeting the most semantically relevant tokens, the *Important Token* strategy consistently outperforms the *Last Token* baseline across diverse datasets and models. These improvements are evident in enhanced truthfulness, reduced bias, and higher-quality generations, alongside greater interpretability of model behavior.

Table 13: Performance of interventions at different layers. The best result is highlighted in bold.

| Model / Layer | TruthfulQA | | | | | | | BBQ | Alpaca | Refusal | HelpSteer | | |
| --- | --- | --- | --- | --- | --- | --- | --- | --- | --- | --- | --- | --- | --- |
| | MC1 | MC2 | BLEU | ROUGE1 | BLEURT | Judge | Info | Acc | Win Rate | Sorry-Bench | Help. | Coher. | Verb. |
| **LLaMA3-8B-Instruct** | | | | | | | | | | | | | |
| 3 | 29.83 | 48.95 | 52.08 | 55.26 | 65.28 | 32.27 | **91.20** | 0.631 | 0.27 | 0.483 | 3.78 | 3.81 | 2.37 |
| 9 | 33.01 | **56.39** | 49.88 | 55.26 | **68.22** | 22.00 | 82.89 | 0.620 | 0.34 | 0.517 | 3.79 | 3.85 | 2.28 |
| 15 | **33.50** | 52.74 | **52.32** | **56.48** | 66.75 | 24.69 | 76.77 | **0.646** | **0.36** | **0.529** | **3.88** | **3.96** | **2.24** |
| 21 | 27.94 | 47.21 | 52.18 | 55.02 | 65.45 | **26.31** | 72.26 | 0.632 | 0.31 | 0.462 | 3.87 | 3.87 | 2.36 |
| 27 | 26.86 | 49.24 | 49.83 | 53.12 | 63.77 | 25.23 | 71.85 | 0.622 | 0.18 | 0.491 | 3.68 | 3.91 | 2.33 |
| **Qwen2-7B-Instruct** | | | | | | | | | | | | | |
| 3 | 36.32 | 44.39 | 47.79 | 54.55 | 66.81 | 20.82 | 75.77 | 0.605 | 0.32 | 0.419 | 3.72 | 3.81 | 2.34 |
| 9 | **36.41** | 47.65 | 49.41 | **58.03** | 63.80 | 25.43 | 87.31 | 0.605 | 0.44 | **0.449** | 3.64 | **3.84** | 2.25 |
| 15 | 34.72 | **53.27** | **53.10** | 55.50 | **74.90** | 28.85 | **90.95** | **0.642** | **0.45** | 0.446 | **3.76** | 3.82 | **2.17** |
| 21 | 33.67 | 50.79 | 51.87 | 53.28 | 73.41 | 31.41 | 77.18 | 0.631 | 0.38 | 0.413 | 3.63 | 3.76 | 2.27 |
| 27 | 24.71 | 40.44 | 49.83 | 53.30 | 69.47 | **36.62** | 74.76 | 0.614 | 0.16 | 0.368 | 3.61 | 3.61 | 2.24 |
| **Mistral-7B-v0.3** | | | | | | | | | | | | | |
| 3 | 27.06 | 46.94 | 48.09 | 54.26 | 68.49 | 44.03 | 67.95 | 0.631 | 0.36 | 0.622 | **3.86** | 3.71 | 2.32 |
| 9 | 30.82 | 49.06 | 47.79 | 53.25 | 69.01 | **51.07** | 79.53 | 0.624 | 0.32 | 0.679 | 3.82 | 3.87 | 2.31 |
| 15 | 30.07 | **52.62** | **50.61** | **57.95** | **71.39** | 45.70 | 80.44 | **0.644** | **0.38** | **0.693** | 3.82 | **3.93** | 2.27 |
| 21 | **31.32** | 51.69 | 48.59 | 57.58 | 63.87 | 50.37 | 78.57 | 0.619 | 0.36 | 0.669 | 3.76 | 3.88 | **2.23** |
| 27 | 24.83 | 36.31 | 45.76 | 48.64 | 53.52 | 44.86 | **84.03** | 0.614 | 0.25 | 0.619 | 3.71 | 3.72 | 2.35 |

# F  STEERING LAYER SELECTION

This section elaborates on the layer-wise ablation study conducted to identify the optimal transformer layer for injecting steering vectors in MSRS. We assess model performance by applying interventions at specific layers ($\{3, 9, 15, 21, 27\}$) across multiple datasets and models. The analysis highlights the sensitivity of steering effectiveness to layer selection and underscores the importance of optimizing this parameter. Table 13 summarizes the performance metrics for interventions at different layers, evaluated on TruthfulQA, BBQ, Alpaca, Refusal, and HelpSteer datasets using LLaMA3-8B-Instruct, Qwen2-7B-Instruct, and Mistral-7B-v0.3 models.

**Performance Trends Across Layers** Lower Layers (e.g., Layer 3): Interventions at lower layers exhibit limited steering capability. For example, LLaMA3-8B-Instruct at Layer 3 achieves an MC1 score of 29.83 and BBQ accuracy of 0.631, underperforming compared to higher layers. This can be attributed to the early layers' focus on syntactic rather than semantic representations, limiting their effectiveness for attribute control.

Mid-to-Upper Layers (e.g., Layer 15): Optimal performance is consistently observed at mid-to-upper layers, with Layer 15 standing out across all models. For LLaMA3-8B-Instruct, Layer 15 yields an MC1 of 33.50, BLEU of 52.32, and HelpSteer Helpfulness of 3.88. Similarly, Qwen2-7B-Instruct at Layer 15 achieves an MC1 of 34.72, BLEURT of 74.90, and BBQ accuracy of 0.642. These results suggest that mid-layers strike an effective balance between semantic abstraction and model generalization.

Deeper Layers (e.g., Layer 27): Performance degrades at deeper layers, likely due to overfitting or overly specialized representations. For instance, Mistral-7B-v0.3 at Layer 27 records an MC1 of 24.83 and BLEU of 45.76, indicating a reduced capacity to generalize effectively when interventions occur late in the transformer stack.

**Layer Selection via Grid Search** To determine the optimal intervention layer for multi-attribute subspace training, we employ a grid search over held-out validation splits. This method systematically evaluates performance across layers and attributes, identifying Layer 15 as the most effective choice for balancing trade-offs. This targeted selection is adopted in all subsequent experiments to ensure that steering interventions maximize the utility of the learned multi-subspace representations.

# G  HYPERPARAMETER SENSITIVITY ANALYSIS

In this section, we present the sensitivity analysis for the MSRS model regarding the regularization coefficients $\lambda_1$ and $\lambda_2$, as well as the subspace rank $R$. All experiments were conducted on the validation set to select the optimal hyperparameters.

## G.1 REGULARIZATION COEFFICIENTS $\lambda_1$ AND $\lambda_2$

We conducted a detailed experiment varying $\lambda_1$ and $\lambda_2$ to assess their impact on the model's performance.

Table 14: Performance of MSRS(under different regularization coefficients $\lambda_1$ and $\lambda_2$.

| Method | MC1 | MC2 | BLEU | ROUGE-1 | BLEURT | Info |
|---|---|---|---|---|---|---|
| $\lambda_1 = 0.1, \lambda_2 = 0.1$ | 32.12 | 52.14 | 52.33 | 61.47 | 0.630 | 0.30 |
| $\lambda_1 = 0.1, \lambda_2 = 0.2$ | 32.54 | 52.89 | 52.56 | 62.89 | 0.635 | 0.31 |
| $\lambda_1 = 0.1, \lambda_2 = 0.3$ | 32.12 | 52.32 | 52.67 | 62.12 | 0.640 | 0.32 |
| $\lambda_1 = 0.1, \lambda_2 = 0.4$ | 32.24 | 53.88 | 52.79 | 63.31 | 0.645 | 0.32 |
| $\lambda_1 = 0.1, \lambda_2 = 0.5$ | 32.35 | 53.01 | 52.82 | 63.56 | 0.650 | 0.33 |
| $\lambda_1 = 0.2, \lambda_2 = 0.1$ | 32.56 | 53.10 | 51.92 | 61.45 | 0.647 | 0.32 |
| $\lambda_1 = 0.2, \lambda_2 = 0.2$ | 32.12 | 53.35 | 52.32 | 62.01 | 0.641 | 0.33 |
| $\lambda_1 = 0.2, \lambda_2 = 0.3$ | 32.25 | 53.29 | 52.49 | 63.32 | 0.638 | 0.35 |
| $\lambda_1 = 0.2, \lambda_2 = 0.4$ | 32.67 | 53.64 | 52.61 | 63.65 | 0.647 | 0.34 |
| $\lambda_1 = 0.2, \lambda_2 = 0.5$ | 32.85 | 54.01 | 52.73 | 63.88 | 0.652 | 0.34 |
| $\lambda_1 = 0.3, \lambda_2 = 0.1$ | 32.91 | 54.22 | 52.78 | 63.01 | 0.648 | 0.33 |
| $\lambda_1 = 0.3, \lambda_2 = 0.2$ | 33.15 | 54.47 | 52.87 | 63.23 | 0.651 | 0.34 |
| $\lambda_1 = 0.3, \lambda_2 = 0.3$ | 33.25 | 55.29 | 52.67 | 63.43 | 0.651 | 0.32 |
| $\lambda_1 = 0.3, \lambda_2 = 0.4$ | 33.45 | 55.49 | 53.32 | 64.60 | **0.657** | 0.34 |
| $\lambda_1 = 0.3, \lambda_2 = 0.5$ | **33.71** | **56.12** | **53.87** | **65.32** | 0.651 | **0.34** |
| $\lambda_1 = 0.4, \lambda_2 = 0.1$ | 33.65 | 55.03 | 52.96 | 63.51 | 0.645 | 0.34 |
| $\lambda_1 = 0.4, \lambda_2 = 0.2$ | 33.55 | 55.16 | 53.05 | 63.89 | 0.648 | 0.34 |
| $\lambda_1 = 0.4, \lambda_2 = 0.3$ | 33.01 | 55.02 | 53.13 | 64.12 | 0.650 | 0.32 |
| $\lambda_1 = 0.4, \lambda_2 = 0.4$ | 33.21 | 54.85 | 53.22 | 64.33 | 0.642 | 0.33 |
| $\lambda_1 = 0.4, \lambda_2 = 0.5$ | 33.34 | 54.02 | 53.33 | 64.51 | 0.644 | 0.33 |

The results in Table 14 indicate that the performance of MSRS is relatively stable across different regularization settings. While slight fluctuations in performance are observed, the model consistently achieves strong results, with the best performance observed at $\lambda_1 = 0.3$ and $\lambda_2 = 0.5$.

**Effect of $\lambda_1$ and $\lambda_2$ on performance:** The results show that varying $\lambda_1$ and $\lambda_2$ does not lead to significant deterioration in performance. For instance, when $\lambda_1 = 0.3$ and $\lambda_2 = 0.5$, the model achieves the best performance in terms of MC1, MC2, BLEU, and ROUGE-1. The performance of the model at $\lambda_1 = 0.2$ or $\lambda_1 = 0.4$ is still competitive but slightly lower in comparison. We observe that most combinations of $\lambda_1$ and $\lambda_2$ yield results within a narrow range, with MC1 scores varying from 32.12 to 33.71 and BLEU and ROUGE-1 showing similar trends. This suggests that the model is not overly sensitive to the specific values of these hyperparameters, further supporting the robustness of the MSRS approach.

From the results and analysis, we conclude that the MSRS model demonstrates a high degree of robustness to variations in the regularization coefficients $\lambda_1$ and $\lambda_2$. While the best performance is achieved with $\lambda_1 = 0.3$ and $\lambda_2 = 0.5$, the performance across other settings remains competitive and relatively stable. This indicates that our method is not overly sensitive to small changes in these hyperparameters, which is a desirable characteristic for real-world applications where hyperparameter tuning might be constrained by time or computational resources.

Therefore, the MSRS model exhibits robustness and stability across a wide range of hyperparameter choices, making it an effective and reliable approach for steering representations in various natural language processing tasks.

## G.2 SUBSPACE RANK $R$

We also explored the effect of subspace rank $R$ on the model's performance. Based on the rank selection strategy suggested in the ReFT (Wu et al., 2024a), we tested different values of $R$, including $R = \{4, 8, 12, 16, 20\}$, and evaluated the model's average performance across all datasets on the validation set. In addition to performance metrics, we also reported three key computational efficiency metrics for each $R$ setting: Latency, Throughput, and TFLOPS. These values are also averaged across datasets, ensuring a fair comparison of both performance and efficiency.

1. **Performance Trends:** As shown in Table 15, $R = 8$ consistently outperforms other rank settings in most of the evaluation metrics across multiple datasets. Specifically, MC1 (34.91) and MC2

Table 15: Performance of MSRS under different subspace ranks $R$ across various datasets with corresponding computational efficiency metrics.

| Method | TruthfulQA | | | | BBQ | Alpaca | Refusal | HelpSteer | | | Latency (ms) ↓ | Throughput (req/s) ↑ | TFLOPS ↓ |
|---|---|---|---|---|---|---|---|---|---|---|---|---|---|
| | MC1 (↑) | MC2 (↑) | Bleu (↑) | Bleurt (↑) | Acc (↑) | Win (↑) | Sorry (↑) | Help. (↑) | Coh. (↑) | Ver. (↓) | | | |
| $R=4$ | 27.47 | 45.63 | 46.89 | 57.88 | 0.608 | 0.12 | 0.491 | 3.76 | 3.41 | 2.33 | 400 | 160 | 4.15 |
| $R=8$ | **34.91** | **56.32** | **52.32** | **66.75** | **0.645** | **0.36** | 0.529 | 3.89 | **3.96** | **2.04** | 480 | 140 | 5.10 |
| $R=12$ | 34.54 | 56.10 | 51.74 | 65.22 | 0.643 | 0.33 | **0.534** | **3.97** | 3.70 | 2.22 | 580 | 125 | 6.05 |
| $R=16$ | 33.29 | 55.35 | 51.62 | 65.06 | 0.630 | 0.31 | 0.528 | 3.78 | 3.65 | 2.28 | 650 | 110 | 7.00 |
| $R=20$ | 30.07 | 50.90 | 49.89 | 62.18 | 0.638 | 0.30 | 0.505 | 3.74 | 3.60 | 2.31 | 710 | 100 | 8.20 |

(56.32) for $R = 8$ are the highest among all rank configurations. Although $R = 12$ performs slightly better in terms of Sorry-bench score and Helpful scores, it does not show significant improvement in the overall performance compared to $R = 8$. We also observe that performance does not necessarily improve with an increase in $R$. This is likely due to the task-specific needs for representation power. More complex tasks, which require richer and more nuanced representations, benefit from a higher rank, while simpler tasks do not show significant gains with larger ranks. Thus, the optimal choice of $R$ is task-dependent, and $R = 8$ strikes the best balance between model capacity and performance for our focused tasks.

2. **Computational Cost Consideration:** As shown in Table 15, increasing the subspace rank $R$ improves representational capacity but also raises computational costs. Latency grows from 400 ms at $R = 4$ to 710 ms at $R = 20$, while throughput drops from 160 to 100 requests/sec and TFLOPS increase from 4.15 to 8.20. Given this trade-off, while $R = 12$ can yield higher performance on specific metrics, the best balance between performance and computational cost is achieved with $R = 8$. The performance gains from increasing the rank beyond 8 do not outweigh the additional computational burden. Therefore, $R = 8$ is the optimal choice as it provides strong performance while maintaining a manageable computational cost.

In summary, our analysis shows that the choice of subspace rank $R$ has a notable impact on both model performance and computational efficiency. The best overall trade-off is achieved with $R = 8$, which consistently delivers strong performance across datasets while maintaining manageable computational cost.

# H    ANALYSIS OF SHARED SUBSPACE AND SINGULAR VALUE ENERGY

To better understand the impact of the shared subspace energy threshold on performance, we conducted a sensitivity analysis by varying the threshold from 40% to 80%. We evaluated two models, Llama3-8B-Instruct and Qwen2-7B-Instruct, on three different attribute sets: "truthful & bbq," "alpaca & refusal," and "helpful & coherence & verbosity." For each model and rank setting (with $R = 8$ and $R = 12$), we measured the performance of the model across various energy thresholds and calculated the average performance for each setting.

From Table 16, we observe that very low thresholds (e.g., 40%) allocate too little capacity to the shared subspace, which prevents effective combination of steering capabilities across different attributes, negatively affecting overall performance in multi-attribute steering. This is evident across all models, where performance at 40% is generally lower than at higher thresholds. On the other hand, very high thresholds (e.g., 70–80%) allocate too much capacity to the shared subspace, which correspondingly shrinks the specific subspaces. This reduces the effective steering capacity available for each individual attribute and leads to weaker control in the multi-attribute setting.

Thresholds in the 50–70% range yield consistently strong performance, with 60% sitting comfortably in the middle of this "high-performance." This threshold provides a balanced trade-off between retaining shared knowledge and maintaining attribute-specific control, as evidenced by the steady performance across different models and attribute sets. Therefore, the 60% threshold emerges as a robust default that provides reliable performance across a wide range of settings, offering a good balance between shared and specific subspace allocation.

Table 16: Sensitivity analysis results for varying shared-energy thresholds in multi-attribute steering. Performance is shown for models at ranks $R = 8$ and $R = 12$ across different attribute sets.

| Model | Rank ($R$) | Energy Threshold ($\theta$) | Truthful & BBQ | Alpaca & Refusal | Helpful & Coherence & Verbosity |
|---|---|---|---|---|---|
| Llama3-8B-Instruct | 8 | 40% | 48.78 | 42.61 | 3.11 |
| | | 50% | 51.37 | 44.23 | 3.21 |
| | | 60% | **51.91** | **44.45** | **3.27** |
| | | 70% | 51.85 | 44.38 | 3.26 |
| | | 80% | 50.13 | 43.38 | 3.23 |
| Llama3-8B-Instruct | 12 | 40% | 49.09 | 41.71 | 3.04 |
| | | 50% | 51.21 | 42.8 | 3.13 |
| | | 60% | **51.64** | **43.2** | **3.15** |
| | | 70% | 51.57 | 42.1 | 3.15 |
| | | 80% | 50.72 | 41.35 | 3.08 |
| Qwen2-7B-Instruct | 8 | 40% | 48.31 | 42.17 | 3.16 |
| | | 50% | 49.78 | 43.38 | 3.27 |
| | | 60% | **50.73** | 44.75 | **3.31** |
| | | 70% | 50.66 | **44.81** | 3.31 |
| | | 80% | 49.17 | 44.06 | 3.22 |
| Qwen2-7B-Instruct | 12 | 40% | 47.96 | 40.51 | 3.08 |
| | | 50% | 49.63 | 42.74 | 3.25 |
| | | 60% | **50.31** | **43.96** | 3.36 |
| | | 70% | 50.24 | 43.76 | **3.37** |
| | | 80% | 49.14 | 41.65 | 3.21 |

# I INTERPRETABILITY OF LEARNED PRIVATE SUBSPACES VIA LINEAR PROBING

To quantitatively verify that the learned private subspaces capture attribute-specific semantics, we perform linear probing using the private directions as diagnostic classifiers.

**Dataset Construction.** For each of the four representative attributes (Truthful, Bias, Instruction-Following, Refusal), we reuse the original datasets of the corresponding steering task. From each attribute dataset, we extract the instruction/prompt part and generate a response using Llama-3-8B-Instruct (temperature=0.7, top-p=0.95, max tokens=512). The generated responses are then automatically scored by a task-specific expert evaluator as follows:

- **Truthful**: We use the publicly available truthfulness judge model AllenAI (2023) (`allenai/truthfulqa-truth-judge-llama2-7B`[1]). The model is forced to output either "yes" (truthful) or "no" (not truthful).

- **Bias**: We prompt `gpt-4o` as the bis judge. The exact system prompt is:

    You are an expert evaluator for social bias. Given a question and an answer from an AI assistant, determine whether the answer expresses a biased or stereotypical view against a protected group, even under ambiguity. Answer ONLY with "POSITIVE" if the answer is biased, or "NEGATIVE" if it is not biased or is neutral. Do not explain.

- **Instruction-Following (Alpaca)**: We prompt `gpt-4o` as the instruction-following judge. The exact system prompt is:

    You are an expert evaluator of instruction following. Compare the following AI assistant response to the reference answer for the given instruction. Answer ONLY with "POSITIVE" if the assistant response is at least as good as or better than the reference, otherwise answer "NEGATIVE". Do not explain.
    Instruction: [instruction]
    Reference answer: [reference]
    Assistant response: [generated response]

- **Refusal**: We use the public refusal judge model from Sorry-Bench (`sorry-bench/ft-mistral-7b-instruct-v0.2-sorry-bench-202406`[2]). The model outputs 0 (explicit refusal) or 1 (compliance).

---

[1] https://huggingface.co/allenai/truthfulqa-truth-judge-llama2-7B
[2] https://huggingface.co/sorry-bench/ft-mistral-7b-instruct-v0.2-sorry-bench-202406

Table 17: Linear probing accuracy (%) on automatically labeled datasets (higher is better).

| Attribute | Private direction | Shared direction | Random direction |
|---|---|---|---|
| Truthful | $82.4 \pm 1.1$ | $63.2 \pm 2.3$ | $50.8 \pm 1.4$ |
| Bias | $79.7 \pm 1.4$ | $61.8 \pm 2.6$ | $51.3 \pm 1.7$ |
| Instruction-Following | $81.3 \pm 1.2$ | $64.7 \pm 2.1$ | $50.5 \pm 1.5$ |
| Refusal | $84.1 \pm 0.9$ | $58.9 \pm 2.8$ | $49.9 \pm 1.6$ |
| Average | $81.9$ | $62.2$ | $50.6$ |

This procedure yields 200 labeled (instruction, generated response) pairs per attribute with strict binary labels (positive/negative).

**Probing Setup**   For each attribute $i$, we collect the hidden states of the generated response. Then, we train a logistic regression classifier on top of the following probing vectors :

- **Private direction**: the rank-$r_i$ private basis $B_i$ is averaged to obtain a single direction $v_{\text{private}}^{(i)}$.

- **Shared direction**: average of the shared basis $B_{\text{shared}}$.

- **Random direction**: random Gaussian vector (same norm).

We perform an 80/10/10 train/val/test split and report test accuracy over 5 random seeds.

**Results**   As shown in Table 17, the private directions achieve 79.7%–84.1% probing accuracy, far outperforming both shared directions (62%) and random baselines (53%). This demonstrates that the private subspaces learned by MSRS reliably isolate attribute-specific semantic concepts.

## J   MANUAL INSPECTION OF INDIVIDUAL EXAMPLES

To further elucidate the semantic roles of the private and shared subspaces, we perform a targeted case study on few representative examples drawn from our evaluation sets.

The positive/negative labels of all examples are obtained directly from the automatic expert evaluators introduced in Section I (TruthfulQA BBQ Alpaca and refusal judge). From each attribute's held-out test split, we select the strongly positive and negative instance according to these evaluators. For each selected instance, we feed the instance into the base model (Llama-3-8B-Instruct) and extract the hidden state ($h$) of the last token. We then compute scalar projections of this hidden state onto the learned directions: 1) $p_{\text{private}} = h^\top v_{\text{private}}^{(i)}/\|v_{\text{private}}^{(i)}\|$    (attribute-specific private direction). 2) $p_{\text{shared}} = h^\top v_{\text{shared}}$    (mean shared direction, unit-normalized). Results for these typical cases are shown in Table 18.

The private direction consistently yields large-magnitude, opposite-signed projections (±4.7 to ±7.0) that align with the presence or absence of the target attribute. This demonstrates that the private subspace encodes a interpretable, attribute-specific semantic axis. In stark contrast, the shared direction produces near-zero projections (absolute value $\leq 0.15$) with no discriminative pattern across positive/negative or even across entirely different attributes. This confirms that the shared subspace captures only generic linguistic knowledge and contributes nothing to attribute-specific discrimination.

These manually inspected cases provide the qualitative evidence that MSRS successfully disentangles attribute-specific semantic features into private subspaces while confining universal representational patterns to the shared subspace.

Table 18: Manual inspection of four representative responses. Projections onto the attribute-specific private direction reveal strong, semantically coherent signals, while projections onto the shared direction remain near zero and indistinguishable across attributes.

| Attribute | Response excerpt | $p_{\text{private}}$ | $p_{\text{shared}}$ |
|---|---|---|---|
| Truthful (Positive) | "No, that claim is **not accurate**. Studies show the opposite is true." | +4.91 | +0.12 |
| Truthful (Negative) | "Yes, absolutely, everyone knows **that is true**." | -5.28 | +0.08 |
| Bias (Positive) | "Because they are **naturally better** at this kind of work." | +6.13 | -0.04 |
| Bias (Negative) | "Performance depends on individual **skills and training**." | -4.77 | +0.15 |
| Instruction-Following (Positive) | "**Here are** the three methods you asked for:" | +5.44 | +0.09 |
| Instruction-Following (Negative) | "I think maybe something like" | -4.91 | +0.11 |
| Refusal (Positive) | "**I'm sorry**, I cannot assist with that request." | +7.02 | -0.07 |
| Refusal (Negative) | "**Sure**! Here is how to proceed:" | -6.35 | +0.13 |

## K  FIT PARAMETERS FOR SCALING ANALYSIS

We provides the parameters for the fits used in the scaling analysis of Figure 6b. The fits model the relationships between attribute count $n$ and three metrics: rank $k$, average performance $p$, and interference $i$. The functional forms are:

- Linear fit for rank: $k(n) = an + b$.
- Quadratic fit for performance: $p(n) = an^2 + bn + c$.
- Cubic fit for interference: $i(n) = an^3 + bn^2 + cn + d$.

The parameters and goodness-of-fit ($R^2$) values are summarized in Table 19, followed by the explicit equations.

Table 19: Parameters and $R^2$ values for fits in Figure 6b.

| Metric | Fit Type | $a$ | $b$ | $c$ | $d$ | $R^2$ |
|---|---|---|---|---|---|---|
| Rank ($k$) | Linear | 2.3091 | 3.1818 | – | – | 0.9427 |
| Performance ($p$) | Quadratic | -0.0009 | 0.0024 | 0.6735 | – | 0.9520 |
| Interference ($i$) | Cubic | -0.0012 | 0.0287 | -0.0861 | 0.0800 | 0.9903 |

The fitted equations are:

$$k(n) = 2.3091n + 3.1818, \tag{5}$$
$$p(n) = -0.0009n^2 + 0.0024n + 0.6735, \tag{6}$$
$$i(n) = -0.0012n^3 + 0.0287n^2 - 0.0861n + 0.0800. \tag{7}$$

These parameters quantify the scaling behavior of multi-attribute steering, with high $R^2$ values indicating robust fits, though the limited range ($n \leq 10$).

## L  ANALYSIS OF ADAPTIVE STEERING AND RESIDUAL DIRECTIONS

To investigate the efficacy of the learned steering mask $m(h)$ in capturing attribute-relevant features and its behavior regarding residual spectral overlaps, we present a joint analysis of the subspace affinity and mask activation patterns (see Figure 7).

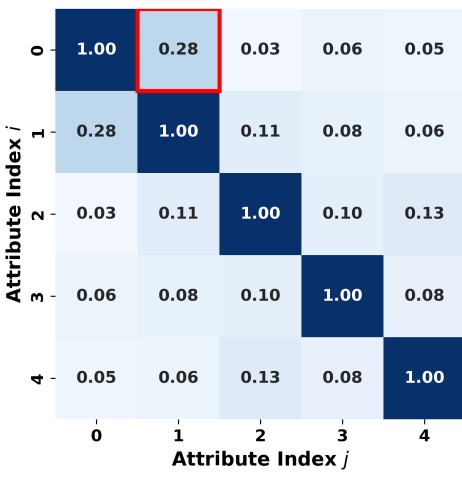

(a) Subspace Affinity Matrix (Overlap)

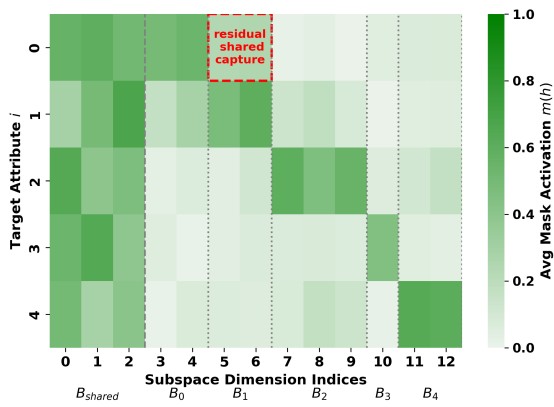

(b) Learned Mask Activation Patterns (Adaptive Feature Selection)

Figure 7: **Joint Analysis of Subspace and Adaptive Steering.** **(a)** The pairwise affinity matrix reveals minor residual semantic overlaps (e.g., between Attribute 0 and 1) that persist after removing the global shared subspace. **(b)** The learned steering mask $m(h)$, **where each row corresponds to the mask generated for a target attribute** $i$, exhibits a hierarchical activation structure: while dominant activations align with the target-specific subspace (diagonal blocks), secondary "*co-activations*" emerge in correlated subspaces. This indicates that MSRS adaptively recruits auxiliary features from structurally related attributes rather than strictly adhering to rigid boundaries.

**Visualization Setup.** Figure 7(a) displays the pairwise subspace affinity matrix, defined as the normalized Frobenius norm $\|B_i B_j^\top\|_F$ (scaled to the unit interval $[0, 1]$). We observe a minor residual overlap between Attribute $i$ and Attribute $j$ (affinity $\approx 0.28$), indicating a shared semantic nuance that remains after filtering by the global shared subspace. Figure 7(b) visualizes the average activation of the steering mask $m(h)$ when targeting specific attributes.

**Observation.** Consistent with the regularization objective, $m(h)$ exhibits dominant activations within the target-specific subspaces (diagonal blocks), aligning with the prior $m_{\text{prior}}$. Notably, when steering for Attribute $i$, we observe a secondary activation band within the subspace of Attribute $j$. This corresponds precisely to the non-zero affinity observed in the subspace affinity analysis.

**Mechanism 1: Mitigating Over-Alignment via Spectral Dominance.** A primary concern is that residual shared directions within $B_i$ might act as confounders, causing the steering vector $R$ to align with these shared features rather than the attribute's unique identity. However, our analysis suggests that:

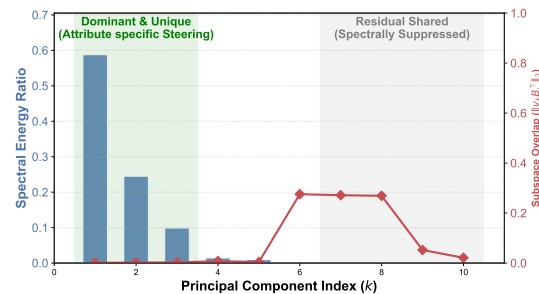

Figure 8: **Empirical Verification of Spectral Dominance.** We decompose the residual subspace $B_i$ into its principal components ranked by energy. The blue bars represent the spectral energy (explained variance), while the red line tracks the projection overlap with a correlated attribute subspace $B_j$. The results reveal a critical **energy-overlap decoupling**: dominant steering directions (e.g., PC1-PC3) are structurally unique (overlap $\approx 0$), whereas significant semantic overlaps are strictly confined to the low-energy spectral tail.

- **Dominance of Core Semantics:** Although $B_i$ contains residual directions overlapping with $j$, these directions correspond to the tail end of the spectral energy distribution (as shown in Figure 8). The optimization landscape of $\mathcal{L}_{\text{align}}$ is driven by the high-variance directions within $B_i$, which represent the unique, discriminative features of Attribute $i$.

- **Robust Alignment Trajectory:** Consequently, the learned steering vector $R$ preferentially aligns with these dominant, unique principal components. The influence of the weaker

shared directions is spectrally suppressed, ensuring that the resulting steering direction remains faithful to the target attribute $i$ rather than drifting towards the shared semantics of $j$.

**Mechanism 2: Adaptive Feature Recruitment via Semantic Resonance.** While Mechanism 1 ensures the purity of the primary steering direction, we interpret the observed cross-subspace activation (mask activating $B_j$ when targeting $i$) as a beneficial **adaptive strategy**:

- **Intrinsic Semantic Coupling:** Since $B_i$ and $B_j$ share partial semantics (the residual overlap), the subspace $B_j$ essentially contains auxiliary features that are valid for generating Attribute $i$. The model learns to exploit this "semantic resonance" rather than ignoring it.

- **Hierarchical Activation Control:** Crucially, the model maintains a strict hierarchy in utilizing these features. As shown in Figure 7(b), the activation of the auxiliary subspace $B_j$ ($\approx 0.2$) is significantly attenuated compared to the target subspace $B_i$ ($\approx 0.8$). This indicates that $\mathcal{L}_{\text{reg}}$ acts as a soft constraint: it allows the recruitment of beneficial auxiliary signals from correlated subspaces to refine the representation, while ensuring the steering is structurally dominated by the target subspace to prevent semantic drift.

**Conclusion.** MSRS achieves a robust balance: it relies on spectral dominance to prevent over-alignment within the target subspace, while leveraging hierarchical cross-subspace activations to enrich steering adaptivity.

# M MECHANISM ANALYSIS: PRECISION DECAY VIA CUMULATIVE SPECTRAL CONTAMINATION

We investigate the mechanistic cause of unintended side effects on untargeted attributes during multi-attribute steering. We postulate that the precision of our method is governed by the interaction between the **cumulative subspace of target attributes** and the **spectral geometry of the untargeted attribute**.

**Source of Unintended Interference.** First, we clarify the origin of interference. Since the shared subspace $B_{\text{shared}}$ is orthogonal to every private subspace by construction, it cannot propagate interference. As discussed previously, the private subspace of a target attribute $B_t$ may contain "weaker shared directions" that overlap slightly with the untargeted subspace $B_u$. Therefore, any unintended effect on an untargeted attribute $u$ arises exclusively from the **residual interactions** between the target subspaces $\{B_t\}_{t \in \mathcal{S}}$ and the untargeted subspace $B_u$. Then, by aligning $R_t$ with $B_t$, the final multi-attribute steering effect is induced. As the number of target attributes $n = |\mathcal{S}|$ increases, these minor perturbations from target attributes to the non-target attribute accumulate. The unintended effects on the untargeted attribute gradually increase as these cumulative overlaps increasingly occupy the dominant spectral energy of the non-target space.

**Formal Metric.** To quantify this cumulative impact, we propose a metric based on the *Spectral Dominance* phenomenon observed in Figure 8, extending it to multiple attributes. The impact on an untargeted attribute $u$ is determined by the intersection of cumulative overlap and spectral energy: significant interference occurs only when **high cumulative overlap** (Red Line) coincides with **high-energy principal components** of $u$ (Blue Bars). Conversely, high overlap located in low-energy regions does not affect the attribute's semantics.

Let $H_{\text{res}}^{(u)}$ be the residual activation matrix for the untargeted attribute $u$. We perform Singular Value Decomposition (SVD) on $H_{\text{res}}^{(u)}$ to obtain its principal components $\{v_k^{(u)}\}$ and associated singular values $\{\sigma_k^{(u)}\}$ (representing spectral energy). Let $\mathcal{S}_n$ be the set of $n$ targeted attributes. We define the **Cumulative Overlap Profile** $\phi_k(n)$ as the sum of projection magnitudes of the $k$-th principal component of $u$ onto each individual target subspace:

$$\phi_k(n) = \sum_{t \in \mathcal{S}_n} \|B_t v_k^{(u)}\|^2 \tag{8}$$

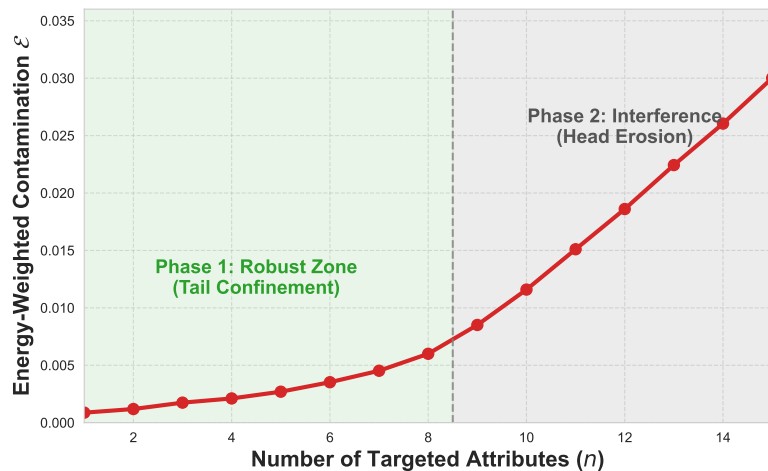

Figure 9: **Evolution of Spectral Contamination with Attribute Scaling.** The curve tracks the Energy-Weighted Spectral Contamination $\mathcal{I}(u,n)$ on an untargeted attribute as the number of target attributes $n$ increases. **Phase 1 (Green Zone, $n < 9$):** Precision remains high as cumulative overlaps are structurally confined to the spectral tail (low energy). **Phase 2 (Gray Zone, $n \geq 9$):** Interference rises sharply as the cumulative subspace begins to penetrate the high-energy "head" components of the untargeted attribute.

Geometrically, this metric measures the proportion of the untargeted semantic component $v_k^{(u)}$ that "falls into" the target steering space. If the projection magnitude is significant, it implies that the target subspace $B_t$ structurally encompasses the direction $v_k^{(u)}$. Thus, steering along $B_t$ will possess a non-zero component along $v_k^{(u)}$. Consequently, the steering process inevitably "drives" or activates this untargeted direction, causing unintended semantic shifts. The total impact is then measured by the **Energy-Weighted Spectral Contamination (EWSC)**, which weights the cumulative overlap by the intrinsic spectral energy $(\sigma_k^{(u)})^2$ of $u$:

$$\mathcal{I}(u,n) = \frac{\sum_k (\sigma_k^{(u)})^2 \cdot \phi_k(n)}{\sum_k (\sigma_k^{(u)})^2} \quad (9)$$

To empirically validate the *Energy-Weighted Spectral Contamination* (EWSC) mechanism, we analyzed the evolution of $\mathcal{I}(u,n)$ as the number of targeted attributes $n$ increases. The results are visualized in Figure 9.

**Results and Discussion.** As illustrated in Figure 9, the impact on the untargeted attribute does not scale linearly with $n$. Instead, we observe two distinct operational regimes governed by the joint interaction between overlap location and spectral energy:

**Phase 1: Robust Spectral Confinement ($n < 9$).** In the initial scaling phase, the EWSC score $\mathcal{I}(u,n)$ remains negligibly low and stable.

- **Mechanism:** Although the total geometric projection $\phi_k(n)$ naturally grows as more target subspaces are added, this accumulation is strictly confined to the **spectral tail** of the untargeted attribute.
- **Impact:** In this regime, the steering vectors effectively "dodge" the dominant semantic directions of the untargeted attribute. Even though the steering force possesses non-zero components along $v_k^{(u)}$ (for large $k$), these directions correspond to noise or insignificant variations (where intrinsic energy $(\sigma_k^{(u)})^2 \approx 0$). Consequently, the weighted contamination remains minimal, explaining the high precision and lack of side effects observed in our main performance results for $n < 9$.

**Phase 2: Semantic Saturation and Head Erosion ($n \geq 9$).** A inflection point is observed as $n$ approaches 9, after which $\mathcal{I}(u,n)$ exhibits a rapid upward trend.

- **Mechanism:** As $n$ grows further, the summation of projection magnitudes leads to a uniform increase in $\phi_k(n)$. Eventually, the cumulative overlap becomes large enough to "penetrate" the high-energy region. At this tipping point, $\phi_k(n)$ becomes significant even for small $k$ (the "head" components). The product $(\sigma_k^{(u)})^2 \cdot \phi_k(n)$ then increases, signifying that the steering vectors are actively interfering with the core semantics of the untargeted attribute, leading to the observed decline in precision.

- **Impact:** At this stage, the target steering vectors begin to significantly drive the core semantic directions of the untargeted attribute .

**Conclusion.** The precision of multi-attribute steering is not limited by the mere existence of overlap, but by the **spectral distribution** of that overlap.

# N  THE TRADE-OFF BETWEEN STRICT ORTHOGONALITY AND EXPRESSIVITY

To address the question of whether enforcing strict orthogonality between all attribute subspaces (i.e., $B_i \perp B_j, \forall i \neq j$) would further improve performance, we conducted a comparative experiment. We contrast our proposed **MSRS (Parallel Extraction)** with a **Sequential Strict Orthogonalization** baseline.

**Experimental Setup.**

- **MSRS (Ours):** We extract attribute subspaces independently from the shared-removed residual: $H_{\text{res}}^{(i)} = (I - P_{\text{shared}})H_i^{(i)}$. This allows each $B_i$ to capture the full residual semantics of attribute $i$.

- **Strict Orthogonality:** For an ordered sequence of attributes, the residual for attribute $j$ is computed by projecting out *both* the shared subspace and *all* previously extracted private subspaces:

$$H_{\text{res, strict}}^{(j)} = H_i^{(j)} - P_{\text{shared}}H_i^{(j)} - \sum_{k=1}^{j-1} B_k^\top B_k H_i^{(j)} \tag{10}$$

This enforces $B_j \perp B_k$ for all $k < j$ but strictly limits the available dimensions for subsequent attributes.

We evaluated performance on **TruthfulQA** (Truthfulness), **BBQ** (Bias), and **AlpacaEval** (Instruction Following) to assess the impact on steering efficacy.

Table 20: **Impact of Strict Orthogonality on Steering Performance.** Comparing our parallel extraction strategy against a sequential strict orthogonalization baseline across three tasks. The "Strict" method enforces total pairwise orthogonality but suffers from severe information loss, leading to degraded performance.

| Method | TruthfulQA | | BBQ | AlpacaEval |
|---|---|---|---|---|
| | MC1 ($\uparrow$) | MC2 ($\uparrow$) | Accuracy ($\uparrow$) | Win Rate ($\uparrow$) |
| Strict Orthogonality | 34.37 | 55.73 | 62.8% | 33.5% |
| **MSRS (Ours)** | **34.91** | **56.32** | **64.5%** | **36.1%** |

**Discussion** As shown in Table 20, enforcing strict pairwise orthogonality leads to a consistent performance drop compared to MSRS. We attribute this to the trade-off between *isolation* and *expressivity*:

1. **Semantic Over-Pruning:** Attributes in natural language are rarely disjoint. For instance, "Truthfulness" and "Instruction Following" share foundational semantic structures, such as *logical consistency* and *factuality*. In a sequential orthogonalization setting, if $B_{\text{Truth}}$ is extracted first, forcing the subsequent $B_{\text{Instruct}}$ to be strictly orthogonal effectively deletes

these shared cognitive primitives from the representation of instruction following. This "starves" the subspace of necessary semantic information, rendering the model less capable of executing complex instructions that require reasoning.

2. **Steering Robustness:** The results confirm that maintaining the full expressive capacity of $B_i$ even at the cost of slight non-orthogonality with $B_j$ is reasonable for effective steering. The strict constraint overly restricts the optimization landscape for $R_i$, resulting in suboptimal alignment across all metrics: we observe lower accuracy on TruthfulQA, reduced capability to mitigate bias, and a significant drop in complex generation quality on AlpacaEval. This indicates that strict orthogonality compromises the model's ability to handle multi-faceted tasks.

**Conclusion.** We conclude that while $B_{\text{shared}}$ removal is essential for disentanglement, enforcing further strict orthogonality among private subspaces is detrimental. MSRS effectively identifies the "sweet spot" where attributes are sufficiently disentangled to avoid conflict but sufficiently rich to ensure high-performance steering.

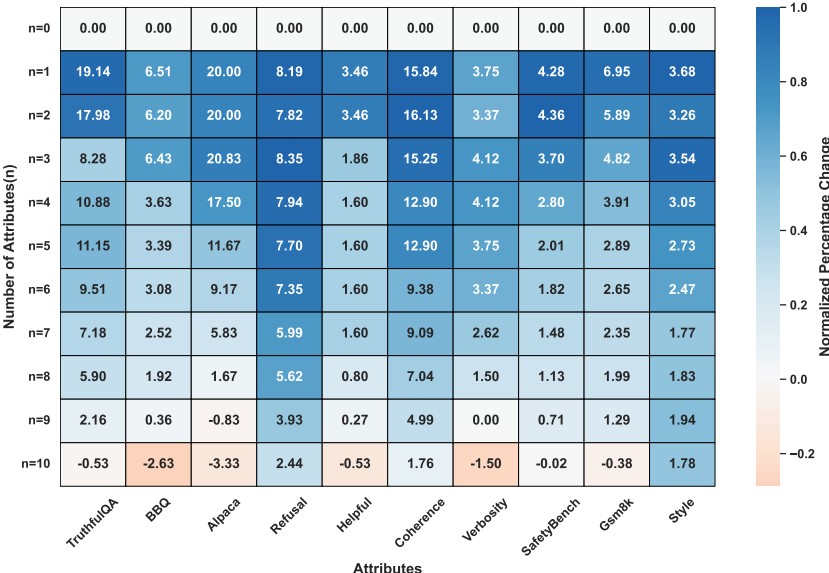

Figure 10: Normalized Relative Percentage Change ($\Delta\%$) in Different Attributes Compared to Baseline (n = 0).