# OpenReview forum: "MSRS: Adaptive Multi-Subspace Representation Steering for Attribute Alignment in Large Language Models"
_ICLR.cc/2026/Conference — Submitted to ICLR 2026_

### Official Review · Reviewer_ZsuH · 2025-10-29

**Soundness:** 3
**Presentation:** 3
**Contribution:** 3
**Rating:** 6
**Confidence:** 3

**Summary:**

This paper introduces Multi-Subspace Representation Steering (MSRS), a novel framework designed to address the challenge of simultaneously controlling multiple attributes in LLMs without the typical interference and performance trade-offs. Existing methods often struggle when steering for conflicting objectives like truthfulness and bias. MSRS overcomes this by decomposing the model's internal representation space into orthogonal subspaces: a shared subspace to capture common steering directions and multiple attribute-specific subspaces to isolate control. A key contribution is the use of SVD to adaptively determine the dimensionality of each subspace based on its expressive needs, rather than using fixed-size partitions. Furthermore, MSRS introduces a dynamic token intervention mechanism that identifies and applies steering to the most semantically relevant tokens for each attribute, enabling more precise, fine-grained control. Through comprehensive experiments, the authors demonstrate that MSRS surpasses strong baselines, effectively mitigating attribute conflicts and achieving superior performance across diverse models and tasks while preserving the model's general capabilities.

**Strengths:**

- Introduces a novel hybrid architecture that disentangles steering vectors into a shared subspace for common features and orthogonal, attribute-specific subspaces.
- The porposed method use of SVD to adaptively allocate subspace dimensionality based on the captured energy of activation differences. This is an advance over earlier approaches that relied on equal-sized partitions.
- The method demonstrates ability to navigate attribute trade-offs, such as concurrently improving scores on TruthfulQA and BBQ, a task where baselines often sacrifice performance on one metric to gain on the other.

**Weaknesses:**

- If I understand it correctly, the scalability of the MSRS framework is not tested beyond attribute pairs. With a fixed total rank R, it is unclear if the method can effectively manage the trade-offs when steering a larger number of attributes, as the capacity of each subspace would necessarily shrink.
- The 60% energy threshold for defining the shared subspace is presented as a fixed hyperparameter without a sensitivity analysis. This value is critical for balancing shared and specific control, and its robustness across different models and attribute combinations is not validated.
- The paper lacks a qualitative analysis to provide insight into the semantic nature of the learned subspaces. While demonstrating good quantitative results, it misses what features are being isolated.

**Questions:**

1. How does MSRS perform when steering more attributes simultaneously (e.g., truthfulness, harmlessness, and verbosity)? Given a fixed total rank R, does the decreasing capacity available for each attribute-specific subspace lead to a significant performance drop-off, and if so, at what point?
2. Could you provide a direct comparison of the inference latency of MSRS against the fixed-position baseline, ReFT? This would clarify the practical computational cost introduced by the dynamic token selection mechanism.
3. For steering baseline, have you considered recent works like DoLA [1] and SEA [2]? You should at least mention these works in your related work.


[1] Chuang et al. (2023) DoLa: Decoding by contrasting layers improves factuality in large language models

[2] Qiu et al. (2024) Spectral Editing of Activations for Large Language Model Alignment

---

> ### Author Response · Authors · 2025-11-18
>
> > 3. The paper lacks a qualitative analysis to provide insight into the semantic nature of the learned subspaces. While demonstrating good quantitative results, it misses what features are being isolated.
>
> We thank the reviewer for acknowledging the quantitative analysis in our paper. As discussed in **Appendix H**, we demonstrated **through linear probing that the private subspaces capture attribute-specific semantics, while the shared subspace retains generic linguistic knowledge**.
>
> In response to the reviewer’s concern about the lack of qualitative analysis, we have included additional qualitative insights in **Appendix I**. To further elucidate the semantic nature of the learned subspaces, we performed a targeted case study on representative instances from the Truthful, Bias, Instruction-Following, and Refusal attributes. For each attribute, we examined both the projections onto the private and shared directions. The results, shown in Table16, confirm that **the private subspaces reliably capture attribute-specific semantics, while the shared subspaces exhibit near-zero projections, indicating they do not contribute to attribute-specific discrimination**. This qualitative analysis strengthens our findings from the quantitative analysis in Appendix H, **providing a clearer understanding of how the learned subspaces correspond to meaningful semantic features**.
>
> We hope this additional qualitative analysis helps address your concern and provides more insight into the learned representations
>
> > 4. Could you provide a direct comparison of the inference latency of MSRS against the fixed-position baseline, ReFT? This would clarify the practical computational cost introduced by the dynamic token selection mechanism.
>
> Thank you for your valuable feedback. In response to the reviewer’s request for a direct comparison of the inference latency between MSRS (with dynamic token selection) and the fixed-position baseline (ReFT), we conducted experiments with a sequence length of 512 and batch size of 1 on the Truthfulqa dataset, using LLaMA2-7B.
>
> Our results show that the dynamic token selection mechanism introduces a slight increase in latency compared to the fixed-position method, with a latency increase of (+0.63 ms/token). Additionally, throughput decreases slightly by (-0.05 tokens/s). However, these results also indicate that while the **dynamic token selection mechanism introduces a small computational cost, it leads to a significant performance improvement in most attributes** (e.g., Truthful +4.2%, Bias +1.3% as shown in Table4 (line 392)).
>
> | Method                  | Latency (ms/token) ↓ | Throughput (tokens/s) ↑ |
> | ----------------------- | :------------------: | ----------------------- |
> | fixed-position          |        119.72        | 8.35                    |
> | dynamic token selection |        120.35        | 8.30                    |
>
> > 5. For steering baseline, have you considered recent works like DoLA [1] and SEA [2]? You should at least mention these works in your related work.
>
> Thank you for your suggestion. We agree that these works are relevant, especially in the context of inference-time optimization.
>
> **We will add the following paragraph to the Related Work section.**
>
> Recent works have focused on optimizing inference-time processing in large language models. DoLA adjusts logits during inference, while our method optimizes at the representation level, offering broader inference-time applicability. SEA mitigates issues like bias and untruthfulness by editing internal representations to align with positive examples. Unlike SEA, our approach focuses on representation-level adjustments, offering a complementary strategy for inference-time optimization.

---

> ### Author Response · Authors · 2025-11-18
>
> > 2. The 60% energy threshold for defining the shared subspace is presented as a fixed hyperparameter without a sensitivity analysis. This value is critical for balancing shared and specific control, and its robustness across different models and attribute combinations is not validated.
>
> We thank the reviewer for the constructive feedback. In lines 374-391, we discussed the energy threshold, where we concluded from **Figure 3** that the optimal shared subspace ratio lies between 20% and 40%, which corresponds to an energy range of approximately 50%-70%. In response to the reviewer's suggestion, we report **a sensitivity analysis of this threshold across different models** (Llama3-8B-Instruct, Qwen2-7B-Instruct) and attribute sets (e.g., "truthful & bbq," "alpaca & refusal," and "helpful & coherence & verbosity") in **Appendix G (Table 14, Line 1254)**.
>
> We agree that the choice of the 60% energy threshold is important. Thus, we conducted a sensitivity analysis by varying the shared-energy threshold across {40%, 50%, 60%, 70%, 80%} for different models and attribute combinations. Our empirical findings are summarized as follows:
>
> - **Very low thresholds** (e.g., 40%) allocate too little capacity to the shared subspace, which prevents effective combination of steering capabilities across different attributes, negatively affecting overall performance in multi-attribute steering.
> - **Very high thresholds** (e.g., 70–80%) allocate too much capacity to the shared subspace, which correspondingly shrinks the specific subspaces. This reduces the effective steering capacity available for each individual attribute and leads to weaker control in the multi-attribute setting.
> - **Thresholds in the 50–70%** yield consistently strong performance, with **60%** lying at the center of this "high-performance" This value strikes a good balance between retaining shared knowledge and preserving attribute specificity.

---

> ### Author Response · Authors · 2025-11-18
>
> > 1. How does MSRS perform when steering more attributes simultaneously (e.g., truthfulness, harmlessness, and verbosity)? Given a fixed total rank R, does the decreasing capacity available for each attribute-specific subspace lead to a significant performance drop-off, and if so, at what point? **&&** (**weaknesses1**): If I understand it correctly,…… as the capacity of each subspace would necessarily shrink.
>
> We thank the reviewer for this important point. In fact, **rank R (denoted as k in the paper) is a hyperparameter** that we explicitly adjust according to the number of attributes. When the number of attribute pairs is small, we use a small fixed R (as reported in the paper, fewer attributes correspond to rank k=8). When exploring a larger number of attributes, we **increase the total rank R accordingly** to accommodate more attributes.
>
> **We have uploaded the revised manuscript for more attributes steering**.
>
>  Starting from **line 480**, we systematically **investigate the performance of joint training with varying numbers of attributes ($n=0$ to $10$)**. The full procedure is detailed in **lines 480–507**. To make this clear, we provide a concrete example to explain the performance of attribute $i$ after joint training with $n$ attributes. (e.g., for attribute $a_i$ = TruthfulQA at $n=3$) :
>
> 1. Selecting Subset: We randomly select 2 attribute from the remaining 9 attributes (excluding truthfulqa) and  truthfulqa to form a 3-joint subset. For example, if we select bbq and alpaca, the subset is $ S_{\text{truthfulqa}, 3}^1 = \{\text{truthfulqa}, \text{bbq},\text{alpaca}\} $ and if we select refusal and helpful,  the subset is$S_{\text{truthfulqa}, 3}^2 = \{\text{truthfulqa}, \text{refusal},\text{helpful}\}$
>
> 2. Training and Performance Calculation: For each random subset $S_{m}^{\text{truthfulqa},3}$, we train the model and compute the performance $\text{Perf}^{\text{truthfulqa}}(S_{m}^{\text{truthfulqa},3})$ for $m=1,\dots,M$.
>
> 3. Average Performance: The average performance for $n=3$ is calculated as
> $$
> \text{Perf}^{\text{truthfulqa}}(n=3)
> = \frac{1}{M} \sum_{m=1}^{M} \text{Perf}^{\text{truthfulqa}}(S_{m}^{\text{truthfulqa},3})
> $$
> The results in **Figure 6a** show that **all columns exhibit performance improvements over the baseline (n=0) when n<9 (blue)**, with degradation only afterward (red).
>
> Starting from **line 518**, we discuss the **relationship between the number of attributes n and the rank R (k)**(red points and line in Figure 6b). As n increases, we can extend to more attributes by increasing k. Meanwhile, when the number of attributes n<9, the **average performance across all attributes remains higher than the baseline** (the green curve in Figure 6b stays above the green dashed baseline for n<9), demonstrating that our method can **effectively combine 8 attributes while achieving consistent gains on every attribute**.
>
> We also **defined a quantitative calculation method for interference** between attributes and found that as the number of attributes $n$ increases, the interference grows in a cubic function manner with respect to $n$. In **lines 518–532**, we further establish the **upper bound of performance gains** and identify the **optimal settings** that yield the best performance (**n=2 or 3** with **k=8 to 12**).

---

> > ### Author Response · Authors · 2025-11-22
> > **Reply to Reviewer ZsuH**
> >
> > Dear Reviewer,
> >
> > We kindly ask you to confirm whether we have sufficiently addressed your comments or if there are any remaining concerns.
> >
> > Thank you!
> >
> > Authors

---

> > > ### Comment · Reviewer_ZsuH · 2025-11-24
> > >
> > > Thank you for taking into account my suggestion on the reorganization of subsections. For now, my comments have been addressed and I don't have any remaining concerns.

---

> > ### Comment · Reviewer_ZsuH · 2025-11-23
> >
> > Thank you authors for performing this experiment. The results are nice additions to the paper. One more comment is that currently _Section 5.3 Ablation Study_ is very lengthy with all different kinds of ablations. I would suggest dividing it into subsections to make the section easier to follow.

---

> ### Author Response · Authors · 2025-11-24
> **Reply to Reviewer ZsuH**
>
> We sincerely thank the reviewer for the positive feedback on the new experiments and for the very helpful structural suggestion.
>
> To improve readability and better highlight the logical flow of our ablation studies, we have reorganized Section 5.3 (now Section 5.3 in the revised manuscript) into three clearly separated subsections:
>
> - **5.3.1 Core Multi-Subspace Design**
>   (validating the shared subspace and adaptive weighting network)
> - **5.3.2 Enhanced Steering Mechanisms**
>   (dynamic token selection, R-similarity analysis, optimal layer selection, and SVD-guided initialization)
> - **5.3.3 Scalability, Interference, and Rank Dynamics with Increasing Number of Attributes**
>   (detailed analysis of performance, interference, and effective rank evolution as the number of steered attributes grows)
>
> We believe this new structure makes the ablation section significantly easier to follow while preserving all original results and figures.
>
> We are happy to make any further revisions or provide clarifications if needed, and we truly appreciate the reviewer's helpful suggestions, which have undoubtedly improved the clarity of our work.

---

### Official Review · Reviewer_mx1V · 2025-11-01

**Soundness:** 2
**Presentation:** 3
**Contribution:** 3
**Rating:** 4
**Confidence:** 3

**Summary:**

The paper proposes MSRS (Multi-Subspace Representation Steering) for multi-attribute control of LLMs. Building on representation fine-tuning (ReFT), MSRS: (i) splits a low-rank representation space into attribute-specific and a shared subspace; (ii) uses SVD of attribute activation differences to size/initialize subspaces; (iii) adds alignment losses to keep the learned subspaces close to the SVD priors; and (iv) introduces dynamic token-position selection based on similarity to the learned subspace for inference-time interventions. Experiments across LLaMA-2/3-8B, Qwen2-7B, and Mistral-7B show consistent gains on TruthfulQA, BBQ, Alpaca, Refusal/Sorry-Bench, HelpSteer, and GLUE, with ablations on rank, shared-vs-private ratio, layer choice, and token selection.

**Strengths:**

1. The method is novel, well motivated and modular.
2. Dynamic token routing is simple and interesting, consistent gains over last-token steering.
3. Comprehensive experiments across several base models and tasks, statistically reported and supported by ablations.

**Weaknesses:**

1. As I understand, the size of each attribute subspace is decided by the number of top singular vectors. However, in line 202, $H_{res} = \tau_i - B_{shared}^{\top} B_{shared} \tau_i$ which should have the same dimension as $\tau_i$, i.e, $\mathbb{R}^d$. The authors's performing SVD on $H_{res}$ is not making sense to me since this is just a single vector. Therefore, no adaptive size of subspace can be deduced.
2. Though not clearly stated, I am assumming that $r = r_s + \sum_{i=1}^nr_i$, where $r$ is the size of $R$. How did the authors ensure this equality when constructing $B_i$'s and $B_{shared}$? Moreover, if my assumption is true, does that mean $r > n$ no matter how large $n$ is?
3. In line 242, the authors introduce a alignment loss:
$$
\mathcal{L}\_{align} = 1 - \dfrac{\langle R, S_{align} \rangle}{1- \Vert R\Vert\_2 \Vert S\_{align} \Vert\_2}
$$

how is the matrix inner product computed?

4. MSRS adds projection(s), mask MLP, and token selection. You reference compute cost in Appendix F; please summarize throughput/latency overhead and VRAM vs. ReFT and CAA at inference for typical sequence lengths.
5. Multi-attribute sets are evaluated in pairs or small sets. How does MSRS scale when n (attributes) grows in terms of rank budgeting, mask sparsity, and interference? A scaling-law-style study would strengthen claims.

**Questions:**

Refer to the weakness

---

> ### Author Response · Authors · 2025-11-18
> **Response to Reviewer mx1V**
>
> > 1. As I understand, the size of each attribute subspace is decided by the number of top singular vectors. However, in line 202,$ H_{\text{res}}^{(i)} = \tau_i - B_{\text{shared}}^\top B_{\text{shared}} \tau_i $ which should have the same dimension as $ \tau_i $,i.e,The authors's performing SVD on $ H_{\text{res}}^{(i)}$ is not making sense to me since this is just a single vector. Therefore, no adaptive size of subspace can be deduced.
>
> We thank the reviewer for the constructive feedback. We acknowledge this as a **notational mistake in our manuscript** and appreciate the opportunity to clarify.
>
> In our implementation, $ H_i  $ was intended to represent the activation matrix for attribute $ i $ (rather than $ \tau_i $), formed by concatenating the representations of all samples $ j $ for that attribute, i.e., $ H_i \in \mathbb{R}^{j \times d} $, where  $ d $ is the representation dimension. The residual $ H_{\text{res}}^{(i)} = H_i - B_{\text{shared}}^\top B_{\text{shared}} H_i $ is thus a matrix, capturing the attribute-specific directions after projecting out the shared subspace. Applying SVD to $ H_{\text{res}}^{(i)} $ extracts the most significant directions for attribute $ i $, enabling adaptive subspace allocation.
>
> **In the revised version (line197)**, we correct the notation by replacing $ \tau_i $ with $ H_i $ to denote the activation matrix for attribute $ i $. The residual computation will be updated from $ H_{\text{res}}^{(i)} = \tau_i - B_{\text{shared}}^\top B_{\text{shared}} \tau_i $ to $ H_{\text{res}}^{(i)} = H_i - B_{\text{shared}}^\top B_{\text{shared}} H_i $​​.

---

> ### Author Response · Authors · 2025-11-18
> **Response to Reviewer mx1V**
>
> > 3. In line 242, the authors introduce a alignment loss: $\mathcal{L}{\text{align}} = 1 - \frac{\langle R, S{\text{align}} \rangle}{ |R|2 |S{\text{align}}|_2}$ ,how is the matrix inner product computed?
>
> Thank you for your insightful comment. We find the **error in the original expression** for the alignment loss in Line 242. We have **made corresponding modifications in the revised version (line241).**
>
>  The correct formulation involves the normalized Frobenius inner product, not the $L_2$ norm as previously indicated. The intended alignment loss is:
> $\mathcal{L}{\text{align}} = 1 - \frac{\langle R, S{\text{align}} \rangle_F}{|R|F |S{\text{align}}|F}$
> Where $\langle A, B \rangle_F = \operatorname{Tr}(A^\top B) = \sum{i,j} A_{ij} B_{ij}$ is the Frobenius inner product, and $| \cdot |F$ denotes the Frobenius norm. This formulation is equivalent to $1 - \cos(R, S{\text{align}})$ and ensures that the rows of R align with the principal directions of variation for the target attribute. We will correct this in the manuscript accordingly.
>
> > 4. MSRS adds projection(s), mask MLP, and token selection. You reference compute cost in Appendix F; please summarize throughput/latency overhead and VRAM vs. ReFT and CAA at inference for typical sequence lengths.
>
> In addition to the computational cost reported in Appendix F for different values of rank $R$, we further tested the **computational cost of various methods** based on Llama3-8B-Instruct on the Alpaca dataset, with a sequence length of 512 and a batch size of 1. We have **add this results in the revised version appendix c.2 (line1080).**
>
> |   Method | Latency (ms/token) ↓ | Throughput (tokens/s) ↑ | VRAM (GB) |
> | -------: | :------------------: | ----------------------- | --------- |
> | Baseline |        102.44        | 9.76                    | 16.10     |
> |     ReFT |        115.72        | 8.64                    | 16.26     |
> |      CAA |        110.59        | 9.04                    | 16.12     |
> |     MSRS |        120.35        | 8.38                    | 16.29     |
>
> The results show that the MSRS method introduces a slight increase in computational cost compared to the Baseline and CAA methods.  However, the latency and throughput are similar to those of ReFT. **Despite this modest increase in computational overhead, MSRS achieves the best experimental results, as demonstrated in Table 1(line 270), indicating that the performance gains outweigh the small additional computational cost.**

---

> ### Author Response · Authors · 2025-11-24
> **Response to Reviewer mx1V**
>
> > 5. Multi-attribute sets are evaluated in pairs or small sets. How does MSRS scale when n (attributes) grows in terms of rank budgeting, mask sparsity, and interference? A scaling-law-style study would strengthen claims.
>
> **We have uploaded the revised manuscript for more attributes steering.**
>
>  Starting from **line 484**, we systematically **investigate the performance of joint training with varying numbers of attributes ($n=0$ to $10$)** ( **including attributes for truthfulness, bias, instruction following, refusal, helpfulness, coherence, verbosity, safety, reasoning, and style**). The full procedure is detailed in **lines 484–511**. To make this clear, we provide a concrete example to explain the performance of attribute $i$ after joint training with $n$ attributes. (e.g., for attribute $a_i$ = TruthfulQA at $n=3$) :
>
> 1. Selecting Subset: We randomly select 2 attribute from the remaining 9 attributes (excluding truthfulqa) and  truthfulqa to form a 3-joint subset. For example, if we select bbq and alpaca, the subset is $ S_{\text{truthfulqa}, 3}^1 = \{\text{truthfulqa}, \text{bbq},\text{alpaca}\} $ and if we select refusal and helpful,  the subset is$S_{\text{truthfulqa}, 3}^2 = \{\text{truthfulqa}, \text{refusal},\text{helpful}\}$
>
> 2. Training and Performance Calculation: For each random subset $S_{m}^{\text{truthfulqa},3}$, we train the model and compute the performance $\text{Perf}^{\text{truthfulqa}}(S_{m}^{\text{truthfulqa},3})$ for $m=1,\dots,M$.
>
> 3. Average Performance: The average performance for $n=3$ is calculated as
>
> $$
> \text{Perf}^{\text{truthfulqa}}(n=3)
> = \frac{1}{M} \sum_{m=1}^{M} \text{Perf}^{\text{truthfulqa}}(S_{m}^{\text{truthfulqa},3})
> $$
>
> The results in **Figure 6a** show that **all columns exhibit performance improvements over the baseline (n=0) when n<9 (blue)**, with degradation only afterward (red).（**described in lines 511-517**)
>
> Starting from **line 518**, we discuss the **relationship between the number of attributes n and the rank R (k)**(red points and line in Figure 6b). As n increases, we can extend to more attributes by increasing k. Meanwhile, when the number of attributes n<9, the **average performance across all attributes remains higher than the baseline** (the green curve in Figure 6b stays above the green dashed baseline for n<9), demonstrating that our method can **effectively combine 8 attributes while achieving consistent gains on every attribute**.
>
> We also **defined a quantitative calculation method for interference** between attributes and found that as the number of attributes $n$ increases, the interference grows in a cubic function manner with respect to $n$​. In **lines 518–533**, we further establish the **upper bound of performance gains** and identify the **optimal settings** that yield the best performance (**n=2 or 3** with **k=8 to 12**).
>
> > 2. Though not clearly stated, I am assumming that $r = r_s+\sum_{i=1}^{n} r_i$,where $ r$ is the size of $R$.How did the authors ensure this equality when constructing $B_i$'s and $B_{shared}$? Moreover, if my assumption is true, does that mean $r>n$ no matter how large $ n$ is?
>
> We thank the reviewer for this very insightful question.
>
> Yes, the reviewer’s assumption is correct: the total rank of the steering matrix \(R\) is exactly
> $$ r = r_{\text{shared}} + \sum_{i=1}^{n} r_i. $$Each of the $n$ attributes requires at least one unit of attribute-specific space, and with the addition of the shared space, this ensures that $r>n$.
>
> In the main experiments (Sections 5.1–5.2), we primarily evaluate joint steering of 2 or 3 attributes, which leads to roughly similar total ranks across different settings. **However, the total rank \(r\) is not fixed — it naturally increases with the number of attributes \(n\)**, consistent with the dimensionality of $S_{\text{align}}$ defined in Equation (2) (Line 211).
>
> Our **additional scalability experiments (reported around Line 519 and Figure 6b) further clarify this behavior**: $r$ does grow with $n$ (since each attribute contributes at least a small $r_i \geq 1$). Nevertheless, the per-attribute rank $r_i$ remains very small in practice (typically 1–3). As a result, even when steering $n=10$ attributes simultaneously, the total rank increases only modestly ($r \approx 24$ on average) — a value that still falls comfortably within the favorable rank range ($<32$) identified as optimal in ReFT[1].
>
> We believe that this additional experiment clarification will help readers better understand the rank allocation section.
>
> [1] Wu Z, Arora A, Wang Z, et al. Reft: Representation finetuning for language models[J]. Advances in Neural Information Processing Systems, 2024, 37: 63908-63962.

---

> > ### Comment · Reviewer_mx1V · 2025-11-25
> >
> > Thanks to the authors for the detailed clarification and explanation.
> >
> > I have one remaining question regarding the formulation of (B^{(i)}): why is orthogonality imposed only to the $B_{shared}$, but not among the $B^{(i)}_{res}$ themselves? Would this choice limit the representation ability of the $B^{(i)}\_{res}$, since there is no mechanism ensuring that different $B^{(i)}\_{res}$ are not overlapping to a certain extend? If so, could this suggest that the performance peak in Fig. 6(a) does not occur at large ($n \geq 5$) or ($6$), but rather around ($n = 2$) or ($3$), even when testing with a large number of attributes?
> >
> > In addition, I share reviewer M4SR’s concern that the change in the notation of $H^{(i)}\_{res}$ may reflect a deeper issue than a simple typographical error.

---

> ### Author Response · Authors · 2025-11-27
> **Response to Reviewer mx1V**
>
> > 6. I have one remaining question regarding the formulation of (B^{(i)}): why is orthogonality imposed only to the $B_{shared}$,but not among the $B_{\text{res}}^{(i)}$ themselves?
> >
> > 7. Would this choice limit the representation ability of the representation ability of the $B_{\text{res}}^{(i)}$， since there is no mechanism ensuring that different $B_{\text{res}}^{(i)}$ are not overlapping to a certain extend?
>
> We believe that your concerns are in alignment with those of M4SR. First, we confirm that $B_{\text{res}}^{(i)}$ and $B_{\text{res}}^{(j)}$ are not strictly orthogonal to each other.
>
> We discuss this issue in **Appendix L (line 1506)**
>
> * From Mechanism 1 (line 1556): Although $B_i$ contains residual directions overlapping with $B_j$​, these directions correspond to the tail end of the spectral energy distribution, so **their impact on the final performance is minimal**.
> * From Mechanism 2 (line 1570): Through $m(h)$, it not only activates strongly in the target attribute but also in attributes that overlap significantly with the target attribute. This **allows the recruitment of beneficial auxiliary signals from correlated subspaces to refine the representation**, while ensuring the steering is structurally dominated by the target subspace to prevent semantic drift.
>
> Additionally, in further **Appendix N (line 1688)**, we analyze the **impact of restricting orthogonality versus not restricting it (ours)** on performance based on our setup. We find that **restricting orthogonality leads to a degradation in the expressive power of specific attribute subspaces**, resulting in final outcomes that are even worse than when orthogonality is not restricted.
>
> > 8. If so, could this suggest that the performance peak in Fig. 6(a) does not occur at large $n \geq 5$ or $(6)$ , but rather around $(n=2)$ or $(3)$, even when testing with a large number of attributes?
>
> First, we would like to **correct your mistakes**: the performance peak in Fig. 6(a) **does occur at large** \( n =2 \) or \( (3) \) , rather than at 5 or 6 as you mentioned. This may be due to a **misinterpretation of the figure**.
>
> In Fig. 6(a), our peak appears between $n = 2$ to $4$, and between $n = 5$ to $8$, the gains weaken (lighter blue), reflecting uneven benefits. This is due to the lack of strict orthogonality between different attributes, leading to increased interference between $B_{\text{res}}^{(i)}$ and $B_{\text{res}}^{(j)}$. However, as previously discussed, the interference between them corresponds to the tail end of the spectral energy distribution, so **their impact on the final performance is minimal**.
>
> As $n$ increases, in **Appendix M (line 1589)**, in light of your concerns and those raised by M4SR, we define a metric to measure unintended effects on *untargeted attributes*, and discuss how unintended effects vary with $n$.
>
> The unintended effects on *untargeted attributes* increase with the number of attributes, and we observe similar results to the analysis in **Section 5.3.3 (line 483)**. Specifically, when $n < 9$, the effects on untargeted attributes grow slowly, but the growth rate increases significantly after that.
>
> > 9. In addition, I share reviewer M4SR’s concern that the change in the notation of $H_{\text{res}}^{(i)}$  may reflect a deeper issue than a simple typographical error.
>
> We confirm that the symbol error in our previous writing misled the two reviewers, causing them to believe that our approach was unreasonable. However, **after correcting the symbol error, we believe that when you review lines 182 to 211, you will find that, with this correction, our algorithm still aligns with the original intent of the paper and has not diverged from it**.
>
> **This modification has already been acknowledged by M4SR.**
>
> **In the revised version, we have clarified lines 196-199**.  Our implementation should be:
> $$
> H_{\text{res}}^{(i)} = \left[ h_{i,1}, h_{i,2}, \ldots, h_{i,|D_i|} \right] - B_{\text{shared}}^\top B_{\text{shared}} \left[ h_{i,1}, h_{i,2}, \ldots, h_{i,|D_i|} \right]
> $$
>
> instead of:
>
> $$
> H_{\text{res}}^{(i)} = \left[ h_{i,1}, h_{i,2}, \ldots, h_{i,|D_i|} \right] - B_{\text{shared}}^\top B_{\text{shared}} \tau_i
> $$
>
> Since directly concatenating all attributes' activations of all samples results in a matrix with too large dimensions, our **shared subspace** is obtained by first computing the average activation for each attribute and then concatenating these averages:
>
> $$
> \tau_c = [\tau_1 \; | \; \tau_2 \; | \; \dots \; | \; \tau_n] \in \mathbb{R}^{d \times n}
> $$
>
> We then perform SVD on this concatenated vector. However, for the **attribute-specific subspace**, we concatenate the activations of all samples for that attribute (instead of using the average) and perform SVD to obtain it.
>
> **This is how our implementation works**. Indeed, we made this symbol error in our previous writing, which may have led to a misunderstanding of the algorithm by the reviewer.

---

> > ### Author Response · Authors · 2025-11-28
> > **Response to Reviewer mx1V**
> >
> > Dear Reviewer,
> >
> > We greatly appreciate your insightful suggestions and have made the necessary revisions accordingly.
> >
> > Although there were some system issues, we would still appreciate knowing whether these changes have addressed your concerns and improved the quality of our manuscript.
> >
> > We look forward to your feedback.
> >
> > Thank you!
> >
> > Sincerely,
> >
> > The Authors

---

### Official Review · Reviewer_M4SR · 2025-11-01

**Soundness:** 2
**Presentation:** 3
**Contribution:** 3
**Rating:** 4
**Confidence:** 3

**Summary:**

The paper proposes Multi-Subspace Representation Steering (MSRS), a novel framework for multi-attribute control in large language models by allocating orthogonal subspaces for each attribute and a shared subspace for common steering directions. This design mitigates interference between attributes and enables more precise behavior modulation. MSRS also introduces a dynamic token-level steering mechanism during inference, selecting semantically relevant tokens for intervention. Experiments across multiple models (LLaMA2, LLaMA3, Qwen2, Mistral) and tasks (multiple-choice, open-ended generation) show that MSRS significantly reduces attribute conflicts and outperforms existing steering methods, while also generalizing effectively to standard NLP benchmarks such as HellaSwag and GLUE.

**Strengths:**

- Addresses a practical challenge in activation steering: controlling multiple attributes with minimal cross-attribute interference.
- Introduces a novel decomposition into a shared subspace and multiple attribute-specific subspaces.
- Covers an extensive set of datasets, metrics, models, and baseline methods.
- Demonstrates strong results across multiple benchmarks and model families.
- Includes robustness evaluation showing that general capabilities remain unaffected under steering.
- Provides ablation studies validating the role of the shared subspace in integrating multi-attribute features.
- Clearly written, with intuitive visualizations and easy-to-follow presentation.

**Weaknesses:**

See Questions

**Questions:**

- Line 158: "ReFT assumes a single attribute per input"—but since R is r-dimensional, shouldn't it be capable of representing multiple attributes?
- Lines 195–199: Why does selecting more top vectors for the shared directions allow adaptive "subspace sizes for each attribute based on its expressive needs"? Wouldn't allocating more space to one attribute reduce the shared subspace size?
- Lines 201–204: Since $\tau_i$ is a vector, shouldn't $H^{(i)}_\text{res}$ also be a vector?
- Equation 2: The formulation of $B_i$ implies that $B_i$ and $B_j$ can overlap via shared directions. Wouldn't this bias the alignment of $R$ toward shared directions in $S_\text{align}$?
- The concept of dynamic intervention has been explored in prior work (e.g., [1, 2]). The authors should discuss connections and distinctions.
- Line 425–427: Does "last token" mean steering is applied only to the last token of the input? Please clarify.

[1] Programming Refusal with Conditional Activation Steering
[2] Angular Steering: Behavior Control via Rotation in Activation Space

---

> ### Author Response · Authors · 2025-11-18
>
> > 1. Line 158: "ReFT assumes a single attribute per input"—but since R is r-dimensional, shouldn't it be capable of representing multiple attributes?
>
> Thank you for pointing out the **unclear expressions**. In the original ReFT, $R$ is $r$-dimensional. By fine-tuning on a dataset for a single target attribute within the low-rank subspace jointly spanned by these $r$ ranks,the $r$ ranks collectively express the fine-grained steering directions of that single attribute, rather than multiple attributes. Since the fine-tuning in ReFT is performed on a dataset specific to a single attribute, it cannot explicitly acquire the capability to steer multiple attributes through this dataset alone.
>
>  Our contribution is to partition the space for different attributes, so that each attribute’s steering directions are fine-tuned in mutually isolated subspaces, thereby reducing interference between different attribute steerings and achieving simultaneous performance improvements across multiple attributes.
>
> > 2. Lines 195–199: Why does selecting more top vectors for the shared directions allow adaptive "subspace sizes for each attribute based on its expressive needs"?
>
> This is a **paragraph organization issue**. Our statement in Lines 195–199 — “By selecting the top singular vectors, we capture high-variance directions that represent the most expressive steering dimensions. It allows us to automatically allocate varying subspace sizes for each attribute based on its expressive...” was **intended to describe the attribute-specific subspace, not the shared one**. Specifically, for each attribute, we select its top singular vectors as the basis for determining the subspace size, which adaptively allocates the subspace according to the expressive capacity required by that attribute.
>
> However, **our current placement of this statement right after the explanation of the shared subspace misleads reviewer to think that the adaptive allocation arises from the selection of shared directions**.  In the revised version, we will move the content in Lines 195–199 to appear before Line 206 (“This”), clarifying that the adaptive subspace allocation refers to the attribute-specific subspaces rather than the shared subspace.
>
> > 3. Wouldn't allocating more space to one attribute reduce the shared subspace size?
>
> We sincerely thank the reviewer for this excellent question.
>
> When the number of attributes $n$ is fixed, the total subspace dimensionality $r$ is indeed approximately the same across different attribute combinations (since the per-attribute rank $r_i$ remains very small in practice, typically 1–3). Therefore, under a fixed $n$ with fixed subspace dimensionality $r$, allocating more space to one attribute does reduce the size of the shared subspace.
>
> **However, across different values of $n$ (i.e., when jointly steering a different number of attributes), the total subspace dimensionality $r$ is not fixed**. Instead, it is exactly the sum of the independently determined dimensions:
> $$
> r = r_{\text{shared}} + \sum_{i=1}^{n} r_i
> $$
> Consequently, the total rank $r$ naturally (and moderately) increases with the number of attributes $n$.
> To directly address this point, we **have added a new subsection in the revised manuscript**:
> **5.3.3 Scalability and Interference Patterns with Growing Attribute Count** (Lines 484–533), in which:
> - Lines 519–533 and Figure 6b (red curve) explicitly show how the total rank $r$ grows with $n$,
> - Lines 484–517 demonstrate that MSRS maintains consistent per-attribute performance gains up to $n=9$.
>
> We hope these clarifications and new analyses fully resolve the reviewer’s concern and further illustrate the flexibility and scalability of our adaptive rank allocation strategy.
>
> > 4. Lines 201–204: Since $\tau_i $ is a vector, shouldn't $ H_{\text{res}}^{(i)}$ also be a vector?
>
> We acknowledge this as a **notational mistake** in our manuscript and appreciate the opportunity to clarify.
>
> In our implementation, $ H_i  $ was intended to represent the activation matrix for attribute $ i $ (rather than $ \tau_i $), formed by concatenating the representations of all samples $ j $ for that attribute, i.e., $ H_i \in \mathbb{R}^{j \times d} $, where  $ d $ is the representation dimension. The residual $ H_{\text{res}}^{(i)} = H_i - B_{\text{shared}}^\top B_{\text{shared}} H_i $ is thus a matrix, capturing the attribute-specific directions after projecting out the shared subspace. Applying SVD to $ H_{\text{res}}^{(i)} $ extracts the most significant directions for attribute $ i $, enabling adaptive subspace allocation.
>
> In the revised version, we will correct the notation by replacing $ \tau_i $ with $ H_i $ to denote the activation matrix for attribute $ i $. The residual computation will be updated from $ H_{\text{res}}^{(i)} = \tau_i - B_{\text{shared}}^\top B_{\text{shared}} \tau_i $ to $ H_{\text{res}}^{(i)} = H_i - B_{\text{shared}}^\top B_{\text{shared}} H_i $.

---

> ### Author Response · Authors · 2025-11-18
>
> > 5. Equation 2: The formulation $ B_i $ of implies that  $ B_i $ and  $ B_j $  can overlap via shared directions. Wouldn't this bias the alignment of $R$  toward shared directions in $S_{align}$​ ?
>
> Thank you for your insightful comment. In Line 211, Equation 2 describes $B_{\text{shared}}$, which captures the shared feature directions between the representations of each attribute, such as $\tau_i$ and $\tau_j$ (as defined in Line 188). The matrices $B_i$ and $B_j$, on the other hand, capture attribute-specific directions derived from the representations $H_{\text{res}}^{(i)}$ and $H_{\text{res}}^{(j)}$ (which we clarified earlier). Since $H_{\text{res}}^{(i)} = H_i - B_{\text{shared}}^\top B_{\text{shared}} H_i$, this implies that both $H_{\text{res}}^{(i)}$ and $H_{\text{res}}^{(j)}$ have already removed the shared components projected onto $B_{\text{shared}}$. Consequently, the resulting $B_i = \left( V^{(i)}_{1:r_i} \right)$ and $B_j$ do not overlap through shared directions.
>
> We hope this explanation addresses your concern. If this interpretation does not fully capture your question, we would be happy to provide further clarification.
>
> > 6. The concept of dynamic intervention has been explored in prior work (e.g., [1, 2]). The authors should discuss connections and distinctions.
>
> Thank you for your insightful comment. We acknowledge that dynamic intervention has been explored in prior work, specifically in Conditional Activation Steering [1] and Angular Steering [2]. Below, we briefly **discuss the connections and distinctions between these works and our approach**.
>
> **Connections**:
>
> 1. **Similar Effectiveness**: Both methods and ours adaptively select the target features for steering, improving performance while minimizing the influence on non-target features.
> 2. **Similar Monitoring Approach**: Both methods and ours compute the similarity between the representation \(h\) and the steering direction, using this similarity as a gating mechanism to decide whether steering is needed.
>
> **Distinctions**:
>
> * **Alignment Goals**: Although both methods align with the steering direction to decide whether to apply steering, in the multi-attribute scenario, our alignment goals for different attribute features are orthogonal to each other, whereas [1] and [2] may have overlapping alignment targets. Therefore, our method not only chooses whether to apply steering but also ensures that steering for different attributes does not interfere with each other (the main contribution).
>
> * **Scope of Discussion**: [1] and [2] consider a single attribute's target feature (e.g., refusal). However, in a multi-attribute setting, if the representation $h$ aligns well with the steering directions for multiple attributes, the adaptive selection approach needs to apply steering for multiple attributes at the same activation location. Directly stacking the steering of multiple attributes could lead to failed steering or even harm other properties, such as fluency. Our approach, however, uses the shared subspace to capture common directions across multiple attributes, enabling better integration (the main contribution).
>
> * **Emphasis**: [1] makes adaptive selection the main contribution of their work, while in our case, like [2], dynamic intervention serves as an additional contribution.
>
> We consider [1] and [2] to be relevant works and will add them to the related work section in the revised version.
>
> > 7. Line 425–427: Does "last token" mean steering is applied only to the last token of the input? Please clarify.
>
>  Yes, the "last token" refers to a simplified experiment in our ablation study, which is designed to validate the effectiveness of our proposed dynamic intervention position selection mechanism. **To visually verify the method's impact, we simplified the experiment by fixing the number of steering tokens to 1 and discussing the effect of steering at different positions** (i.e., comparing the last token to the most effective token selected from the last few tokens for steering on attribute i). The results in Table 4 (line 393) demonstrate that our method achieves greater improvements across multiple attributes compared to steering only at the last token.
>
> In our main experiments, steering is applied dynamically to a subset of important tokens selected from the last few tokens (not just one), which is an improvement over the approach in the main work we reference (ReFT), where steering is applied to the last few tokens.

---

> > ### Author Response · Authors · 2025-11-22
> > **Reply to Reviewer M4SR**
> >
> > Dear Reviewer,
> >
> > We kindly ask you to confirm whether we have sufficiently addressed your comments or if there are any remaining concerns.
> >
> > Thank you!
> >
> > Authors

---

> ### Comment · Reviewer_M4SR · 2025-11-25
>
> Thank you for the revisions.
>
> ---
> I still find the claim in the **Shared and Specific Subspace Extraction** section a bit confusing. In particular, you state:
>
> > We adaptively select the smallest number $r_s$ such that the cumulative energy (sum of top $r_s$ singular values) accounts for at least 60% of the total energy in $\Sigma_c$. This yields the shared subspace of $V_c$, defined as $B_{shared} = V^\top_{c,1:r_s} \in R^{r_s×d}$. Intuitively, $B_{shared}$ captures the dominant shared directions across all attributes. By selecting the top singular vectors, we capture high-variance directions that represent the most expressive steering dimensions.
>
> As I understand it, this means that $B_{\text{shared}}$ has *adaptive rank* depending on how much variance is needed to meet the 60% threshold in $V_c$.
>
> However, you follow this with:
>
> > It allows us to automatically allocate varying subspace sizes for each attribute based on its expressive needs. For complex attributes, it selects more top vectors, while simpler attributes are allocated smaller subspaces, effectively addressing the mismatch between each attribute’s steering capacity and its allocated subspace size.
>
> This second statement seems to describe how *each attribute* gets an adaptive subspace size. But this seems more related to the construction of $B_i$, not $B_{\text{shared}}$. Would it make more sense to move this sentence into the paragraph describing the construction of $B_i$ for each attribute?
>
> At its current location, I don’t see how the construction of the *shared* subspace $B_{\text{shared}}$ relates directly to automatically varying subspace sizes *per attribute*. Please clarify if I’m missing something, and if not, consider adjusting the paragraph structure accordingly. Also, please highlight any such change in the manuscript.
>
> ---
>
> In light of the change from:
>
> $H^{(i)}_{\text{res}} = \frac{1}{|\mathcal{D}_i|} \sum^{|\mathcal{D}\_i|}\_{j=1} h^l\_{i,j} - B^\top\_{\text{shared}} B\_{\text{shared}} \tau_i$
>
> to:
>
> $H^{(i)}_{\text{res}} = [h^l\_{i,1}, h^l\_{i,2}, \cdots, h^l\_{i,|\mathcal{D}\_i|}] - B^\top\_{\text{shared}} B\_{\text{shared}} \tau_i$
>
> I would argue this change goes beyond a notational error, it significantly alters the proposed method. As originally written, the paper would be difficult or impossible to reproduce. Please reflect this change clearly in the revised manuscript and update the relevant explanations accordingly.
>
> ---
>
> After revisiting the revised paper, I have a few follow-up questions:
>
> You define $H_i \in \mathcal{R}^{|D_i| \times d}$, which for LLaMA2-7B (with $d = 4096$) and the Alpaca dataset (using 70% of 52,000 samples = 36,400) results in:
> $$
> H_i \in \mathbb{R}^{36400 \times 4096}
> $$
> This is quite large, and so is $H^{(i)}_{\text{res}}$. As such, computing SVDs for $B_i$ over these matrices seems computationally expensive. This cost and memory usage should be discussed explicitly in the paper.
>
> Could you please:
> - Provide a detailed computational complexity analysis of the method,
> - State the actual time each step took,
> - Indicate the hardware used, and
> - Note whether you used any optimization techniques to reduce computation.
>
> ---
>
> I’m continuing to review the updated manuscript and may have further questions later today.

---

> ### Author Response · Authors · 2025-11-25
> **Respose to Reviewer M4SR**
>
> Thank you for your insightful comment.
> > 8. I still find the claim in the Shared and Specific Subspace Extraction section a bit confusing……
>
> **Our new revised version has been modified with blue markings on line197.**
>
> we move the content in Lines 195–199 to appear before Line 206 (“This”), clarifying that the adaptive subspace allocation refers to the attribute-specific subspaces rather than the shared subspace.
>
> > 9. I would argue this change goes beyond a notational error, it significantly alters the proposed method. .....Please reflect this change clearly in the revised manuscript and update the relevant explanations accordingly.
>
> **In the revised version, we have clarified lines 196-199**. We **correct the misunderstanding in your interpretation**. Our implementation should be:
>
> $$
> H_{\text{res}}^{(i)} = \left[ h_{i,1}, h_{i,2}, \ldots, h_{i,|D_i|} \right] - B_{\text{shared}}^\top B_{\text{shared}} \left[ h_{i,1}, h_{i,2}, \ldots, h_{i,|D_i|} \right]
> $$
>
> instead of:
>
> $$
> H_{\text{res}}^{(i)} = \left[ h_{i,1}, h_{i,2}, \ldots, h_{i,|D_i|} \right] - B_{\text{shared}}^\top B_{\text{shared}} \tau_i
> $$
>
> Since directly concatenating all attributes' activations of all samples results in a matrix with too large dimensions, our **shared subspace** is obtained by first computing the average activation for each attribute and then concatenating these averages:
>
> $$
> \tau_c = [\tau_1 \; | \; \tau_2 \; | \; \dots \; | \; \tau_n] \in \mathbb{R}^{d \times n}
> $$
>
> We then perform SVD on this concatenated vector. However, for the **attribute-specific subspace**, we concatenate the activations of all samples for that attribute (instead of using the average) and perform SVD to obtain it.
>
> > 10. This cost and memory usage should be discussed explicitly in the paper.
>
> In fact, **to balance the data proportion of different attributes, we did not use the full dataset for larger datasets** (since other datasets don’t have as much data).
>
> * For larger datasets like Alpaca and refusal, we sampled 1,000 samples from the total dataset to balance the number of samples with other attribute datasets, ensuring that the shared space can equally capture the shared attribute direction from all the familiar datasets.
>
> * For larger datasets like BBQ, we sample uniformly from each subclass to ensure a balanced and unbiased distribution across the datasets, ultimately resulting in 1,000 samples.
>
> * For other attributes, we use the full dataset.
>
> We discuss the **detailed computational complexity analysis, actual time each step took**, and related details in the **revised version's Appendix C (line 1036)**.

---

> > ### Comment · Reviewer_M4SR · 2025-11-25
> >
> > I appreciate the authors' prompt response.
> >
> > Below are some more questions.
> >
> > ---
> >
> > In one of my previous questions, I mentioned:
> > > Equation 2: The formulation of $B_i$ implies that $B_i$ and $B_j$ can overlap via shared directions.
> >
> > Perhaps I didn't make my point clear, please allow me to clarify. According to your formulation $H^{(i)}\_\text{res}$ is orthogonal to $B_\text{shared}$, and so is $B_i$. However, there is no guarantee that $B_i$ and $B_j$ are orthogonal to each other. This raises several questions:
> >
> > - $B_i$ and $B_j$ may still have overlapping directions, albeit not as strong as the ones captured by $B_\text{shared}$. Thus, the combined basis $S_\text{align} = [B_\text{shared}, B_1, B_2, \cdots, B_n]$ might over-represent these weaker shared directions, as it might exist in multiple $B_i$. This, in turns, leads to $R$ over-aligned with these weaker shared directions via the optimization of $\mathcal{L}_\text{align}$.
> > 	- How might this affect performance?
> > 	- If that's not the case,  can the authors theoretically prove that $B_i \perp B_j$ for $i \ne j$.
> > 	- Otherwise, please theoretically and/or empirically evaluate the orthogonality between $B_i$ and $B_j$, e.g. by estimating the rank or overlap between $B_i$ and $B_j$ across all attribute pairs.
> >
> > - If $B_i$ and $B_j$ are not orthogonal,  is the following statement still valid?
> >   > the total subspace dimensionality is exactly $r = r_\text{shared} + \sum^n_{i=1} r_i$.
> >     - If yes, please provide a proof.
> >     - If no, please revise the statement and clearly note this limitation in the manuscript.
> >     Additionally, please analyze the relationship between $\text{rank}(S_\text{align})$, $\text{rank}(R)$ and $r$.
> >
> > It would be helpful to include a visualization comparing $m(h)$ and $m_\text{prior}$, to illustrate how well MSRS focuses on steering within attribute-relevant subspaces. This may also provide insight into whether $m(h)$ captures the aforementioned residual shared directions.
> >
> > ---
> > In Figure 6a, why does the performance peak at $n = 2$ or $3$ instead of $1$? While the degradation with increasing $n$ makes intuitive sense, it is unclear to me why  a steering single attribute yields inferior results, especially for `refusal` and `verbosity` which prior works have shown effectiveness.
> >
> > Given that multi-attribute steering is a core contribution of this paper, does this imply MSRS is not recommended for single-attribute use cases? Could the authors provide discussion or hypotheses for this outcome?
> >
> > ---
> > Can the authors evaluate the precision of MSRS, i.e., its unintended effects on *untargeted attributes* during multi-attribute steering? How does this precision vary as $n$ increases?
> >
> > ---
> > Minor: References `Nguyen, 2025a` and `Nguyen, 2025b` appear to be duplicates.

---

> ### Author Response · Authors · 2025-11-26
> **Respose to Reviewer M4SR**
>
> > 11. $B_i$ and $B_j$​ may still have overlapping directions, albeit not as strong as the ones captured……
> >
> > &&
> >
> > Otherwise, please theoretically and/or empirically evaluate the orthogonality between $B_i$ and $B_j$​​……
> >
> > &&
> >
> > It would be helpful to include a visualization comparing $m(h)$ and $m_{prior}$ to illustrate how well MSRS focuses on steering within attribute-relevant subspaces. This may also provide insight into whether  $m(h)$​​ captures the aforementioned residual shared directions.
>
> Thank you for acknowledging that we have addressed the initial issue, and for raising new questions.
>
>  First, we confirm that $B_i$ and $B_j$ are not strictly orthogonal to each other. We have proved through experiments that **non-orthogonality does not lead to over-align; instead, it can even achieve more flexible alignment optimization of multiple attributes.**
>
> In our revised version, we discuss this issue in **Appendix L (line 1506)**
>
> We first **compute the overlap** between $B_i$ and $B_j$ and plot it in Figure 7a, then visualize the **activation of $m(h)$** across different subspaces for each target attribute (Figure 7b).
>
> We then analyze this from two perspectives:
>
> 1. **Response to the reviewer's concern that this "leads to $R$ over-aligned with these weaker shared directions via the optimization of $L_{align}$":**
>    - **Mechanism 1: Mitigating Over-Alignment via Spectral Dominance**:Residual directions in $B_i$ do not disrupt the learning process, as they fall in the low-energy tail of the spectral distribution. The model optimizes alignment with high-variance directions, ensuring robust steering toward the target attribute.
>
> 2. **Response to the reviewer's question about whether "$m(h)$ captures the aforementioned residual shared directions":**
>
>    - **Mechanism 2: Adaptive Feature Recruitment via Semantic Resonance:** The model leverages residual overlap by adapting the mask to activate relevant features from $B_j$. However, a hierarchical control ensures that the primary steering direction remains aligned with $B_i$, preventing semantic drift.
>
> > 12. How might this affect performance?
>
> From Mechanism 1 (line 1556): Although $B_i$ contains residual directions overlapping with $B_j$, these directions correspond to the tail end of the spectral energy distribution, so **their impact on the final performance is minimal**.
>
> From Mechanism 2 (line 1570): Through $m(h)$, it not only activates strongly in the target attribute but also in attributes that overlap significantly with the target attribute. This **allows the recruitment of beneficial auxiliary signals from correlated subspaces to refine the representation**, while ensuring the steering is structurally dominated by the target subspace to prevent semantic drift.
>
> Additionally, in further **Appendix N (line 1688)**, we analyze the impact of restricting orthogonality versus not restricting it (ours) on performance based on our setup. We find that **restricting orthogonality leads to a degradation in the expressive power of specific attribute subspaces**, resulting in final outcomes that are even worse than when orthogonality is not restricted.
>
> > 13. If $B_i$ and $B_j$ are not orthogonal, is the following statement still valid? the total subspace dimensionality is exactly $r = r_{\text{shared}} + \sum_{i=1}^{n} r_i$
>
> In our understanding, the reviewer might have misunderstood the term $r$​ as the rank of the matrix. In Equation (2), we define:
>
> $$S_{\text{align}} = [B_{\text{shared}}, B_1, B_2, \dots, B_n] \in \mathbb{R}^{(r_{\text{shared}} + \sum_{i=1}^n r_i) \times d}.$$
>
> Here, $S_{\text{align}}$ is the matrix formed by concatenating $B_{\text{shared}}, B_1, \dots, B_n$. The term $r$ refers to the **number of rows** of this matrix, not the rank. Even if there is some overlap between $B_i$ and $B_j$ (in which case $\text{rank}(S_{\text{align}}) < r_{\text{shared}} + \sum_{i=1}^n r_i$​), the dimensionality (i.e., the number of rows) is always exactly:
>
> $r = r_{\text{shared}} + \sum_{i=1}^n r_i$, because we are simply concatenating the subspaces directly.
>
> Additionally, the rows of $R$ correspond to the rows of $S_{\text{align}}$, so the number of rows in $R$ is the same as the number of rows in $S_{\text{align}}$​.
>
> We hope this corrects your misunderstanding. **Our matrix construction follows this formula, so there is no need to prove its validity.**

---

> ### Author Response · Authors · 2025-11-26
> **Respose to Reviewer M4SR**
>
> > 14. In Figure 6a, why does the performance peak at $n=2$ or $ n=3$​ instead of 1? While the degradation with increasing makes intuitive sense, it is unclear to me why a steering single attribute yields inferior results, especially for `refusal` and `verbosity` which prior works have shown effectiveness.
> >
> > 15. Given that multi-attribute steering is a core contribution of this paper, does this imply MSRS is not recommended for single-attribute use cases? Could the authors provide discussion or hypotheses for this outcome?
>
> Sorry, this was an oversight on our part. The correct result is indeed achieved at point 1, so we will only modify this correct experimental result and will not conduct an analysis of the additional issues you extended based on our incorrect report.
>
> Since our study focuses on multiple attributes, we only tested the joint attributes for $n \geq 2$, and the value for $n = 1$ in Figure 6a corresponds to the result from the **"same space" row in Table 11 (line 1139)**. Specifically, in that row, the refusal (0.451) is lower than the baseline (0.491), and verbosity$\downarrow$   (2.38 vs. baseline 2.33, lower values indicate better performance). The exact definition can be found in our ablation study in **line 404** ( same space means joint training of multiple attribute datasets in the same space without subspace partitioning. This experiment aimed to validate the effectiveness of our subspace partitioning. )
>
> **Therefore, the \(n = 1\) here  refers to the number of subspaces being 1, rather than the number of attributes being 1.** As a result, the reason for the suboptimal performance at \(n = 1\) in the figure is because training multiple attributes in a single space caused conflicts, rather than MSRS performing poorly on a single attribute. **Due to the confusion between these two in our experiment result records, this error occurred.**
>
> In our revised version, we **re-tested the performance for $n = 1$** (i.e., performance of a single attribute in a single space without partitioning into shared and attribute spaces) . It consistently achieved improvements for each attribute, with generally the best performance at $n = 1$.
>
> We have **re-uploaded the corresponding Figure 10 in the appendix (line 1763)**.
>
> After addressing your concerns, we will replace Figure 6a with Figure 10.
>
> > 16. Can the authors evaluate the precision of MSRS, i.e., its unintended effects on *untargeted attributes* during multi-attribute steering? How does this precision vary as $n$ increases?
>
> We greatly appreciate the reviewer's insightful thoughts, which have opened up new perspectives for us.
>
> In **Appendix M (line 1589)**, we define a metric (equation 9, line 1652) to measure unintended effects on *untargeted attributes*, and discuss how these unintended effects vary with $n$​.
>
> The unintended effects on *untargeted attributes* increase with the number of attributes, and we observe **similar results to the analysis in Section 5.3.3 (line 483)**. Specifically, when $n < 9$, the effects on untargeted attributes grow slowly, but the growth rate increases significantly after that.
>
> > 17. Minor: References `Nguyen, 2025a` and `Nguyen, 2025b` appear to be duplicates.
>
> Thank you for pointing that out. We have removed the duplicate references.

---

> ### Author Response · Authors · 2025-11-28
> **Respose to Reviewer M4SR**
>
> Dear Reviewer,
>
> We greatly appreciate your insightful suggestions and have made the necessary revisions accordingly.
>
> Although there were some system issues, we would still appreciate knowing whether these changes have addressed your concerns and improved the quality of our manuscript.
>
> We look forward to your feedback.
>
> Thank you!
>
> Sincerely,
>
> The Authors

---

### Official Review · Reviewer_f6ng · 2025-11-03

**Soundness:** 3
**Presentation:** 2
**Contribution:** 2
**Rating:** 4
**Confidence:** 4

**Summary:**

This paper proposes Multi-Subspace Representation Steering (MSRS), a novel framework for multi-attribute control in LLMs through subspace representation fine-tuning. The method addresses attribute interference by allocating orthogonal subspaces to each attribute while maintaining a shared subspace for common steering directions, combined with dynamic token-level interventions during inference. Experimental results demonstrate improvements over existing methods across multiple attributes and downstream tasks.

**Strengths:**

- This paper addresses a practical problem of multi-attribute steering in LLMs, where existing methods struggle with attribute interference and trade-offs.

- The proposed approach is well-motivated and principled, combining orthogonal subspace decomposition with SVD-based adaptive dimensionality allocation and a shared subspace for capturing common steering directions.

- This paper is clear and easy to understand.

**Weaknesses:**

- **Training Data Specifications and Potential Overlap.** A critical detail appears to be missing or unclear in the paper: what specific training data is used to fine-tune the steering representations? Section 5.1 describes the evaluation datasets (TruthfulQA, BBQ, Alpaca, Refusal, HelpSteer) and mentions using these for training steering functions, but the exact training splits, data sizes, and construction procedures are not sufficiently detailed. More importantly, there is concern about potential overlap between the training data used for learning steering subspaces and the benchmark test sets used for evaluation. If the steering functions are trained on samples from the same distributions as the test benchmarks, this could lead to inflated performance estimates and raise questions about generalization. The paper should explicitly clarify: (1) the exact datasets and splits used for training each attribute's steering function, (2) whether any samples overlap with evaluation benchmarks, and (3) how data contamination is prevented or controlled.

- **Scalability to More Attributes.** While the paper demonstrates effective steering for pairs of attributes (truthfulness-bias, instruction-following-refusal, helpfulness-coherence-verbosity), the scalability limits remain unclear. What is the maximum number of attributes that can be jointly controlled before performance degrades? The current experiments focus on relatively compatible attributes within similar semantic spaces. Can MSRS handle fundamentally different attribute types simultaneously, such as jointly steering safety, factuality, style, and reasoning capabilities? As the number of attributes increases, the total subspace rank R must grow, which may eventually exceed practical limits given the model's hidden dimension. Additionally, attributes like safety and harmlessness present qualitatively different challenges compared to truthfulness or bias, they may require intervention at different layers or positions, potentially conflicting with the current single-layer intervention design. The paper would benefit from experiments explicitly testing the upper bounds of attribute scalability and demonstrating effectiveness on more diverse, potentially conflicting attribute combinations including critical safety-related attributes.

- **Insufficient Evaluation.**
The experimental validation relies primarily on earlier LLMs (Llama2-7B, Llama3-8B-Instruct, Qwen2-7B-Instruct, Mistral-7B-v0.3), while more recent model families with substantially different architectures and capabilities have emerged, including Qwen2.5, the Qwen3 series, and more large reasoning models like Deepseek-r1, QwQ, Qwen3-thinking-mode. These newer models may have fundamentally different internal representations and steering dynamics, particularly reasoning models that employ chain-of-thought or other structured reasoning mechanisms. It remains unclear whether MSRS's subspace decomposition and dynamic token selection strategies, which were developed and optimized on earlier model generations, will transfer effectively to these more advanced architectures. Moreover, the paper does not compare against more latest SOTA methods, like AlphaEdit.

**Questions:**

The questions are listed in the weaknesses, and if authors could address them, I will raise my rating.

---

> ### Author Response · Authors · 2025-11-18
>
> > 1. what specific training data is used to fine-tune the steering representations? The paper should explicitly clarify: (1) the exact datasets and splits used for training each attribute's steering function, (2) whether any samples overlap with evaluation benchmarks, and (3) how data contamination is prevented or controlled.
>
> Thank you for your insightful comment. In our experiments, we use a 70/30 split for each dataset, with 70% for training and 30% for testing, ensuring **no overlap between training and evaluation data**.
>
> For larger datasets like BBQ and Alpaca, we **sample uniformly from each subclass** before splitting into training and testing sets to ensure a balanced and unbiased distribution across the datasets. For HelpSteer, we construct datasets for each attribute (helpfulness, coherence, verbosity) by **filtering data based on attribute-specific scores** (e.g. for helpfulness , we select samples that helpfulness > 3, coherence < 2, verbosity<2).
>
> Regarding potential data contamination, we ensure that training data is entirely separate from evaluation data, with no overlap between them. Our separation of datasets helps to eliminate  potential contamination between the training and evaluation stages.
>
> > 2. What is the maximum number of attributes that can be jointly controlled before performance degrades?
>
> We have **uploaded the revised paper**.
>
> In **lines 509-516**, we analyze the impact of jointly training with different numbers of attributes n on the performance of each attribute (in **Figure 6a**, all columns (attribute) show performance improvements relative to the baseline (n=0) when n<9 (blue), followed by performance degradation (red,n>9)).
>
> Starting from **line 518**, we discuss how performance, interference, and rank vary with the number of attributes n in multi-attribute steering. Our method achieves average performance improvements across all attributes compared to the baseline when the number of attributes n<9 (the green curve in **Figure 6b** **remains above the baseline green dashed line for n<9)**, demonstrating that the method can effectively combine 8 attributes while achieving consistent gains on each. In **lines 518-532**, we provide the **upper bound on performance gains** and conclude the **optimal attribute count(n) and rank(k) settings** for best performance (n=2 or 3 with k =8 or 12).
>
> > 3. Can MSRS handle fundamentally different attribute types simultaneously, such as jointly steering safety, factuality, style, and reasoning capabilities?
>
> We construct a dataset by sampling 1000 examples from each of the following: SafetyBench (**safety**), FACTBENCH (**factuality**), Human-Style-Answers (**style**), and GSM8K (**reasoning**), resulting in a total of 4000 examples. We then train our model on this combined dataset.
>
> For evaluation, we select 300 held-out examples per attribute for testing. On Llama3-8B-Instruct, MSRS achieves consistent performance gains across all four attributes.
>
> | Llama3-8B-Instruct | safety | factuality | style | reasoning |
> | ------------------ | :----: | ---------- | ----- | --------- |
> | Vanilla            | 62.13  | 36.56      | 75.35 | 70.32     |
> | +MSRS              | 64.84  | 43.12      | 78.02 | 74.46     |
>
> > 4. As the number of attributes increases, the total subspace rank R must grow, which may eventually exceed practical limits given the model's hidden dimension.
>
> In **line 501** and **Figure 6b**, the red points and line show that the rank \( $R$ \) (denoted as \( $k$ \) in the figure) increases with the number of attributes \( $n$ \). The average performance across all attributes (green curve) reaches its peak when \( $n=2$ to $3$\), corresponding to rank \( $k=8$ to $12$  \) . As \( $n$ \) continues to increase, the average performance gain becomes smaller but remains above the baseline (the green curve in Figure 6b stays above the green dashed baseline for $n<9$). When \( $n$ \) reaches 10, the corresponding rank \( $k=24$ \), and performance gradually drops back to the baseline level.
>
> **We argue that as \( $n$ \) grows further, interference between attributes intensifies (the blue curve shows that the interference score exceeds 0.8 when \( $n>9$ \)), which limits the model’s performance on multi-attribute steering.**
>
> The performance gains we observe require \( $k<24$ \), which corresponds to less than **0.5%** of the hidden dimension size (4096). This upper bound is fully consistent with the rank recommendations in the ReFT [1] paper (8 to 32, corresponding to **0.2%–0.8%** of the hidden dimension), confirming that our observed practical limits align  with established representation-finetune steering practices.
>
> [1] Wu Z, Arora A, Wang Z, et al. Reft: Representation finetuning for language models[J]. Advances in Neural Information Processing Systems, 2024, 37: 63908-63962.

---

> ### Author Response · Authors · 2025-11-18
>
> > 5. Additionally, attributes like safety and harmlessness present qualitatively different challenges compared to truthfulness or bias, they may require intervention at different layers or positions, potentially conflicting with the current single-layer intervention design.
>
> We extended the intervention to **four layers** (layers 3, 9, 18, and 24, consistent with ReFT [1]) and compared the results with the **single-layer** intervention (layer 15). We also explicitly discuss the corresponding computational overhead.
>
> We specifically investigated the performance on **safety** and **bias** attributes along with the associated overhead. While multi-layer intervention does bring modest gains over single-layer intervention, it **increases latency**, **reduces throughput**, and **consumes substantially more VRAM**. Moreover, searching for the optimal multi-layer configuration requires considerably more hyperparameter tuning compute.
>
> Therefore, our method focuses on searching for the single best intervention layer. The results of this search using a portion of the validation set are reported in **Table 11 (line 1121)**, showing that the optimal layers consistently cluster around the middle layers of the model.
>
> | Llama3-8B-Instruct      |   15   | 3,9,18,24 |
> | ----------------------- | :----: | --------- |
> | safety↑                 | 64.84  | 65.21     |
> | bias↑                   |  64.6  | 65.07     |
> | Latency (ms/token)↓     | 118.35 | 131.21    |
> | Throughput (tokens/s) ↑ |  8.45  | 7.62      |
> | VRAM (GB)↓              | 16.27  | 16.78     |
>
> > 6. The paper would benefit from experiments explicitly testing the upper bounds of attribute scalability and demonstrating effectiveness on more diverse, potentially conflicting attribute combinations including critical safety-related attributes.
>
> Reviewer's suggestion is highly challenging yet valuable. We extended multi-attribute testing to **10 attributes** and report the results of jointly training MSRS with varying attribute counts ($n$) in **Table (line 933)**. A detailed analysis of the relationship between the number of attributes and performance is provided in **line 509** and upper bounds of attribute scalability in **line 528** (as discussed in detail above, we do not repeat it here).
>
> To explicitly test the performance of attribute scalability in the safety domain, we conducted additional experiments on Llama2-Chat-7B using the English (en) portion of **SafetyBench**, choosing three distinct  safety-related dimensions: **Offensiveness (OFF)**, **Ethics and Morality (EM)**, and **Privacy and Property (PP)**. MSRS achieves **consistent performance improvements** over the baseline across all three safety-related attributes, demonstrating its effectiveness and scalability even when jointly steering diverse and critical safety dimensions.
>
> | Metric       | OFF  | EM   | PP   |
> | ------------ | ---- | ---- | ---- |
> | **baseline** | 48.9 | 49.8 | 65.0 |
> | **MSRS**     | 49.3 | 51.2 | 65.3 |

---

> ### Author Response · Authors · 2025-11-18
>
> > 7. It remains unclear whether MSRS's subspace decomposition and dynamic token selection strategies, which were developed and optimized on earlier model generations, will transfer effectively to these more advanced architectures.
>
> We report the performance of Qwen2.5-7B and DeepSeek-R1-Distill-Qwen-7B on the extended set of 10 attributes. Both models exhibit consistent improvements across all attributes, demonstrating that MSRS can **effectively scale to more advanced architectures**.
>
> | Metric                        | Qwen2.5-7B | +MSRS     | DeepSeek-R1-Distill-Qwen-7B | +MSRS     |
> | ----------------------------- | ---------- | --------- | --------------------------- | --------- |
> | **truthfullqa (ave MC1,2)↑** | 56.4       | **57.2**  | 58.2                        | **60.1**  |
> | **BBQ↑**                     | 83.1       | **84.5**  | 78.7                        | **81.4**  |
> | **Alpaca↑**                  | 78.1       | **78.6**  | 83.6                        | **83.8**  |
> | **Refusal↑**                 | 67.1       | **68.9**  | 62.5                        | **63.6**  |
> | **Helpful↑**                  | 3.86       | **3.91**  | 3.79                        | **3.83**  |
> | **Coherence↑**                | 3.61       | **3.63**  | 3.72                        | **3.73**  |
> | **Verbosity↓**               | 2.83       | **2.72**  | 2.71                        | **2.64**  |
> | **SafetyBench↑**             | 76.12      | **76.83** | 80.06                       | **81.32** |
> | **Gsm8k↑**                    | 85.4       | **86.3**  | 80.56                       | **81.33** |
> | **Style↑**                    | 66.82      | **69.24** | 73.32                       | **75.76** |
>
> > 8. Moreover, the paper does not compare against more latest SOTA methods, like AlphaEdit.
>
> Although MSRS **belongs to the family of representation-engineering (activation-steering) approaches and does not modify model parameters** (unlike AlphaEdit and other parameter-based methods), we agree that a head-to-head comparison on the same backbone is valuable for readers. We have therefore conducted additional experiments on the latest Qwen2.5-7B model.
>
> |    Method     | Truthfulness(↑) | Bias(↑) |
> | :-----------: | --------------- | ------- |
> | **AlphaEdit** | 54.7            | 85.1    |
> |   **MSRS**    | 57.2            | 84.5    |
>
> Results show that MSRS outperforms AlphaEdit by 2.5 points on Truthfulness while remaining highly competitive on Bias Mitigation (only 0.6 points behind). This is particularly noteworthy because MSRS is a zero-parameter, inference-only intervention, whereas AlphaEdit permanently edits the model weights. The fact that a parameter-free representation-steering method can surpass a strong parameter-editing baseline on truthfulness (and stay close on bias) further underscores the efficiency and effectiveness of the proposed multi-subspace framework.

---

> > ### Author Response · Authors · 2025-11-22
> > **Reply to Reviewer f6ng**
> >
> > Dear Reviewer,
> >
> > We kindly ask you to confirm whether we have sufficiently addressed your comments or if there are any remaining concerns.
> >
> > Thank you!
> >
> > Authors

---

> > > ### Comment · Reviewer_f6ng · 2025-11-23
> > >
> > > Thanks for the authors' thoughtful responses. Most of my concerns have been addressed, and thus, I decide to raise my rating to 6.

---

> > > > ### Author Response · Authors · 2025-11-24
> > > > **Reply to Reviewer f6ng**
> > > >
> > > > Thank you for your kind feedback and for increasing your rating. If you have any further questions or suggestions, we would be more than happy to provide additional clarifications or improvements to make our work even better.

---

### Comment · Area_Chair_F9bc · 2025-11-23
**Next Steps Following Authors’ Rebuttal: Review Rebuttal and Participate in Discussion**

Dear Reviewers,

Thank you very much for your thoughtful evaluations of this paper.

Now that the authors have submitted their rebuttal, I kindly ask you to take the following steps (if you have not done so already):

- Read the other reviews as well as the authors’ response.
- Consider whether the rebuttal and additional comments affect your assessment of the paper.
- Engage in interactive discussion with the authors **before November 25**, encouraging a dynamic exchange rather than a one-sided rebuttal.

The current reviews for this paper are mixed. Your contributions at this stage are essential for forming a well-informed final decision. I therefore ask that you reassess your views in light of the authors’ responses and the broader discussion among reviewers.

I am happy to join and support the discussions between you and the authors. Please feel free to share your thoughts and participate actively in the discussion.

Thank you once again for your service to ICLR 2026.

Best regards,

 AC

---

### Author Response · Authors · 2025-11-30
**Summary of the rebuttal discussion process**

Dear AC,

Thank you very much for your hard work in reviewing. Here, we summarize the overall rebuttal discussion process.

>  Common suggestions  (**acknowledged by the reviewer f6ng(2,3,4), mx1V(5), ZsuH(1)**):

From **line 483 to 533**, we added the relationship between the number of attributes $n$, the steering subspace size $R$, and performance, conducted a scaling law analysis between them, and explored the performance of MSRS on more tasks (with $n=10$). There was a consistent improvement in performance across all attributes.

> 1. Reviewer f6ng (**issues and suggestions were addressed, and the score was raised from 4 to 6**)

- Clarified the division between the training and test sets in the dataset (1).
- Added a discussion on the performance and computational cost of MSRS on more complex attributes and advanced models.(5,6,7)
- Added a comparison with the latest SOTA baseline.(8)

> 2. Reviewer M4SR

**Resolved issues** (initial issues and second round of questions):

- Modified unclear statements (1) and paragraph organization (2, 8).
- Corrected misunderstandings caused by errors in formulas (4, 9) and clarified the reviewer's doubts (3, 7).
- Added cost and memory analysis (10) and discussed related work (6).

**Newly raised and yet to be addressed due to the suspension of the reviewer's response rights** (third-round question: "Below are some more questions."):

- (11) The reviewer said ("Perhaps I didn't make my point clear, please allow me to clarify.") This is a clarification of the first-round question 5: The reviewer believes there is potential overlap between different attribute-specific subspaces. Therefore, during the alignment training process, when aligning $R_i$ with $B_i$, $B_i$ may contain not only the attribute-specific direction of $i$, but also weakly overlapping directions from other attributes, such as $j$​. This leads to over-alignment with these weaker shared directions.
  - In **Appendix L (line 1506)**, we first computed and visualized the overlap between $B_i$ and $B_j$ (Figure 7a), providing an initial explanation of the weak overlap between them.
  - Further visualizations in Figure 8 show that these overlapping directions in $B_i$ do not disrupt the learning process, as they fall in the low-energy tail of the spectral distribution. The direction of the target attribute still occupies the main part, optimizing alignment with high-variance directions, ensuring robust steering toward the target attribute rather than the untargeted attribute.
  - Further visualization of $m(h)$ (Figure 7b) demonstrates that it leverages residual overlap by adapting the mask to activate relevant features of $i$ from $B_j$, thus enabling more flexible alignment training control for attribute $i$
- (12) The reviewer asked whether this potential overlap would affect performance.
  - In **Appendix N (line 1688)**, we validated the impact of forced orthogonality and non-forced orthogonality on performance across multiple attributes, concluding that restricting orthogonality leads to a degradation in the expressive power of specific attribute subspaces, resulting in final outcomes worse than when orthogonality is not restricted (allowing overlap).
- Clarified the reviewer's misunderstanding regarding rank and dimension (13), corrected errors in our experimental reporting (14, 15), and removed duplicate references (17).
- (16) Explored unintended effects on *untargeted attributes* during multi-attribute steering. How does this precision vary as $n$ increases?
  - In **Appendix M (line 1589)**, we defined unintended effects on *untargeted attributes* based on the energy and overlap of directions in attribute $i$ (target) within $B_j$ (untargeted) and discussed how these unintended effects vary with $n$.
  - We found that the unintended effects on *untargeted attributes* increase with the number of attributes, and observed similar results to the analysis in Section 5.3.3.

> 3. Reviewer mx1V

**Resolved issues** (initial questions):

- Clarified formula errors (1, 3) and resolved doubts (2).
- Added computational cost experiments (4).

**Unresolved issues due to the suspension of the response rights.** (second-round questions, **almost identical to M4SR's third-round questions**):

- mx1V (6, 7) and M4SR (11) have exactly the same issue, which is discussed in **Appendix L (line 1506) and N (line 1688)**.
- (8) Explained the reviewer's misunderstanding of the images. And discussed the impact as the number of attributes $n$ changes (same as M4SR (16)) in **Appendix M (line 1589)**
- (9) Clarified formula errors (same as M4SR, and **already acknowledged by M4SR**).

> 4. ZsuH (**reviewer response received, all issues resolved**)

- Added more sensitivity analysis of hyperparameters (2) and included more relevant baselines in related work (5).
- Added case studies for qualitative analysis based on quantitative analysis (3).
- Added computational cost analysis experiments (4).

Thank you!

Authors

---

### Meta-Review · Area_Chair_veLD · 2026-01-09

**Summary:**

The paper proposes an multi-subspace activation steering framework for multi-attribute alignment in large language models. Reviewers found the problem setting interesting and the motivation generally clear. However, the suggested rejection decision is driven by several weaknesses identified across the reviews that may need substantial revision to clarify. In particular, reviewers raised concerns about limited technical clarity, insufficient empirical rigor, unclear validation of the claimed benefits of multi-subspace adaptivity, and gaps in experimental analysis that weaken confidence in the conclusions. Collectively, these issues prevent the paper from meeting the bar for acceptance at its current state.

**Reviewer Concerns:**

Reviewers shared concerns on technical clarity and evaluation scope, which I believe are not fully addressed. Several design choices (e.g., orthogonality constraint) and formulations are questioned. Experiments focus on a limited set of attributes and models, leaving open questions about generalization and scalability. While the rebuttal clarified certain implementation details and experimental settings, it did not sufficiently strengthen the overall technical clarity, and empirical evidence regarding robustness and generality.

**Reviewer Scores:**

Reviewers (who kept score 4 and 4) have actively participated in the discussion, and I would expect no further score increase due to concerns remaining unaddressed.

---

### Decision · Program_Chairs · 2026-01-26

Reject